# Punctuated decline of human cooperation

Nicholas Sabin[1,2,3 ✉], David Klinowski[4 ✉] & Felix Reed-Tsochas[2,3 ✉]

Human cooperation is dynamic and often declines even under favourable conditions[1–4]. Many prevailing theories explain the decrease of cooperation in terms of strategic behaviour or learning, framed as evidence of rational behaviour or progression towards rationality[5–9]. Here we show that a key source of long-term decline derives from deviations from rational behaviour that systematically vary over time. We analyse a natural social dilemma in the field—that is, group lending in Sierra Leone—tracking cooperative dynamics over a five-year period. Borrowers enter a joint-liability contract, structured so that if the group loan is not repaid in full, all members lose access to future credit[10]. This produces a threshold social dilemma with incentives to free-ride[11,12]. The dataset includes 47,931 group payments made by 7,108 borrowers, augmented with a two-stage cluster sample of semi-structured interviews. We find a statistically robust pattern of punctuated decline driven by behavioural mechanisms[13]. Cooperation rates start out high but gradually decline due to decreases in group members' cooperative motivation and effort. Sharp rebounds occur when loans are restarted and clients resensitized to their cooperative responsibilities, even though the group membership and dilemma structure are largely unchanged. This pattern persists over the five-year observation window, but with each successive restart the subsequent decline is more rapid. The findings have direct implications for preventing behavioural decline in cooperative programmes and institutions.

Human cooperation is widespread but fragile, with long-term outcomes ranging from sustained cooperation to collapse[1,3,14]. Extensive research has identified factors that make cooperation more likely, such as the ability to communicate[2,15], selective interaction[16,17], observability of contributions or reputation[18–20], and the opportunity to punish[21,22]. However, a question that remains disputed is why, even under favourable conditions, cooperation is dynamic and prone to decline[4,23,24].

It is now widely accepted that humans have heterogenous cooperative behaviours, with some effectively characterized as unconditionally self-interested, others unconditionally cooperative, and a majority as conditionally cooperative[25–27]. However, explanations diverge as to why groups consisting of such actor types often start out cooperating but decline over time. Most theories based on strategic behaviour (such as backward induction[5,7] or calculated enticement[6,28]) and learning (such as social learning[8,29] or confusion followed by error reduction[9,28,30,31]) suggest that decline arises either from rational behaviour itself or from the progression towards rationality. An alternative line of thought is that behavioural mechanisms are a key source of the decline in cooperation. Behavioural in this sense refers to psychologically realistic explanations typically associated with deviations from unbounded rationality, pure self-interest and complete self-control[13,32]. Although numerous behavioural biases have been identified in the cooperative context[33–37], the extent to which systematic behavioural deviations shape long-term dynamics in social dilemmas remains unclear.

Here we analyse a social dilemma in a field setting that systematically tracks cooperative behaviour over a five-year period. The empirical context is group lending in Sierra Leone. Borrowers enter a joint-liability contract such that if the group loan is not repaid in full, all group members are held financially responsible and lose access to future credit[10,38]. The formal expectation is that "on each repayment date, each group member will need to immediately pay for any members that are late with their repayments" (from the policy manual of the lending institution). In practice, group members do compensate for each other and apply informal sanctions to enforce cooperation, such as social pressure, embarrassment and ostracism[38]. This structure produces a threshold social dilemma with multiple equilibria[11]. The cooperative conflict lies in the incentive for each member to personally contribute as little as possible while maintaining provision of the collective good—joint access to progressively larger amounts of credit. Groups typically consist of five members. The quantitative dataset includes 47,931 group-level payments made by 1,589 groups with a total of 7,108 members. The statistical analysis of this dataset is augmented with a qualitative analysis of 73 in-depth interviews.

The setting and research design offer several unique benefits. The first is long-term stability: group members face the same dilemma on a monthly basis for up to five years in the dataset. The loan structure and group membership are largely stable. Much empirical research on cooperation is conducted in laboratory settings on a timescale of minutes or hours[16]. Over the span of years, behaviour may undergo substantially different dynamics. The second benefit is outcome measurement: a common challenge of research in natural settings is inconsistencies in outcome measurement[39]. Here, participants face high-stakes financial and social tradeoffs that are integral to their daily lives, but the economic incentives and cooperative outcomes remain well-defined. We are able to define two objective measures of cooperation: the financial contribution rate and the cooperative effort rate, both of which enable

[1]Facultad de Administración y Economía, Universidad de Santiago de Chile (USACH), Santiago, Chile. [2]Saïd Business School, University of Oxford, Oxford, UK. [3]CABDyN Complexity Centre, University of Oxford, Oxford, UK. [4]Department of Economics, William & Mary, Williamsburg, VA, USA. ✉e-mail: nicholas.sabin@usach.cl; dklinowski@wm.edu; felix.reed-tsochas@sbs.ox.ac.uk

robust interpretation of statistical patterns. The third is cooperative mechanisms: the structure of the lending environment is particularly well-suited for discriminating between potential mechanisms underlying the longitudinal patterns. For example, the same group may go through multiple loan cycles in sequence. This provides groups with the chance to 'play the game' again. Restarts, as well as other features of the data, provide powerful tools for discriminating between alternative causal explanations[28,40]. The fourth is in-depth interview data: the quantitative data are augmented with detailed descriptions of the cooperative dynamics in the clients' own words, substantiating the underlying mechanisms. We used a two-stage cluster sampling design to select a representative sample of clients[41]. We also included a purposive sample of clients and staff to cross-validate and strengthen the internal validity of the findings[42].

We find that cooperation is prone to decline, but when viewed on a longer timescale the pattern is clearly non-monotonic and reflects a dynamic process of punctuated decline. Sharp rebounds occur when loans are restarted and clients are resensitized to their cooperative responsibilities and long-term implications, even though the group membership and dilemma structure are largely unchanged. The combined quantitative and qualitative analyses show that the primary cause of the systematic dynamics are behavioural mechanisms, where decays in effort and the motivation to cooperate over time lead to individual and collective inefficiency; that is, partially repaying a loan but still losing access to future credit. Our analysis argues against learning, strategic behaviour and changes in financial ability as causal explanations in this context. Instead, group members show shifts in their cooperative behaviour, which are directly referred to in interviews as becoming 'lethargic', 'relaxed' and 'tired of paying'. This produces an increased tendency to partially free-ride and often pushes groups towards collective failure.

Accounting for the effects of behavioural mechanisms over time can reshape our expectations for long-term cooperation in specific ways. We show that behavioural decline can be pervasive and persistent for up to five years even in a favourable cooperative setting. For example, groups that have successfully sustained cooperation over multiple loan cycles are still at risk of unravelling due to a decline in cooperative effort. Cooperation can be consistently—but temporarily—revitalized by ostensibly minor behavioural cues that do not alter the underlying social dilemma. The size of the immediate impact of a restart on cooperation rates increases over five years; however, the sustained impact of each restart is increasingly short-lived. As groups habituate to the behavioural cues, the rate of cooperative decline accelerates after each restart.

We analyse data from a microfinance institution in Sierra Leone that focuses on group lending. The joint-liability groups are formed through a self-selection process. However, members of the same group cannot be direct kin; that is, parents, spouses or siblings. Of the members, 73.3% are female. All members of the group receive an equal share of the loan. These amounts are substantial to the financial lives of the clients. Typically, a borrower's monthly repayment is approximately 25% of their expected monthly profits. Before all loan disbursements, group members formally undergo joint liability orientation by the institution to ensure that the clients understand the collective consequences. If the group does not repay the loan in full, all members lose access to future credit. Accordingly, the institution does not track or distinguish individual-level repayment contributions.

The group lending structure corresponds to a threshold social dilemma, in which the provision of a collective good is non-linear and occurs only when aggregate contributions reach a specified level[2,12]. In a threshold dilemma, individuals have an incentive to contribute if they believe their contribution will be pivotal to producing the collective good. Both zero-contribution and collective provision constitute equilibria, with the latter being Pareto superior[11,43]. The conflict between opportunistic behaviour and the collective interest arises when the threshold can be met with less than full contributions from all of its members. Individuals maximize their payoff by contributing less than their fair share while still benefiting from the good: a behaviour referred to as partial free-riding or cheap riding[11]. Over repeated interactions, this distributional conflict can undermine cooperation and ultimately lead to the collapse of the collective good. In-depth interviews with clients and staff confirm that participant understanding and practical implementation of group lending in the field appropriately corresponds to this cooperative framework (refer to the Methods and Supplementary Information for further theoretical background and empirical evidence).

The longitudinal structure of group lending has key similarities to that of laboratory experiments involving repeated social dilemmas. After a loan is disbursed, it is scheduled for monthly group repayments, analogous to rounds in the laboratory[16]. If a group repays its loan in full, it may qualify for another loan cycle. Eligibility is on the basis of payment amount and timeliness throughout the previous cycle. Before receiving the next loan and resuming group payments, groups are formally resensitized by the staff of the lending institution to their cooperative responsibilities and the collective implications of joint liability. Importantly, group loan membership is largely stable across cycles and the joint-liability structure remains the same.

We model a group's cooperative behaviour using two dependent measures: (1) the financial contribution rate and (2) the cooperative effort rate. The measures are complementary ways of capturing a group's cooperative outcomes; they are informative in their own right and jointly strengthen construct validity. The financial contribution rate is the percentage of the monthly scheduled amount paid by the group within 30 days. If groups are unable to make the full monthly payment on time, they may make partial payments. By contrast, the cooperative effort rate is based on the timeliness of the group's preliminary payment, which may be partial or full. The two measures are highly correlated, but the second has the advantage of confirming that the longitudinal patterns are not driven by systematic variation in group members' financial ability to pay (refer to Extended Data Figs. 1 and 2 for distributions of these variables over time). Hereafter, if not specified, the term cooperation rates refers to both the financial contribution rate and cooperative effort rate (refer to Extended Data Table 1 for descriptive statistics).

## Long-term dynamics of cooperation

The empirical results show a pattern of punctuated decline occurring systematically over the five-year period. Figure 1 shows aggregate trends overlaid on the full dataset. Cooperation rates start out high in the first round and gradually decay until the final round of the first loan cycle. After a restart, the rates rebound in the first round of the second cycle, but again commence a period of gradual decay. The pattern repeats over all loan cycles. As expected, the patterns of the financial contribution and cooperative effort rates are substantively similar. Not only do the cooperation rates decline across rounds within a loan cycle, but there is also significant decline across loan cycles, with each loan cycle having lower average cooperative behaviour than the previous cycle. Table 1 presents results that are based on group fixed-effects panel regressions. They confirm that the decline within and across loan cycles is statistically significant for both the financial contribution and cooperative effort rates.

When interpreting the longitudinal trends, it is important to consider that after each cycle the lending institution selectively excludes low-performing groups from subsequent loans. In the dataset, 1,589 groups received at least one loan cycle, but only 197 groups received five loan cycles. Successful groups may choose not to proceed to the next cycle, but more often the source of cycle-to-cycle attrition is the removal of groups with substantially delayed or incomplete repayment. Supplementary Table 1 provides a breakdown of group continuation

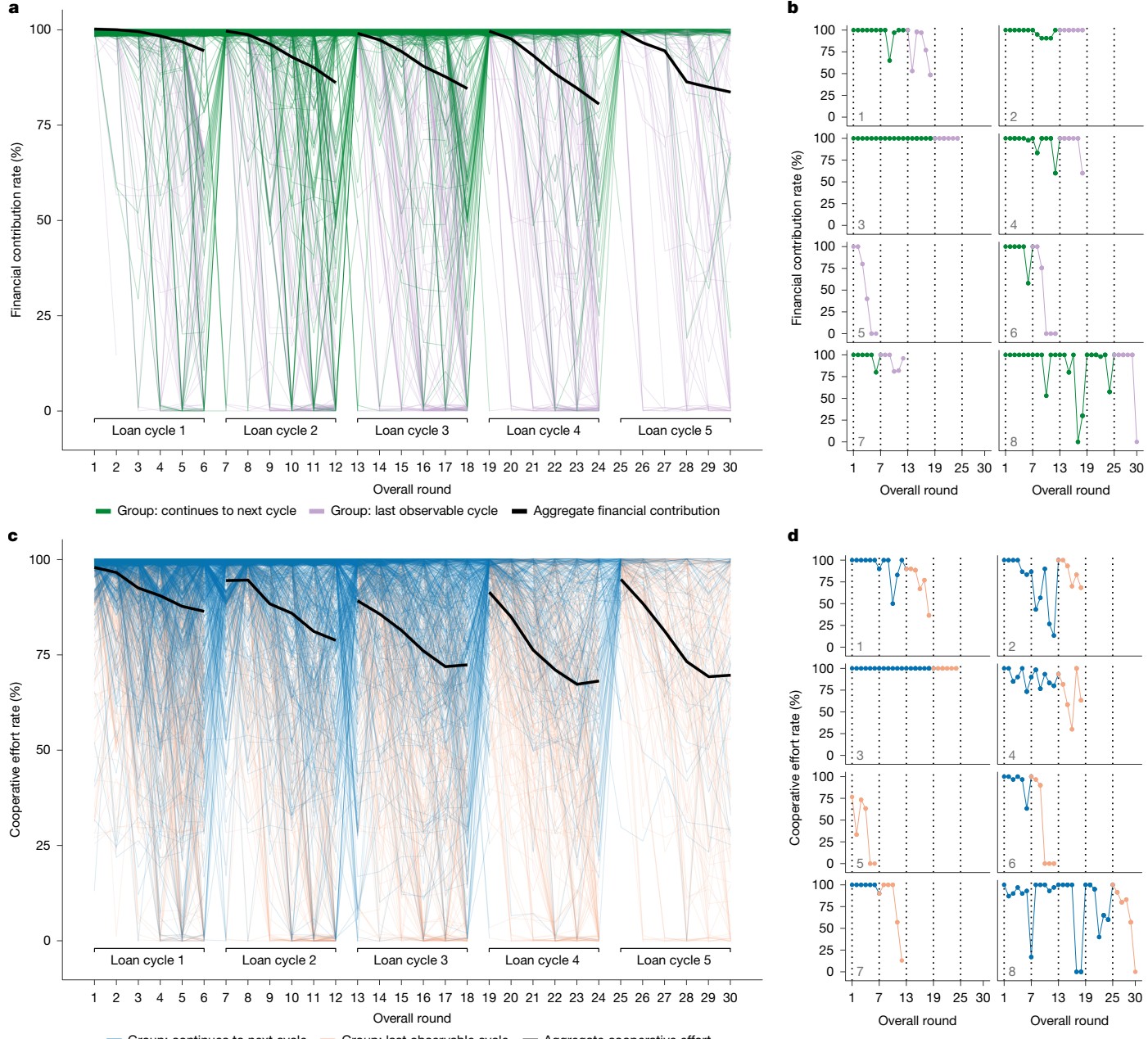

**Fig. 1 | Punctuated decline of cooperation in group lending over five years.**
Average cooperation gradually declines over time but sharply rebounds
when loans are restarted and members resensitized to their cooperative
responsibilities. The rate of decline accelerates after each restart. **a,c** Total
sample: $n = 31{,}199$ monthly measures of group cooperation, based on 47,931
partial or full payments, nested in 1,589 groups comprising 7,108 borrowers.
Rounds denote monthly scheduled group loan repayments. Each thin line
represents the cooperation rate of a single group over time (vertical jitter = 2).
The bold lines indicate the mean cooperation rate for the total sample. **a,b**, The
financial contribution rate is the percentage of the scheduled group payment
amount received within 30 days. **c,d**, The cooperative effort rate is based on
the number of days overdue before a group records any attempt at partial or
full payment. **b,d**, Single groups plotted separately in each subpanel (1–8) to
illustrate variation in underlying group-level behaviour. Vertical dotted lines
denote the first round of a loan cycle. **a–d**, Sample size decreases over loan
cycles (Methods). The colour of the lines indicates whether the group continues
to the next cycle or whether it is the group's last observable cycle in the dataset.
The pattern of punctuated decline is statistically robust to the attrition of
groups.

status and different sources of attrition across loan cycles. The pattern
of increasingly pronounced decline across loan cycles in Fig. 1 is more
striking given that failing groups are excluded from subsequent loans.

It is therefore crucial to examine how cross-cycle patterns look when
controlling for selective attrition. We conducted equivalent analyses
on subsets of data with fixed sample sizes; that is, no attrition. In Fig. 2
we plot average cooperation rates for fixed samples of groups that
survive at least a given number of cycles. For example, the top panel
restricts the sample to all groups that survived at least one cycle and

continued to cycle 2, the panel below restricts the sample to all groups
that survived at least two cycles and continued to cycle 3 and so on. In
general, even a fixed subset of successful groups that survive five cycles
shows a pattern of punctuated decline (see Table 1 for statistical results).

## The restart effect

To estimate the size of the rebound in cooperation rates follow-
ing a restart, we compare the average round-over-round change in

## Table 1 | Cooperation rates over time in group lending

| | Financial contribution rate | | Cooperative effort rate | |
|---|---|---|---|---|
| | All groups | Five-cycle groups | All groups | Five-cycle groups |
| | (1) | (2) | (3) | (4) |
| Round | −2.191 | −1.223 | −2.921 | −2.162 |
| | (0.097) | (0.148) | (0.101) | (0.179) |
| | [$P < 0.0001$] | [$P < 0.0001$] | [$P < 0.0001$] | [$P < 0.0001$] |
| Cycle | −4.682 | −4.911 | −6.382 | −6.750 |
| | (0.806) | (1.133) | (0.874) | (1.281) |
| | [$P < 0.0001$] | [$P < 0.0001$] | [$P < 0.0001$] | [$P < 0.0001$] |
| Number of groups | 1,589 | 197 | 1,589 | 197 |
| Number of observations | 31,199 | 8,006 | 31,199 | 8,006 |
| $R^2$ | 0.2916 | 0.2165 | 0.4108 | 0.3295 |

Coefficient estimates from ordinary least squares regressions estimating the group's financial contribution and cooperative effort rates in the round. The samples in columns 1 and 3 comprise all groups, whereas those in columns 2 and 4 comprise the subset of groups that received five cycles. Regressors are round and loan cycles. All regressions control for loan officer, loan size, loan duration, calendar year fixed effects, rainy season fixed effects and group fixed effects. Standard errors clustered at the group level in parentheses, and $P$ values from two-sided $t$-tests of the coefficient estimates in brackets.

cooperation rates across all rounds of a given cycle to the change in cooperation rates between the last round of the cycle and the first round of the following cycle, restricting the sample to groups that go through the given restart (Fig. 3). On average, cooperation rates decline round-over-round within loan cycles even when excluding groups that do not continue (financial contribution rate = −0.64%, $P < 0.0001$; cooperative effort rate = −0.97%, $P < 0.0001$) (Extended Data Table 2). In contrast to average decline, restarts lead to a substantial immediate increase in cooperation rates, with an average restart effect (across all four restarts) of 4.24% for the financial contribution rate ($P < 0.0001$) and 2.87% for the cooperative effort rate ($P < 0.0001$) (Extended Data Table 2). The effects are robust to the inclusion of controls and are, in fact, larger when controls are included (Extended Data Fig. 3 and Supplementary Table 2).

### Persistence and the rate of decline

Here we test statistically which features of the punctuated pattern of decline are persistent and which change systematically. First, the risk of cooperative failure is pervasive and persistent, as the percentage of groups whose cooperation rates collapse to zero grows within a cycle and across all cycles ($P < 0.0001$ for both round and cycle coefficients with the outcome indicator that the group contributes zero in the round; Extended Data Figs. 1 and 2). Second, the size of the restart effect noticeably increases with each restart for both cooperation rates (Extended Data Table 2 and Supplementary Tables 2–4). Third, following each restart, the average financial contribution rate resets to similar levels statistically, whereas the average cooperative effort rate tends to reset to slightly lower levels following each restart (Supplementary Table 5). Fourth, the average rate of decline within cycles is negative and accelerates after each restart. This is shown in Table 2, which estimates a significant negative interaction effect between rounds and cycles on cooperation rates ($P < 0.0001$ for both cooperation rates) for both the full sample and the subset of groups that participate in five loan cycles.

### Behavioural mechanisms

To deepen our understanding of the mechanisms underlying the statistical patterns, we call on a distinct type of data: 73 in-depth

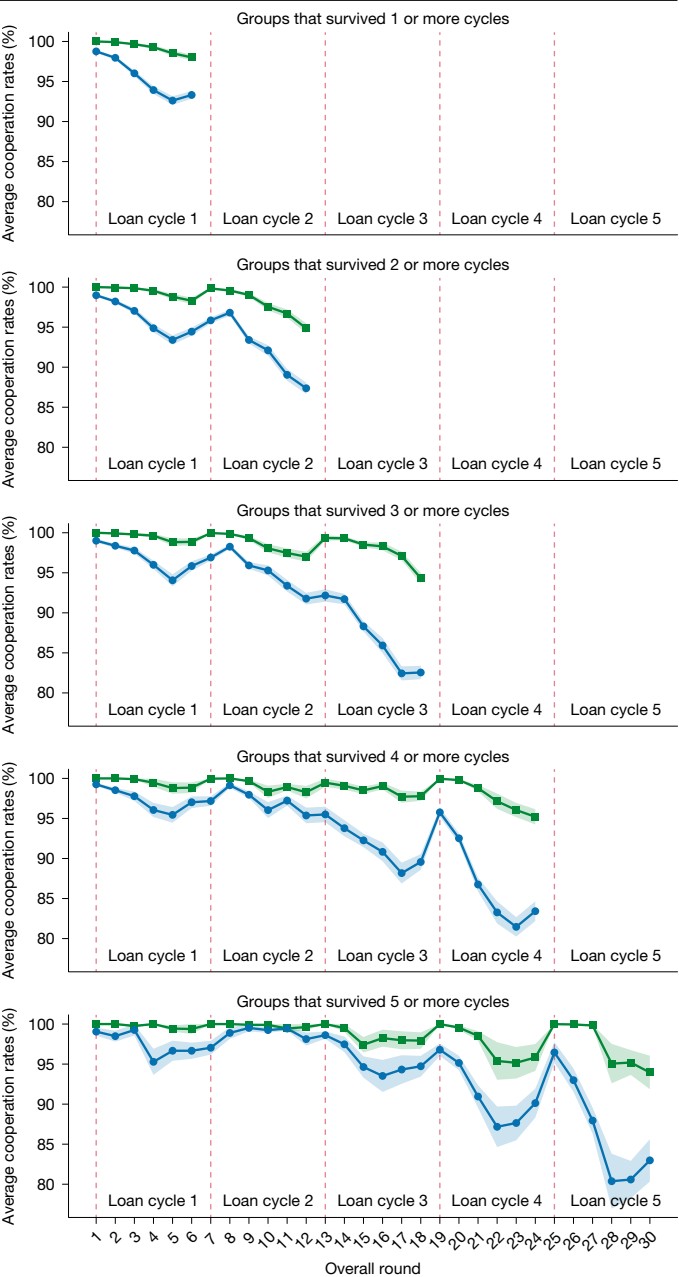

**Fig. 2 | Longitudinal cooperation rates in group lending with subsets for loan cycle survival.** Each panel removes the effect of group attrition over loan cycles by plotting the financial contribution (on the basis of financial amount) and cooperative effort (on the basis of timeliness) rates for groups that survived a specified number of cycles. The longitudinal patterns are robust to group attrition; this indicates that when failing groups are removed from the analysis, successful groups on average also feature a pattern of gradual decay and rebound (refer to Table 1 for statistical results). Subsets are based on all groups that survived a given cycle and continue to the next cycle. Subsample sizes per panel are denoted as $n$ = number of groups and $N$ = total observations: $n_1 = 1,026$, $N_1 = 6,494$; $n_2 = 741$, $N_2 = 9,636$; $n_3 = 462$, $N_3 = 9,858$; $n_4 = 197$, $N_4 = 5,998$; $n_5 = 53$, $N_5 = 2,094$. The vertical red dashed lines indicate the first round of a new loan cycle. Error ribbons denote ± s.e.m.

semi-structured interviews totalling 56 hours, which were conducted by the research team contemporaneously with the collection of the quantitative data. Interviewees were drawn from a representative sample of joint liability groups, which allowed us to assess the relative frequency of specific behaviours; this was complemented with a purposive sample of further group members and staff, which allowed us to

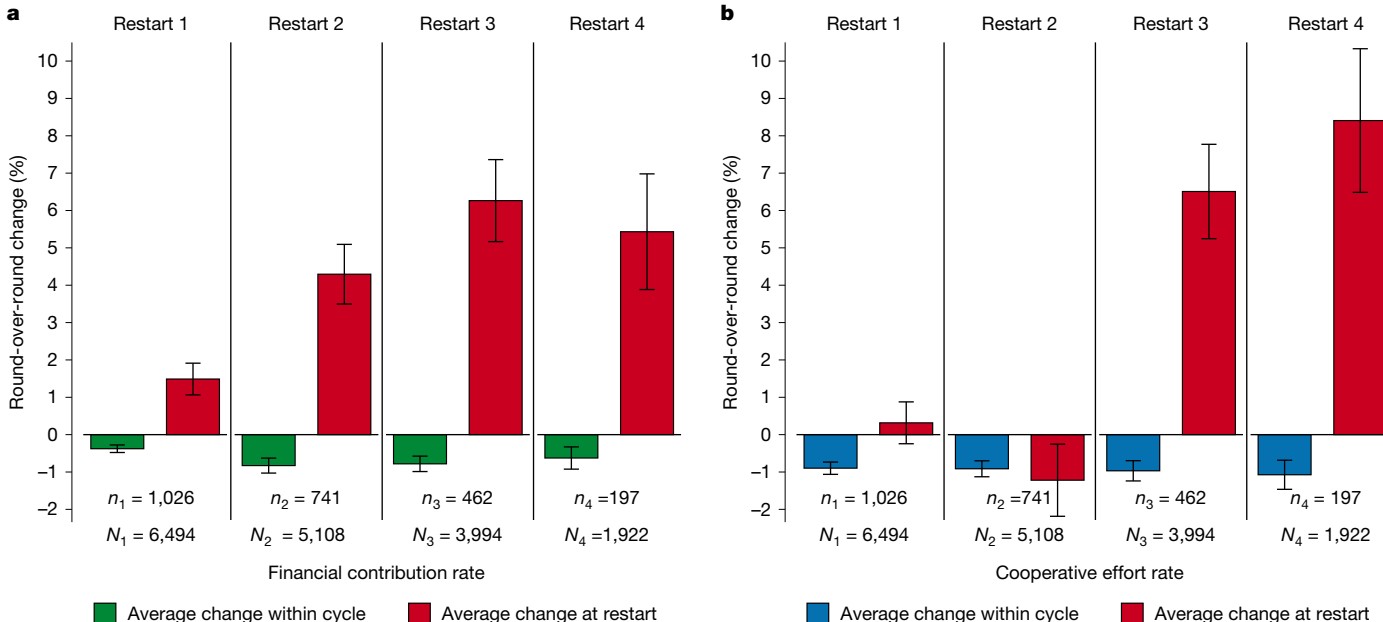

**Fig. 3 | Sequential restart effects on cooperation rates for continuing groups.** Cooperation rates decline round-over-round on average within loan cycles, but increase on average at restarts even when groups are stable. **a**,**b**, For any given restart number, the average change within a cycle is calculated as the within-group round-over-round difference in cooperation rates, averaged across all rounds in the loan cycle preceding the given restart and across all groups. The average change at the restart is calculated as the difference in cooperation rates between the final round of the preceding loan cycle and the first round of the following loan cycle. Both averages are based on subsamples

that are restricted to groups that go through a given restart to estimate effect sizes without attrition of groups. Subsample sizes: $n$, number of groups that go through a given restart; $N$, round-over-round total observations within cycle and at restart for a given restart. Error bars denote the mean ± s.e. **a**, The financial contribution rate is the percentage of the scheduled group payment amount received within 30 days. **b**, The cooperative effort rate is based on the number of days overdue before a group records any attempt at partial or full payment.

cross-validate descriptions and increase internal validity (Fig. 4a). The interviews provided illuminating accounts of clients' perceptions, motivations, and direct experiences of cooperation and defection. Extended Data Table 3 and Supplementary Tables 6 and 7 provide anonymized identifiers and descriptive statistics of the interview sample (Fig. 4b; refer to the Methods, Extended Data Tables 4–6 and Supplementary Information for details on the data collection and analysis process, the semi-structured interview protocol, tables of coded cooperative behaviour and extended quotes).

The qualitative analysis produced four insights into the mechanisms underlying group outcomes regarding: (1) diversity of cooperative motivation; (2) prevalence of distributional conflict and partial free-riding; (3) behavioural changes over time; and (4) heterogeneity in patterns of defection. We summarize the main findings here and provide representative quotes of the reported behaviours.

First, we found that borrowers reported diverse motives for contributing to the group loan (Fig. 4c and Extended Data Table 4). Consistent with the theoretical framing as a threshold social dilemma in which one's contribution may be viewed as pivotal to producing the collective good, cooperation was motivated by both self-interest and other-regarding interests. However, borrowers strongly distinguished between what is seen as paying one's own share versus paying another member's share and expressed a different range of motivations for each. Motivations reported for paying one's own share were broad and diverse, including economic self-interest and other-regarding concerns. For example, one borrower described why she pays her own share as "I want more money. I want so that they will give me more loans. I don't have anybody who can give money to me" (client interview G21.C41), whereas another client emphasized her fear of public shame: "I am always afraid of embarrassment because this is happening several times with our [neighbours]" (client interview G5.C8). Most frequently, borrowers reported other-regarding motives related to solidarity for

paying their own share, for example: "Because if I don't pay, our members of the group undergo the same constraints. So I don't want them to strain" (client interview G2.C4). By comparison, we found that motivations reported for paying another member's share were comparatively narrow, focusing almost exclusively on prosociality and obligations, as emphasized in this client's description, "We love ourselves … If any one of us has a job to do which is very difficult, we meet, we call, we mobilize ourselves and assist each other" (client interview G14.C26).

Second, distributional conflict and partial free-riding were common. A consistent theme in borrowers' reports was that distributional conflict was widespread and persistent. Even groups that achieved full on-time repayments at the group level often relied on social enforcement or internal compensation for defecting members. Systematically coding for the presence of behaviour corresponding to defection in each group, we found that full free-riding behaviour was rare in this setting (that is, an individual defecting throughout the entire loan cycle), but that partial free-riding behaviour was common, reported to have occurred by at least one member in 91.4% of interviewed groups (Extended Data Table 5). Detailed accounts indicated that repeatedly navigating the distributional conflict within groups was central to their collective outcomes. For example, "[Ramatu] failed to pay … That is the problem with traitors. They hardly give out money … And she came with this money today. And she pleads to me, says 'Please [Hawa], forgive me'" (client interview G5.C8). We saw that clients consistently distinguished between the willingness to contribute and genuine inability to pay. Often group members were able to personally assess the reasons for defection: " … we stay in the same market area. So we are able to know those whose business are going very fast and those whose business they are not moving very fast" (client interview G14.C28).

Third, behavioural changes occur over time: the interviews show that decline in group outcomes was often driven by a reduction in cooperative motivation and effort, reducing the likelihood of contributing one's

**Table 2 | Rate of decay in cooperation rates after restarts in group lending**

| | Financial contribution rate | | Cooperative effort rate | |
|---|---|---|---|---|
| | All groups | Five-cycle groups | All groups | Five-cycle groups |
| | (1) | (2) | (3) | (4) |
| Round | –1.363 | 1.685 | –2.252 | 1.401 |
| | (0.167) | (0.351) | (0.180) | (0.381) |
| | [P<0.0001] | [P<0.0001] | [P<0.0001] | [P=0.0003] |
| Cycle | –3.262 | –1.454 | –5.236 | –2.514 |
| | (0.775) | (0.934) | (0.877) | (1.209) |
| | [P<0.0001] | [P=0.1211] | [P<0.0001] | [P=0.0389] |
| Round×cycle | –0.308 | –0.772 | –0.249 | –0.946 |
| | (0.063) | (0.110) | (0.063) | (0.106) |
| | [P<0.0001] | [P<0.0001] | [P<0.0001] | [P<0.0001] |
| Number of groups | 1,589 | 197 | 1,589 | 197 |
| Number of observations | 31,199 | 8,006 | 31,199 | 8,006 |
| $R^2$ | 0.2934 | 0.2349 | 0.4117 | 0.3457 |

Coefficient estimates from ordinary least squares regressions estimating the group's financial contribution and cooperative effort rates in the round. The sample consists of all groups in the columns labelled 'All groups', and groups that received five cycles in columns labelled 'Five-cycle groups'. Regressors are round, loan cycle and round×loan cycle interactions. All regressions control for loan officer, loan size, loan duration, calendar year fixed effects, rainy season fixed effects and group fixed effects. Standard errors clustered at the group level in parentheses, and $P$ values from two-sided $t$-tests of the coefficient estimates in brackets.

own share, as well as compensating for another member. Members were described as becoming 'relaxed' (client interview G2.C3), 'stubborn' (client interview G18.C36), and 'tired of paying' (client interview G12. C21). Client G17.C34 emphasized the change in behaviour over time as "Even this second cycle, we didn't at the initial stage, we didn't have any problem … The last two months, people have become very lethargic to pay this money." Both first- and third-person accounts indicated that as time passed members became more likely to allocate their own potential monthly repayment towards self-interested uses at the cost of burdening the group, "What she did here is, she kept all this money and made use of it in a different situation" (client interview G19.C38; G4.C6; G7.C11; G9.C17). The decline in motivation to contribute to the group exhibited reduced regard for others as well as reduced regard for long-term self-interest. For example, some interviews revealed emotional responses to weakening social relationships: "I feel disappointed about it" (client interview G27.C54), "[previously] a sisterly friend" (client interview G30.C57). Others illustrated members' deteriorating regard for long-term interests: "They rather prefer to buy dress, cloth dress, so that just to decorate their body. They live in world of fashion" (client interview G25.C50). Describing the shift in behaviour, "I think they are not responsible. They show irresponsibility in their mind" (client interview G32.C59). Furthermore, the evidence attests that clients were attuned to the behavioural choices of their group members and suggests that awareness of a shift in behaviour may have further undermined the cooperative motivation of the remaining contributors, compatible with a psychology of conditional cooperation[3].

The qualitative data also convey borrowers' attempts at preventing decline in cooperative motivation and effort within their groups. Reports underscored the role of repetition in counteracting decay, with some borrowers reiterating the long-term financial consequences, for example, "I continue to say we must work very hard to pay this money, otherwise they will not give us another loan" (client interview G35.C64), and others drawing on more emotional and prosocial sentiments, for example, "We have sympathy to each other. We normally encourage

each other … " (client interview G17.C35). Similarly, the lending institution used loan restarts as an opportunity to potentially reanimate motivation, reminding borrowers of financial consequences and their social commitments. As one staff member described, "They begin to drag their feet … So we [resensitize] them … and that will, like, give them a rethink that, 'Oh, I should stop this … ' So they will catch up again" (staff interview 7).

Fourth, heterogeneity in patterns of defection: the qualitative data suggest a common tendency for cooperative motivation and effort to decay gradually over the months of repeated payments, yet also include considerable individual heterogeneity—from participants who maintained full cooperation for years to those who rapidly shifted to defection and never contributed again. Extended Data Table 6 shows the underlying heterogeneity in the severity of individual defection behaviour and the frequency of different cooperative patterns in each group. The most common pattern is intermittent defection—which is reported in 62.9% of interviewed groups—where members defected in at least one round, but later contributed in a subsequent round of the loan cycle. In these groups we observe an increasing likelihood of individuals burdening the group over time, but with the defectors still attempting to maintain access to the collective good. More abrupt shifts in cooperation are also present: 17.1% of groups featured walk-away defections, in which members defected for more than one round without returning to contribute, whereas the other members continued to contribute, primarily out of regard for the remaining members or a sense of obligation (for example, G3.C5; G11.C20; G23.C46; G28.C55). In some cases, we found walk-away defections prompting a cascade of defections or producing full collapse, "with each one [defecting] after the other one" (client interview G25.C50), potentially exacerbated by additional mechanisms for non-cooperation, such as social contagion or financial reassessment. We found 8.6% of groups had a cascade of escalating defections; and 2.9% of groups exhibited full collapse, in which all members ultimately defected. Refer to the Supplementary Information for full coding criteria.

## Alternative explanations

Our leading explanation is that the long-term pattern of punctuated decline documented here is driven by the behavioural mechanisms of cooperative motivation and effort decay. We now summarize the evidence in relation to the three most relevant alternative explanations: structural or social learning, strategic decline and systematic change in financial ability. Note that our argument is not that such alternative mechanisms do not have a role in other cooperative contexts. Instead, the evidence suggests that these alternatives are not the primary drivers of behaviour in our setting.

Learning explanations typically refer to either learning the rules and payoff structure[9,28,30,31], or the cooperativeness of other players in one's group[8,29]. Initial cooperation results from confusion or uncertainty and is followed by decline as participants learn the consequences, update their beliefs or reduce their errors. This could also be extended in the group-lending context to include learning the enforcement consequences of defection. However, the substantial rebounds in cooperation following restarts, and the risk of cooperative failure years after a group receives a first loan, when loan structure and group membership are predominantly stable over loan cycles (Figs. 1–3, Table 1, Extended Data Figs. 1–3, Extended Data Table 2 and Supplementary Tables 2–4), argue against cumulative mechanisms, which should generally progress as the group continues.

Explanations of strategic decline are based on forward-looking individuals who intend to maximize their private return over an expected horizon. Relevant examples include backward induction or endgame effects[5,7] and strategic enticement, that is, the use of early cooperation as a calculated attempt to induce norms of reciprocity despite plans to defect later and maximize overall individual payoff[6,28].

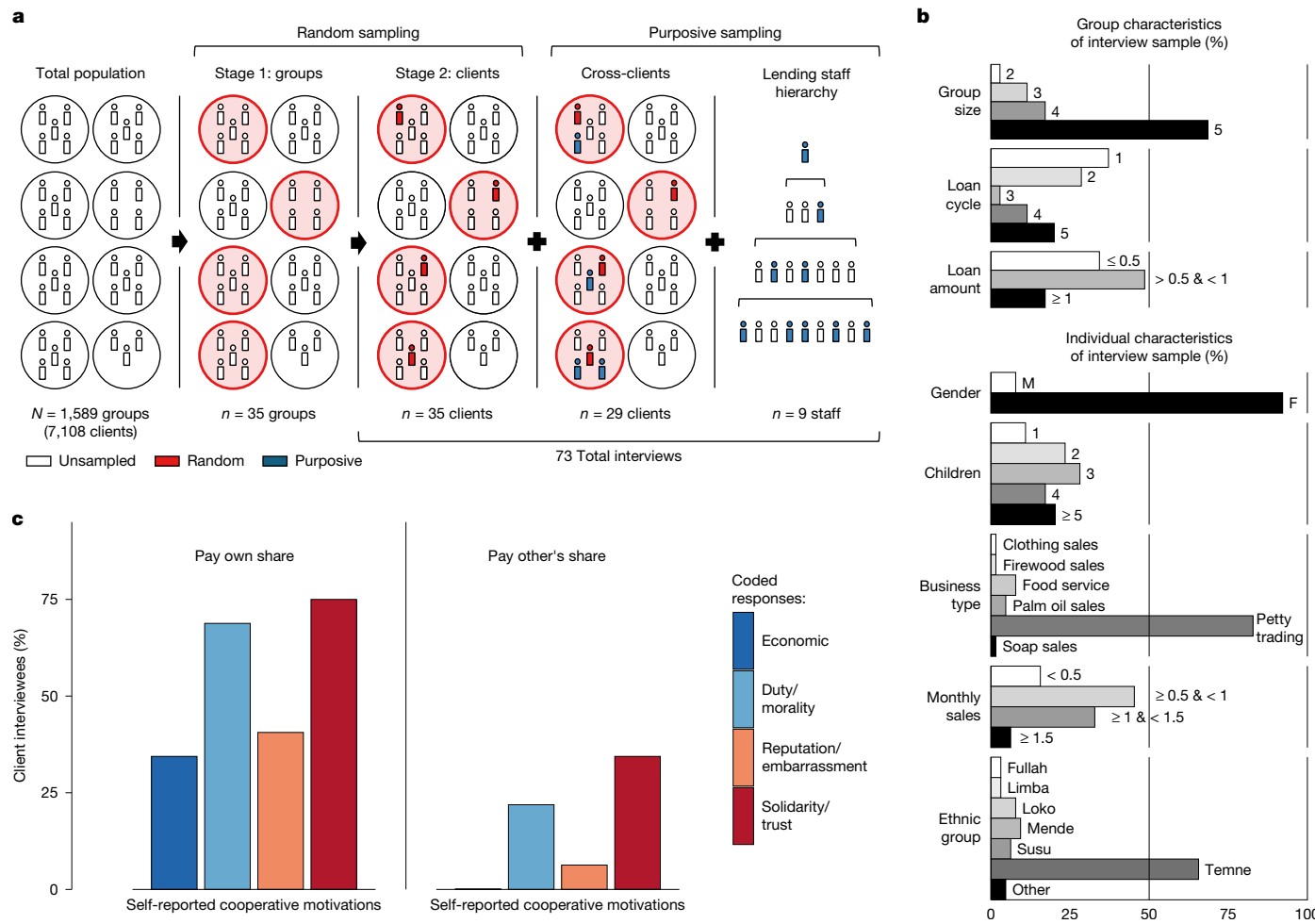

**Fig. 4 | Interview data on cooperative dynamics in group lending.** Semi-structured interviews (*n* = 73; totalling 56 h) were conducted with group lending clients and staff to understand the cooperative process in their own words. **a**, Sampling methodology schematic: two-stage cluster sampling with a purposive enhancement. Random sampling from the target population (Methods) was used to produce a representative sample of clients. A purposive sample of clients from the randomly selected groups and staff from multiple levels of the lending institution hierarchy was added to cross-validate and enhance the internal validity of the findings. **b**, Group and individual characteristics of the client interview sample. Loan amounts and monthly sales are per group member in millions of Sierra Leonean leones (SLL). **c**, Clients

report heterogenous motivations for cooperating in group lending and strongly distinguish paying one's own share from compensating for another member. The interview data indicate that members often attempt to partially free-ride, thereby heightening group reliance on a narrower set of motivations to reach the collective threshold. Thematic analysis indicates that cooperative decline is often driven by reduced motivation over time to pay one's own contribution and to compensate for another member. Refer to the Methods for details on the analysis process, Extended Data Tables 4–6 for coded cooperative behaviour, and the Supplementary Information for the interview protocol, codified themes and extended direct quotes.

Here, several pieces of evidence suggest that these are not the main drivers of our results. First, the lending institution has no set limit on the number of loan cycles. At the time of data collection, the maximum number of cycles completed by any group was eight—notably more than when most groups experience a collapse in cooperation. Second, the structure of group lending as a threshold dilemma does not favour strategic decline within a loan cycle. Rather, the incentives are to defect immediately in full after receiving one's share of the loan or to avoid delayed defection that causes the group to not reach provision, avoiding one's contributions going to waste[44]. Empirically, delayed decline in cooperation often results in loss of access to future credit and inefficient outcomes (Fig. 1, Extended Data Figs. 1 and 2, Extended Data Table 5 and Supplementary Table 1).

The final class of alternative explanations, specific to group lending, is based on systematic change in the borrowers' financial ability to pay. Within loan cycles, the decline could be driven by a progressive reduction in available cash or the gradual accumulation of exogenous shocks to the members' businesses[45,46]. Across loan cycles, it is plausible that as loan amounts increase, they become more difficult to

repay[45,47]. Although clients are often faced with economic constraints, four different types of evidence indicate that they are not the cause of the longitudinal patterns. First, organizational policies specify that a group loan is to be rescheduled by the microfinance institution if the loan officers judge the group to have genuinely encountered an inability to repay. Furthermore, the microfinance institution requires that groups keep a portion of their original loan in a savings account to use in case of inability to pay. This is rarely accessed by defaulting groups, indicating unwillingness rather than inability to pay. Second, client interviews demonstrate that groups are often able to overcome genuine financial difficulty, for example, a house fire (client interview G22.C43), unexpected medical bills (client interview G9.C16), slow business (client interviews G5.C8, G1.C1) or theft (client interview G7.C11), but consistently struggle to repay when a member's behaviour becomes uncooperative (for example, client interviews G12.C22; G27.C54; G35.C64). Third, our statistical analysis shows that decline within a loan cycle is not monotonic. Cooperation often increases in the final rounds of a loan cycle (Supplementary Table 8), indicating behavioural motivations rather than a strict decline in financial

ability over rounds. Furthermore, statistical models include group-level controls for variation in ability to recover from economic shocks as group fixed-effects or as covariates (for example, average monthly sales, business diversity and geographic region), temporal controls for seasonal weather shocks and other idiosyncratic adverse events by year, and cycle controls for progressive changes in borrowers' financial situation in terms of loan amount and duration (Tables 1 and 2, Extended Data Table 2, Supplementary Tables 2–5 and 9). Fourth, the construction of the cooperative effort rate as an alternative outcome variable to financial contribution, applied to the entire quantitative dataset, minimizes the role of financial liquidity by considering timeliness of any partial payment, no matter how small. The longitudinal patterns and statistical tests largely replicate across outcome measures (Figs. 1–3, Tables 1 and 2, Extended Data Figs. 1 and 2, Extended Data Table 2, and Supplementary Tables 2–4 and 8–9).

## Discussion

By examining a long-term repeated social dilemma with meaningful stakes in the field, and by eliciting first-person accounts of motivations through interviews, this study departs from a broad interdisciplinary literature on human cooperation that is frequently based on short-term laboratory experiments. Despite great differences in stakes and timescale, we observe a pattern of punctuated decline in cooperation that aligns with findings from decades of experimental research[6,7,28,36,40,48,49]. As far as we are aware, this is the first study to analyse the cooperative restart effect in a natural social dilemma.

Although there is strong empirical evidence for a decline in cooperative behaviour over time, there is an active debate about understanding this phenomenon from a theoretical perspective with regard to different potential mechanisms. This study proposes that dynamic behavioural deviations can be a fundamental driver of punctuated decline. An advantage of the group lending context is that it excludes standard explanations predicated on increasingly rational or strategic behaviour as the key sources of decline[5,7,9,28,30,31]. Our findings align with growing evidence that a majority of humans are conditional cooperators who are susceptible to behavioural biases[3,26,29,36,50]. However, we suggest that predictable changes in psychological states may have a larger role than currently recognized and should be considered as key drivers of cooperative decline. Rather than a stable disposition towards cooperation or defection, we find that cooperative motivation and effort naturally decay, intensifying the tendency to undercontribute over time and heightening the risk of group collapse. This observation is consistent with similar psychological processes known to produce internally inconsistent choices over time, such as a decline in goal persistence due to changes in motivational states[51,52] or habituation to salient behavioural cues[34,53,54]. Future research would benefit from elucidating the underlying psychological processes in more controlled settings and from exploring the role of behavioural decay and its interaction with other cooperative mechanisms across diverse empirical contexts.

The dominant approach to modelling social dilemmas to date has been as a prisoner's dilemma, or its generalization, the linear public goods game[15,55]. However, it has been argued that many cooperative dilemmas in practice involve a common project whose provision is 'all or nothing' and are better conceptualized as a threshold public good game[43,56–58]. In such situations, it is often clear what is each group member's fair-share contribution. Our finding that group members report distinct motivations for contributing towards their own share versus contributing towards someone else's share emphasizes the value of using threshold games to model cooperation. If some motivations, such as those more prosocial in nature, are more prone to natural decay than others, such as those more self-interested in nature, this may be a crucial feature to help to model and explain the decline in cooperation over the long term.

Incorporating behavioural dynamics into models of collective action may be essential for explaining why cooperation falters even under structurally favourable conditions. For institutions reliant on long-term cooperation, the findings suggest a class of solutions aimed at mitigating behavioural decay, such as contribution automation, strategic resets, intrinsic motivation enhancement or habit formation[34,59–61]. We hope our study spurs further research on the role of behavioural decay in cooperation and leads to the development of solutions for better combatting or circumventing this human tendency.

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

# Methods

## Ethical approval

This research was approved by the Central University Research Ethics Committee (CUREC) at the University of Oxford (reference no. SSD/CUREC1A/10-099). The approval included the collection process and analysis of administrative microfinance data and primary data from human participants collected through semi-structured interviews. The research was performed in accordance with all relevant guidelines and regulations outlined by CUREC. Informed consent was obtained from all participants. The microfinance institution and its clients are de-identified as per the data use agreement.

## Empirical background

The microfinance institution was founded in 2002 and had a client base of approximately 18,000 borrowers at the time of data collection. In Sierra Leone, 68% of the population was estimated to be living below the national poverty line[62]. The organization offers small loans to low-income clients with the goal of local poverty alleviation. Group lending is a standard arrangement in developing countries, where potential clients seeking a loan enter into a joint liability contract[63]. An organization's motivation for this arrangement is that potential clients often lack sufficient financial collateral, but a group contract may provide a form of social collateral[64]. Past research has noted that this lending structure creates a natural social dilemma[38,65,66]. More specifically, the group lending structure can be conceptualized as a threshold social dilemma, as the collective good—access to future credit—is provided only when full repayment is reached[2,11,12].

Potential group lending clients are instructed to select group members that they know and trust[67]. Furthermore, loan officers ensure that each member meets basic eligibility criteria. Each client is required to have their own micro-business capable of meeting the minimum loan repayments.

Typical micro-business examples include petty trading, food services, barbershops, tailoring and motorbike taxi services. The loan amount ranges from 200,000 to 2,250,000 SLL per member per loan (at the time of data collection, the nominal exchange rate was US $1 = 4,300 SLL). All loans have the same fixed interest rate: 2% per month. Scheduled loan duration ranges from six to twelve months. Group size in the dataset ranges from two to six members, with an average of 4.36 members. Consistent with the organization's objective to promote financial inclusion, 73.3% of the clients are female.

Eligibility for a subsequent loan cycle depends on a group's repayment performance in the previous cycle. Full repayment is required, but timeliness of payments throughout the whole loan cycle also influences the decision. The institution incentivizes better group repayment with greater potential increases in the subsequent loan amounts. Groups that repay in full and on time receive the standard maximum loan increase for the next cycle. If the loan is paid in full, but with delayed payment(s), the group may be assigned a lesser loan increase, no loan increase or no loan renewal, on the basis of the frequency and severity of the delayed payment(s). The outcome is the same for all group members. Decisions regarding loan renewal and amount are made jointly by the group's loan officer and loan portfolio manager. Variation in subsequent loan amounts implies that a single provision point for the collective good is a theoretical simplification of the full set of potential group outcomes. However, the core distributional conflict and individual incentive to contribute as little of one's own resources as possible while reaching each provision point remains consistent.

Group loan enforcement operates in two phases: (1) internal enforcement of active loans and (2) institutional enforcement of delinquent loans. Group members are responsible for enforcing on each other during the active phase. Members use a variety of positive and negative enforcement mechanisms, such as encouragement, social pressure, embarrassment and ostracism. Past empirical research in this context has shown that natural variation in the social and spatial structure of joint-liability groups has a critical role in the ability and willingness to apply such social sanctions[38]. During this phase, the collective good is still achievable; that is, access to future credit. After approximately 30 days of delayed payment, enforcement gradually shifts to the institutional phase. In this phase, the possibility of qualifying for a subsequent loan rapidly declines. The loan is classified as inactive, and the institution initiates formal recovery procedures—typically involving debt collectors and legal threats. These efforts generally continue for at least a year until the group repays or the organization officially records the loan as a write-off and ceases efforts at collection. In the 'Modelling' section below, we define cooperative outcome measures using a 30-day window to capture meaningful behaviour attributable to internal group dynamics, distinct from institutional debt recovery efforts.

## Quantitative materials and methods

**Quantitative data collection.** The data were collected by the microfinance organization between 2005 and 2011. During the initial group approval process, the organization records personal, demographic and financial information for each member of the group. After loan disbursement, the organization records group-level repayment in terms of date and amount. The repayments are self-organized by the group but must be registered at the local branch office. It is not required that all members be present when making a group payment. The microfinance organization provided consent for the analysis and publication of the administrative data in accordance with the data-use agreement, indicating that the research team may disclose and/or publish data such that "the data shall be anonymized and/or aggregated so that no reference is made to individuals' names, applying to both [microfinance institution] clients and staff."

For each scheduled repayment the organization records whether the payment was made in a single payment or in partial payments on different dates. For example, consider a group that has a scheduled monthly payment of 300,000 SLL due on 6 June 2010. If the group is unable to repay by the due date, the group might pay 100,000 SLL on 8 June 2010 and the remaining balance of 200,000 SLL on 20 June 2010.

Once a loan is disbursed, group membership is fixed throughout the duration of the loan cycle. When proceeding to a new loan cycle, there may be minor changes in group membership. The most common change is the removal of a defecting member. Across all loan cycles, 81% of groups maintained the same group size, 18% reduced the group by one or more members, and fewer than 1% of groups increased group membership across cycles. Note that the potential exclusion of a group member from the next loan cycle is an internal group decision. The lending institution only approves a subsequent loan cycle at the group level, that is, it does not differentiate how much each individual contributed to repayment (refer to Extended Data Table 7 and Supplementary Table 10 for a more detailed breakdown of group size over loan cycles). The average days elapsed since repayment of the last round in the previous cycle to disbursement of the next loan is 24.7 days.

The statistical analysis and figure generation were conducted using STATA v.18.5, including package Markstat v.2.1, and the software R v.4.2.3. The dataset was cleaned so that loan cycles are not truncated midway by the data window, that is, both the initial disbursement and final scheduled payment fall within September 2005 to January 2011. We include a further four-month data collection buffer (until May 2011) to allow for delayed repayments that were originally scheduled during the data window. The maximum number of cycles completed by any group at this organization was eight cycles at the time of data collection. However, the sample size decreases with each cycle, with few groups having taken more than five cycles (53 out of 1,589 groups). For this study, the dataset is limited to the first five loan cycles with a sample size useful for statistical analysis. The resulting cleaned dataset includes 7,108 borrowers involved in 31,199 scheduled group repayments. Including partial payments, the dataset consists of 47,931 repayment transactions.

Extended Data Table 1 provides a summary of descriptive statistics of the quantitative dataset.

**Modelling approach.** A group's cooperative behaviour is modelled using two measures: the financial contribution rate (FCR) and the cooperative effort rate (CER). We model the FCR as the percentage of the monthly scheduled amount paid by the group within 30 days:

$$y_{FCR} = \frac{o_{icr}}{s_{icr}} \tag{1}$$

where $o_{icr}$ is the observed payment amount of group $i$ in round $r$ of cycle $c$, and $s_{icr}$ is the scheduled payment amount for the associated group, round and cycle. Repayment activity may continue to be recorded by the lending institution beyond 30 days. We apply the 30-day boundary to construct a regular time interval that captures variation in a group's internal cooperative behaviour rather than institutional efforts at debt recovery.

Most measures of economic cooperation in a field setting will reflect some degree of an individual's financial ability to contribute; however, it is important in this context to confirm that changes in repayment are not driven by changes in clients' financial ability to repay, that is, not a systematic reduction in financial liquidity over rounds. We constructed the CER to further increase our confidence in measuring cooperative behaviour. If a group is willing to cooperate but experiencing a financial difficulty, it may make a smaller partial payment. The measure is based on the number of days overdue before the group makes its first partial or full payment for each month. The organization accepts any minimal partial payment. The data show that partial payments as small as 5% of the monthly amount due were recorded by the organization. The timeliness of the first partial payment reflects a group's effort in signalling their repayment intentions, rather than their ability to pay the full amount. The CER is calculated as:

$$y_{CER} = 1 - \frac{d_{icr}}{30} \tag{2}$$

where $d_{icr}$ is the days overdue for the first partial payment bounded from 0 to 30 for group $i$ in round $r$ of cycle $c$. The CER can be interpreted as follows: if a group makes at least some partial payment by the scheduled due date, the group effort rate equals 100; if the group's first payment is 15 days overdue, the group effort rate equals 50; if no payment is made within 30 days, partial or full, the group effort rate equals 0. In combination, the FCR (based on amount paid) and the CER (based on timeliness) offer greater insight to a group's cooperative behaviour.

Loan durations in the dataset are 6, 8, 10 or 12 months. For plotting purposes, the rounds of loans with different durations were rescaled to the minimum duration. For example, for a 12-month loan, the first and second payments are grouped as round 1; the third and fourth payments are grouped as round 2; and so on. A consistent base makes it visually easier to compare cooperation rates for the aggregated dataset. Importantly, we do not use rescaled rounds in the regression models when statistically testing longitudinal trends.

**Econometric models.** To model longitudinal cooperative behaviour, we estimate group fixed-effects regressions of the form:

$$y_{irc} = \beta_0 + \beta_1 c + \beta_2 r + \beta_3 c \times r + \alpha_i + X'_{irc}\gamma + \mu_{irc} \tag{3}$$

where $y_{irc}$ is the cooperation rate of group $i$ in round $r$ of cycle $c$, $\alpha_i$ is a group fixed effect, $X'_{irc}$ is a set of controls and $\mu_{irc}$ is the error term. We also run regressions that are identical to equation (3) except that they treat cycle as a fixed effect rather than a continuous variable.

To estimate the size of the restart effect, we estimate group fixed-effects regressions that compare round-to-round changes in cooperation rates across all rounds of a cycle to the change in cooperation rates between the last round of a cycle and the first round of the subsequent cycle, including observations only from groups that participate in both cycles:

$$\Delta y_{irt} = \beta_0 + \beta_1 n_{irt} + \lambda_t + \alpha_i + X'_{irt}\gamma + \mu_{irt} \tag{4}$$

where $\Delta y_{irt}$ is the change in the cooperation rate from round $r - 1$ to round $r$ for group $i$ in restart number $t$; $n$ is an indicator that the observation corresponds to a restart change (that is, a change from the last round of a cycle to the first round of the subsequent cycle); $\lambda_t$ is a restart number fixed effect; $\alpha_i$ is a group fixed effect; $X'_{irt}$ is a set of controls; and $\mu_{irt}$ is the error term. Observations corresponding to restart number 1, for example, comprise all round-to-round changes in cycle 1 plus the change from the last round of cycle 1 to the first round of cycle 2.

We also fit random-effects panel regression models to examine the group factors (for example, group size, proportion female, monthly sales, business type diversity) and loan factors (for example, loan amount, loan officer, seasonal variation) correlated with cooperation rates (Supplementary Table 9). Columns 'average cycle 1' estimate a simple ordinary least squares regression:

$$y_i = \beta_0 + X'_i\beta + \mu_i \tag{5}$$

where $y_i$ is group $i$'s average contribution or effort rate in cycle 1 and $X'_i$ is a vector of group and loan characteristics of group $i$ in cycle 1 (specifically, group size, female proportion, proportion married, average number of children, average monthly sales, the s.d. of the monthly sales, average business equity size, s.d. of business equity, index of business diversity within the group, proportion of group members engaged in petty business, an indicator that at least a member of the group is engaged in service business type, loan size, rainy season and microfinance institution branch controls). For observations with missing covariate data, values were imputed using mean substitution. The 'RE all rounds' columns in Supplementary Table 9 estimate the random-effects panel equation:

$$y_{irc} = \beta_0 + \beta_1 r + \delta_c + \beta_2 r \times \delta_c + \alpha_i + X'_{irc}\gamma + \mu_{irc} \tag{6}$$

where $y_{irc}$ is the contribution or effort rate of group $i$ in round $r$ of cycle $c$; $\delta_c$ is a cycle fixed effect; $\alpha_i$ is a group random effect; and all of the other terms are as defined above.

Consistently for the different specifications, greater group size is significantly associated with lower contribution and effort rates, whereas the proportion of female individuals in the group is significantly associated with higher effort rates. Perhaps counterintuitively, larger average monthly sales are significantly associated with lower contribution and effort rates. We do not interpret these correlations as causal, as it is possible that the microfinance institution uses these variables when deciding whether and what kind of loan to give to the group.

## Qualitative materials and methods

**Qualitative data overview.** We conducted 73 in-depth semi-structured interviews: 64 interviews with group lending clients and 9 interviews with members of the lending institution staff. On average, client interviews lasted 39 min and staff interviews lasted 1 h and 34 min. Interview time totalled approximately 56 h. The interviews were conducted by the principal investigator between 5 April 2011 and 6 June 2011, contemporaneous to the collection of the quantitative data.

The client interviews provide detailed descriptions of the cooperative dynamics in the borrowers' own words. The staff interviews are valuable for understanding the organization policies and their practical implementation. Collectively, the interview data perform two key functions in the study: (1) they help to contextualize the cooperative dilemma, to ensure proper understanding of the quantitative patterns and their appropriate interpretation; (2) they provide insight to the behavioural mechanisms underlying the longitudinal trends.

The qualitative data may be viewed as serving both a 'confirmatory' function, that is, ensuring that the results are not dependent on a single type of data, as well as a 'complementary' function, filling the interpretative gaps inherent to the quantitative dataset[68].

**Sample selection.** The sample of clients for interviews was drawn from the overall quantitative dataset, using a two-stage cluster random sampling, plus a purposive enhancement[41]. In the first stage of the random sampling, we used simple randomization of groups, after restricting the population of potential groups based on two criteria: (1) we geographically restricted the pool to groups that were administered at the lending institution's principal branch. This was implemented for practical efficiency of interview logistics; (2) we restricted the pool to groups that had been engaged in borrowing within the last six months. This was implemented to reduce recall bias during the interviews. This resulted in 35 joint liability groups drawn by simple randomization from the subpopulation. Supplementary Table 6 provides descriptive statistics of the client interview sample at the group level.

In the second stage of the random sampling, we selected one member per group to be interviewed using simple randomization within the group. We then enhanced this sampling design by implementing a purposive sampling of an additional member from within the randomly selected groups. The choice of whether to conduct another interview and with which specific member was on the basis of the content provided in the first member's interview, following the researcher's discretion regarding which additional group member's perspective would provide the most valuable information. For example, if the first interviewee indicated that a specific member X had been the main source of cooperative disruption in the group, member X was selected for a direct interview to hear his or her version of the events. The intent of additional within-group interviews was to cross-validate the initial interview, collect potentially contradictory data, and understand a complex phenomenon from different points of view[69]. This resulted in 29 further interviews, producing a total of 64 client interviews. Extended Data Table 3 provides descriptive statistics of the client interview sample at the individual level. The two-stage cluster random sampling with the additional purposive sampling used here promotes both external validity, in the first and second stage through the random selection of a representative sample, and internal validity, through the triangulation of multiple interviewee perspectives.

Interviews were also conducted with a non-random sample of nine staff members of the lending institution—including three loan officers, two information and accounting officers, two loan portfolio managers, and two executive directors—regarding organization policies, practices in the field, and the organization's record keeping process. Supplementary Table 7 provides descriptive statistics of the staff interview sample. The interview content was instrumental in interpreting the quantitative dataset, providing context, and offering cross-validation of borrowers' descriptions of the group dynamics from an external perspective. Furthermore, to assess the generalizability of the cooperative behaviour at this lending institution as compared to other organizations in Sierra Leone, interviews were conducted with staff at three independent lending institutions in Sierra Leone offering group loans. The interviews indicated that the organizational practices and group dynamics at the primary institution were highly similar to those at the other lending institutions.

**Qualitative data collection.** Interviews were conducted in-person in Sierra Leone by the principal investigator. The interviews were conducted at a location preferred by each client or staff member to promote their ability to speak candidly (for example, in their home or place of work). Before each interview was conducted, the research purpose and use of the interview data was explained to the client or staff member and a physical document of informed consent was reviewed together. All participants provided informed consent by signature or thumbprint. To safeguard participant confidentiality, personally identifiable information was redacted from the interview data.

All client interviews were conducted with a professional translator from Sierra Leone present. The translator was independent from the lending institution and presented no conflict of interest. Choice of interview language was at the preference of the client. The languages used by the clients during the interviews were Krio, Temne, and English. Extended Data Table 3 includes the language predominately spoken at each interview. All staff interviews were conducted in English. During all the interviews, field notes were taken by the principal investigator and audio was recorded for later transcription and analysis.

Supplementary File 1 provides the semi-structured interview protocol for clients, an outline that provides a topic structure, a list of prompts and example questions to direct the conversation. The value of the semi-structured interview format in this context is the ability to adapt flexibly to the interviewee responses, to pursue interesting topics in greater depth, and uncover unexpected or disconfirming evidence[70]. The interview guide consisted of the following primary elements: (1) introductions and informed consent; (2) background/warmup; (3) group lending verification; (4) social connections within group; (5) client perception of the dilemma structure and responsibilities; (6) repayment process; (7) cooperative motivation and incentives; (8) free-riding and/or cooperative behaviour; (9) loan enforcement; (10) behavioural, financial and structural changes over time; and (11) closing.

Accounting for positionality and mitigating biases were central to the design and conduct of the interviews[70]. In this context, the perceived dynamic between group lending clients and an external researcher shapes the nature of the conversation and requires the practice of reflexivity to promote impartial data. For example, to avoid misinterpretation of his role, the interviewer would clarify at the outset of each interview that he was not acting on behalf of the lending institution and that the discussion would have no effect on the client's standing with the lending institution. To help to build participant comfort, the protocol was structured to begin with relatively easy questions before progressing to more challenging questions regarding group behaviour. If client responses appeared to be culturally embedded, the interviewer would further probe the topic to ensure that the answers would be appropriately interpreted.

To reduce recall bias, interview questions focused on the current or most recent loan cycle. The protocol included probes designed to elicit concrete details and examples from the clients. For example, printed copies of client and group information, loan characteristics, and repayment data as recorded by the lending institution were brought to each interview by the researcher. The researcher familiarized himself with this information before each interview. The information could then be used during the interviews to reduce social desirability bias if a client was misrepresenting their group's behaviour as overly positive. The information also served the benefit of verifying the accuracy of the formal repayment data and probing for any systematic inconsistencies in the repayment recording process. To further reduce social desirability bias, the sampling strategy incorporated additional member(s) of the same lending group, describing the same event but from different perspectives. We found the third-person accounts valuable as they were more open to describing socially undesirable behaviour.

**Qualitative data coding and analysis.** Audio recordings of the interviews were manually transcribed verbatim by the research team. The data were managed and coded using NVivo software (v.14.24.1). Pseudonyms are used in the direct quotes to protect client and staff confidentiality. Anonymized identifiers for joint-liability members were created at the group and client levels. The identifiers (for example, G34.C63) consist of a group number (G1–G35) and a unique client number (C1–C64). Anonymized identifiers for staff are designated as S1–S9. We randomly assigned the staff identifiers across roles to avoid compromising confidentiality.

We analysed the transcribed interviews to identify themes and concepts concerning clients' perceptions, motivations, and direct

experiences of cooperation and defection in the joint liability groups[71]. Codes were primarily derived from existing literature, but also included 'open codes,' adaptable to unexpected themes and disconfirming evidence[72]. Codes were organized into preliminary lower- and higher-order themes. All authors reviewed the codes and themes to assess their relevance. Revisions were made through multiple group discussions, following an iterative and reflexive process. The analytic method was abductive, recursively iterating between data and existing theory, to develop themes that coherently reflect the patterns present in the data[70,71].

Conceptually, when we refer to cooperative motivation and cooperative effort, we draw on a common distinction between motivation (as an internal process or psychological force that initiates and sustains goal-directed behaviour) and effort (as the observable exertion or intensification of mental or physical activity towards that goal)[51,73–76]. The conceptual distinction is informative, as a group member may have the motivation to cooperate (for example, strong fear of social embarrassment), but may be externally constrained and not able to exhibit observable effort (for example, she is sick and unable to contribute). In this study, we solicit cooperative motivations directly from clients in interviews and analyse their observable behaviour in the lending group as a signal of cooperative effort. We are able to analyse cooperative effort both with the interview data, using first-hand accounts of group member behaviour, as well as with the quantified measure of cooperative effort applied across the entire dataset over time.

Codified themes and direct quotes are available in the Supplementary Information, organized according to the following higher-order themes: (1) cooperative dilemma structure and client understanding in the field; (2) extent and evidence of free-riding; (3) self-reported motivations for cooperation; and (4) changes in behaviour over time.

### Reporting summary

Further information on research design is available in the Nature Portfolio Reporting Summary linked to this article.

## Data availability

The deidentified group lending data is publicly available at the Open Science Framework (https://doi.org/10.17605/OSF.IO/26BFC)[77]. Source data are provided with this paper.

## Code availability

Code to reproduce the statistical analysis and figure source data are publicly available at the Open Science Framework (https://doi.org/10.17605/OSF.IO/26BFC)[77].

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

**Acknowledgements** We would like to thank D. Baldassarri, D. Barron, N. Christakis, T. Powell, T. Snijders and P. Tufano for extremely valuable feedback on previous drafts of this article, and the lending institution staff for sharing their local expertise and for their support in the data collection process. We acknowledge funding from the Departamento de Administración, Facultad de Administración y Economía, Universidad de Santiago de Chile (to N.S.); ANID FONDECYT de Iniciación en Investigación 2020 folio (grant no. 11200781 to N.S.); the European Commission, through the 7th Framework FET-Open Project FOC-II (grant no. 255987 to F.R.T.); the Oxford Martin School (grant no. LC1213-006 to F.R.T.); Saïd Business School, University of Oxford (N.S. and F.R.T.); CABDyN Complexity Centre, Oxford University (N.S. and F.R.T.); Oxford University Centre for Corporate Reputation (N.S.); the Skoll Centre for Social Entrepreneurship, Oxford University (N.S.); and the Department of Economics, Stanford University (D.K.).

**Author contributions** Authorship follows the policy guidelines for inclusion and ethics in global research. N.S. performed investigations. N.S., D.K. and F.R.T. conceptualized the work, designed the methodology, performed the formal analysis, visualized the work, acquired funding, administered the project, and wrote, edited and reviewed the original draft. F.R.T. supervised the work.

**Competing interests** The authors declare no competing interests.

**Additional information**
**Correspondence and requests for materials** should be addressed to Nicholas Sabin, David Klinowski or Felix Reed-Tsochas.

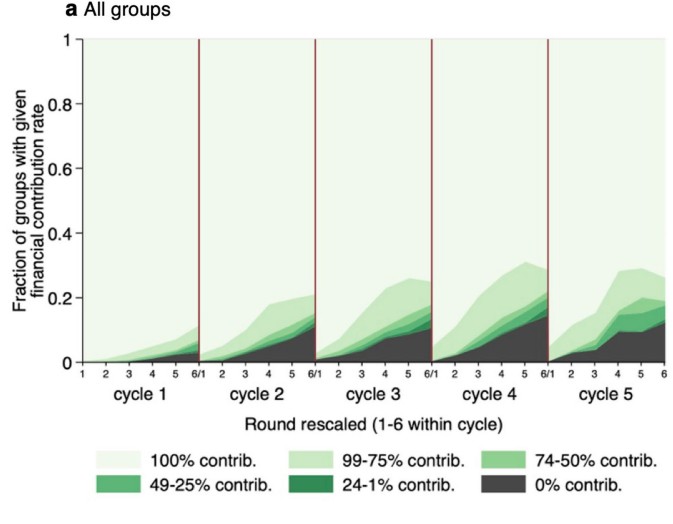

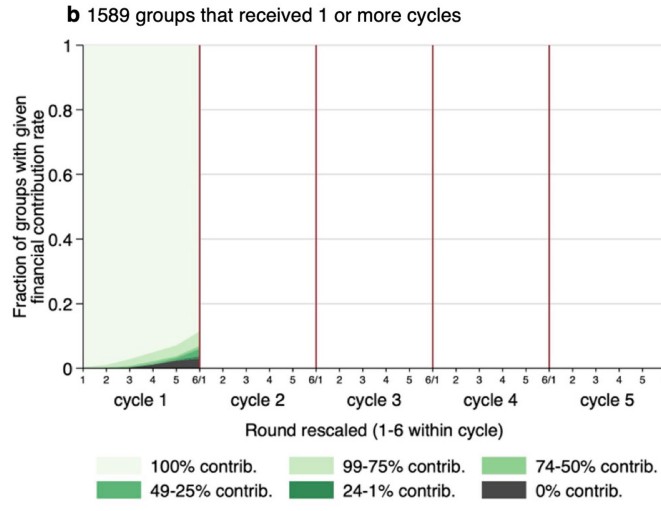

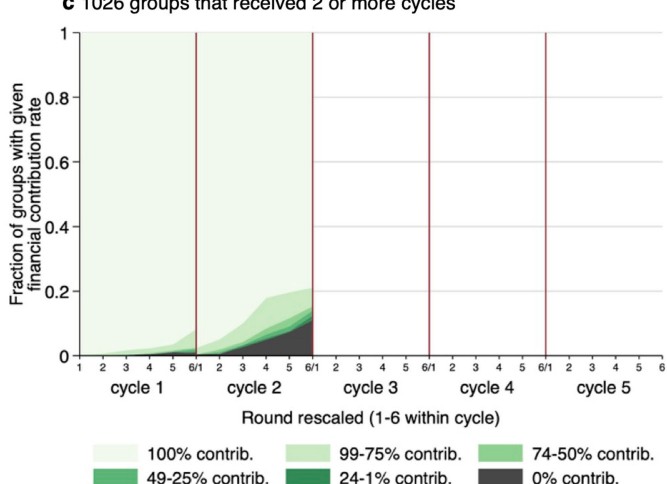

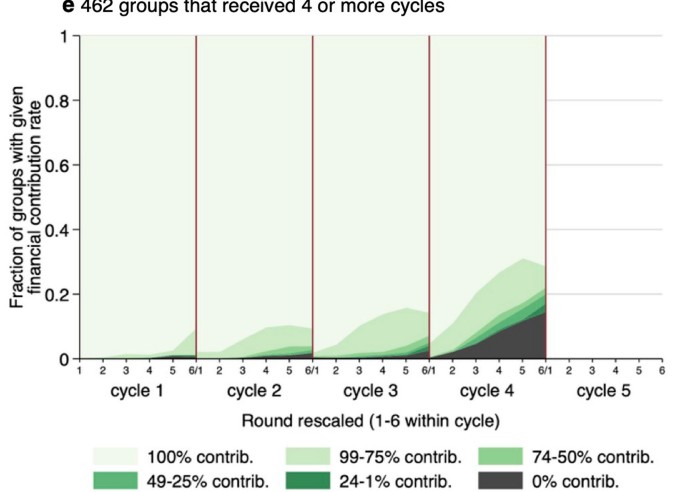

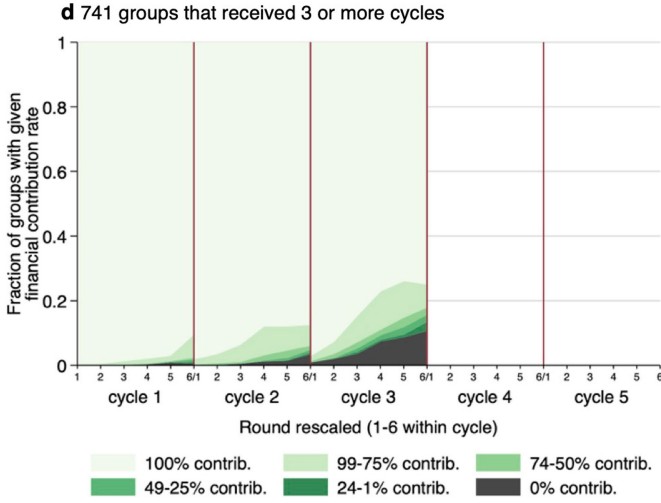

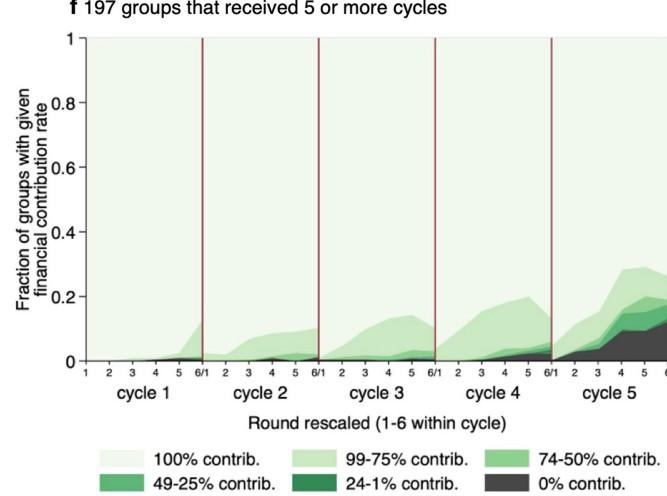

**Extended Data Fig. 1 | Longitudinal distribution of groups by financial contribution rate.** Each panel plots the fraction of groups whose financial contribution rate in a given round is (i) 0 percent, (ii) greater than 0 percent and less than 25 percent, (iii) greater than or equal to 25 percent and less than 50 percent, (iv) greater than or equal to 50 percent and less than 75 percent, (v) greater than or equal to 75 percent and less than 100 percent, and (vi) 100 percent. Vertical red lines indicate the end of a cycle and start of the new cycle. In panel **a**, the number of groups is 1589 in cycle 1; 1026 in cycle 2; 741 in cycle 3; 462 in cycle 4; and 197 in cycle 5.

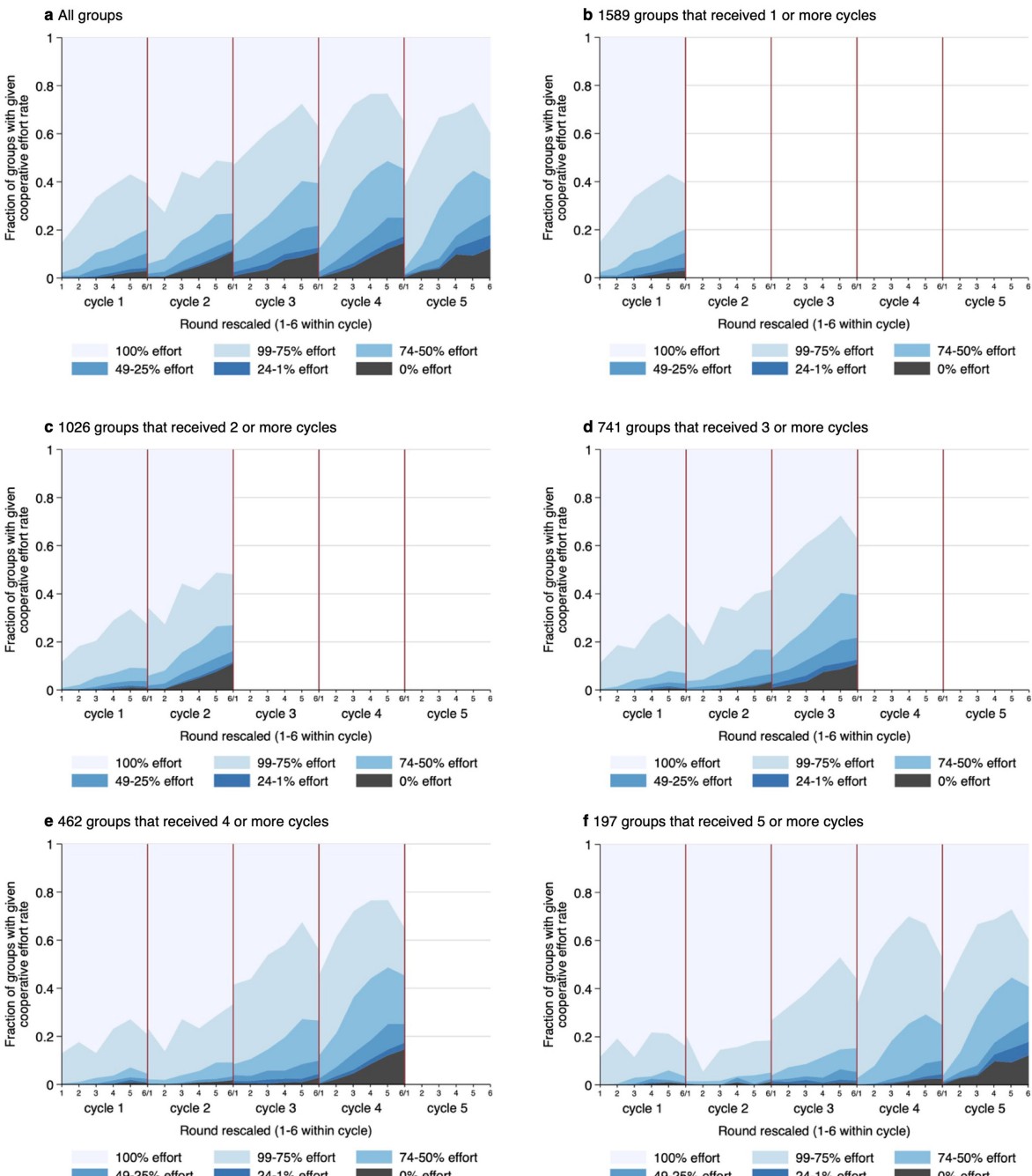

**Extended Data Fig. 2 | Longitudinal distribution of groups by cooperative effort rate.** Each panel plots the fraction of groups whose cooperative effort rate in a given round is (i) 0 percent, (ii) greater than 0 percent and less than 25 percent, (iii) greater than or equal to 25 percent and less than 50 percent, (iv) greater than or equal to 50 percent and less than 75 percent, (v) greater than or equal to 75 percent and less than 100 percent, and (vi) 100 percent. Vertical red lines indicate the end of a cycle and start of the new cycle. In panel **a**, the number of groups is 1589 in cycle 1; 1026 in cycle 2; 741 in cycle 3; 462 in cycle 4; and 197 in cycle 5.

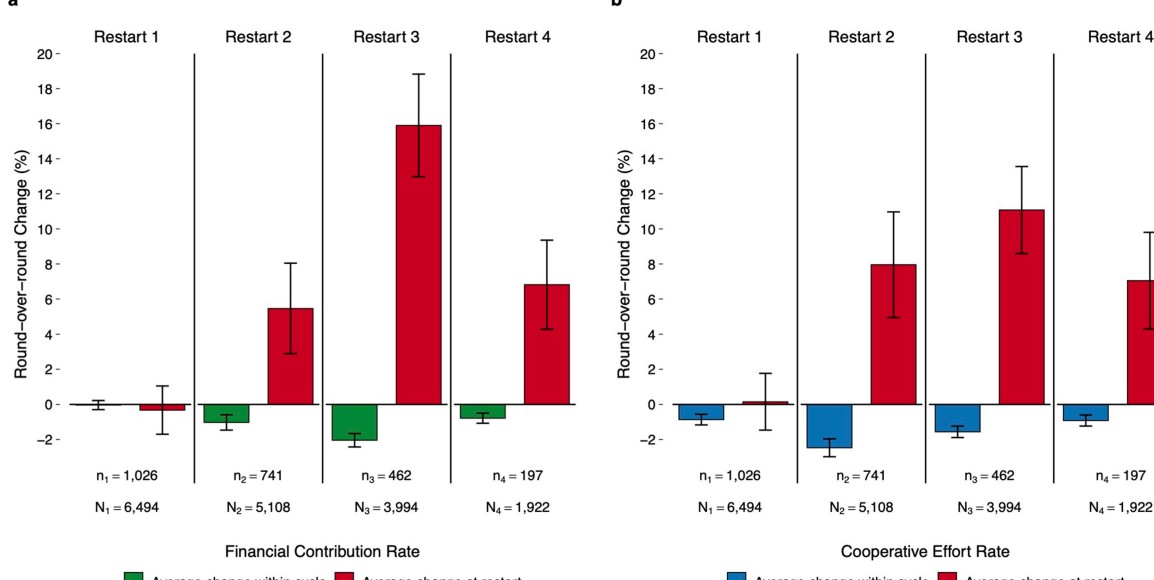

**Extended Data Fig. 3 | Sequential restart effects on cooperation rates for continuing groups with controls.** Cooperation rates decline round-over-round on average within loan cycles, but increase on average at restarts even when groups are stable. a, b For any given restart number, the average change within cycle is calculated as the within-group round-over-round difference in cooperation rates, averaged across all rounds in the loan cycle preceding the given restart and across all groups. The average change at restart is calculated as the difference in cooperation rates between the final round of the preceding loan cycle and the first round of the following loan cycle. Both averages are based on subsamples that are restricted to groups that go through a given restart to estimate effect sizes without attrition of groups. Estimates from regressions of the round-to-round change in the cooperation rate on an indicator that the change corresponds to a restart and group fixed effects, controlling for loan size, loan duration, calendar year fixed effects, and rainy season fixed effects (Supplementary Table 2). Subsample sizes: $n$ = number of groups that go through a given restart, $N$ = round-over-round total observations within cycle and at restart for a given restart. Error bars denote ± standard errors. a The Financial Contribution Rate is the percentage of the scheduled group payment amount received within 30 days. b The Cooperative Effort Rate is based on the number of days overdue before a group records any attempt at partial or full payment.

**Extended Data Table 1 | Descriptive statistics of the group lending quantitative dataset**

| | Cycle 1 | Cycle 2 | Cycle 3 | Cycle 4 | Cycle 5 | All cycles |
|---|---|---|---|---|---|---|
| Loan size[a] | 0.383 | 0.505 | 0.651 | 0.800 | 0.954 | 0.540 |
| | (0.107) | (0.146) | (0.181) | (0.168) | (0.221) | (0.224) |
| Loan duration[b] | 6.855 | 7.173 | 8.866 | 9.877 | 10.315 | 7.825 |
| | (1.027) | (1.323) | (1.120) | (0.910) | (1.144) | (1.651) |
| Group size | 4.469 | 4.476 | 4.393 | 4.022 | 3.508 | 4.358 |
| | (0.772) | (0.863) | (0.959) | (1.094) | (1.132) | (0.923) |
| Female[c] | 0.751 | 0.712 | 0.707 | 0.741 | 0.783 | 0.733 |
| | (0.292) | (0.304) | (0.304) | (0.289) | (0.283) | (0.297) |
| Married[c] | 0.943 | 0.932 | 0.926 | 0.934 | 0.944 | 0.936 |
| | (0.135) | (0.146) | (0.155) | (0.148) | (0.146) | (0.144) |
| Num children[d] | 3.085 | 3.122 | 3.144 | 3.174 | 3.322 | 3.127 |
| | (0.914) | (1.000) | (1.114) | (1.167) | (1.288) | (1.027) |
| Monthly sales[a] | 0.786 | 0.892 | 1.042 | 1.530 | 1.609 | 0.987 |
| | (0.364) | (0.502) | (0.633) | (1.028) | (1.012) | (0.658) |
| Business equity[a] | 0.922 | 1.005 | 1.263 | 1.723 | 2.077 | 1.155 |
| | (0.579) | (0.509) | (0.723) | (0.898) | (1.049) | (0.741) |
| Petty trader[c] | 0.585 | 0.517 | 0.499 | 0.591 | 0.667 | 0.557 |
| | (0.302) | (0.317) | (0.325) | (0.325) | (0.340) | (0.318) |
| Clothing[c] | 0.071 | 0.069 | 0.074 | 0.069 | 0.084 | 0.072 |
| | (0.143) | (0.140) | (0.168) | (0.175) | (0.210) | (0.154) |
| Food[c] | 0.139 | 0.170 | 0.204 | 0.179 | 0.125 | 0.163 |
| | (0.188) | (0.218) | (0.243) | (0.227) | (0.221) | (0.214) |
| Single-item sales[c] | 0.122 | 0.157 | 0.142 | 0.110 | 0.078 | 0.131 |
| | (0.169) | (0.207) | (0.215) | (0.201) | (0.172) | (0.193) |
| Service[c] | 0.012 | 0.010 | 0.009 | 0.011 | 0.004 | 0.010 |
| | (0.059) | (0.054) | (0.048) | (0.065) | (0.048) | (0.056) |
| Rainy season[e] | 0.459 | 0.519 | 0.480 | 0.482 | 0.505 | 0.484 |
| | (0.498) | (0.500) | (0.500) | (0.500) | (0.500) | (0.500) |
| City of residence | | | | | | |
| City A | 0.222 | 0.166 | 0.190 | 0.171 | 0.041 | 0.187 |
| City B | 0.275 | 0.404 | 0.482 | 0.517 | 0.492 | 0.385 |
| City C | 0.325 | 0.253 | 0.224 | 0.268 | 0.421 | 0.286 |
| Other | 0.178 | 0.177 | 0.104 | 0.043 | 0.046 | 0.142 |
| N loans disbursed | 1,589 | 1,026 | 741 | 462 | 197 | 4,015 |
| N repayment rounds | 10,855 | 7,308 | 6,505 | 4,523 | 2,008 | 31,199 |

Notes: Standard deviations in parentheses. [a]Average per group member in millions of SLL. [b]In months. [c]Proportion of the group members with the corresponding characteristic or engaged in the corresponding business type. [d]Average per group member. [e]Rainy season indicates the fraction of repayment rounds scheduled for the rainy season in Sierra Leone.

**Extended Data Table 2 | Sequential estimates of the restart effect on cooperation rates, no controls**

**a**

| Financial contribution rate | 1st cycle & 1st restart | 2nd cycle & 2nd restart | 3rd cycle & 3rd restart | 4th cycle & 4th restart | Overall |
|---|---|---|---|---|---|
| Mean round-to-round change | -0.378 (0.077) [p<0.0001] | -0.831 (0.133) [p<0.0001] | -0.784 (0.142) [p<0.0001] | -0.627 (0.177) [p=0.0004] | -0.635 (0.061) [p<0.0001] |
| Mean change at restart | 1.494 (0.412) [p=0.0003] | 4.311 (0.784) [p<0.0001] | 6.282 (1.088) [p<0.0001] | 5.441 (1.551) [p=0.0005] | 3.604 (0.382) [p<0.0001] |
| Diff in means (restart effect) | 1.872 (0.489) [p=0.0001] | 5.141 (0.917) [p<0.0001] | 7.066 (1.231) [p<0.0001] | 6.068 (1.728) [p=0.0004] | 4.239 (0.443) [p<0.0001] |
| Controls | N | N | N | N | N |
| N groups | 1,026 | 741 | 462 | 197 | 1,026 |
| N obs | 6,494 | 5,108 | 3,994 | 1,922 | 17,518 |
| $R^2$ | 0.0062 | 0.0149 | 0.0254 | 0.0180 | 0.0134 |

**b**

| Cooperative effort rate | 1st cycle & 1st restart | 2nd cycle & 2nd restart | 3rd cycle & 3rd restart | 4th cycle & 4th restart | Overall |
|---|---|---|---|---|---|
| Mean round-to-round change | -0.916 (0.101) [p<0.0001] | -0.973 (0.160) [p<0.0001] | -0.971 (0.162) [p<0.0001] | -1.072 (0.216) [p<0.0001] | -0.973 (0.078) [p<0.0001] |
| Mean change at restart | 0.412 (0.538) [p=0.4437] | -0.881 (0.944) [p=0.3507] | 6.540 (1.236) [p<0.0001] | 8.393 (1.892) [p<0.0001] | 1.897 (0.483) [p<0.0001] |
| Diff in means (restart effect) | 1.328 (0.639) [p=0.0377] | 0.092 (1.104) [p=0.9335] | 7.510 (1.398) [p<0.0001] | 9.465 (2.108) [p<0.0001] | 2.870 (0.561) [p<0.0001] |
| Controls | N | N | N | N | N |
| N groups | 1,026 | 741 | 462 | 197 | 1,026 |
| N obs | 6,494 | 5,108 | 3,994 | 1,922 | 17,518 |
| $R^2$ | 0.0013 | 0.0000 | 0.0183 | 0.0261 | 0.0040 |

Notes: Predicted restart effect estimated from OLS regressions of the group's round-to-round change in the financial contribution rate (Panel **a**) or cooperative effort rate (Panel **b**). The sample includes observations only from rounds that are part of a restart. For example, "1st cycle & 1st restart" compares the average round-to-round change in cooperation rates for all rounds in cycle 1 ("Mean round-to-round change") to the change in cooperation rates between the last round of cycle 1 and the first round of cycle 2, when the sample includes only groups that continue from cycle 1 to cycle 2. Regressors are an indicator for the restart round and group fixed effects. Standard errors clustered at the group level in parentheses, and p-values from two-sided t-tests of the coefficient estimates in brackets.

## Extended Data Table 3 | Interview sample client individual-level descriptive statistics

| Client ID | Ethnic Group[a] | Gender[b] | Marital[b] Status | Children[b] | Business[b] Type | Monthly[b,c] Sales | Language[d] |
|---|---|---|---|---|---|---|---|
| G1.C1 | Temne | Male | Married | 6 | Petty Trading | 1.25 | Temne |
| G1.C2 | Temne | Female | Widowed | 5 | Petty Trading | 1.00 | Temne |
| G2.C3 | Mende | Female | Married | 3 | Petty Trading | 0.90 | Krio |
| G2.C4 | Loko | Female | Married | 2 | Soap | 0.80 | Krio |
| G3.C5 | Temne | Female | Married | 2 | Palm Oil | 0.90 | Temne |
| G4.C6 | Fullah | Female | Married | 1 | Petty Trading | 0.80 | Krio |
| G4.C7 | Temne | Female | Married | 1 | Petty Trading | 0.90 | Temne |
| G5.C8 | Temne | Female | Married | 4 | Petty Trading | 0.80 | Temne |
| G6.C9 | Susu | Female | Married | 1 | Food | 1.00 | Krio |
| G6.C10 | Mende | Female | Married | 3 | Petty Trading | 1.00 | Krio |
| G7.C11 | Fullah | Female | Married | 4 | Petty Trading | 0.25 | Krio |
| G8.C12 | Temne | Female | Married | 4 | Petty Trading | 0.88 | Temne |
| G8.C13 | Temne | Female | Married | 2 | Palm Oil | 1.15 | Temne |
| G8.C14 | Mende | Female | Married | 1 | Petty Trading | 1.00 | Krio |
| G8.C15 | Temne | Female | Married | 2 | Petty Trading | 0.90 | Temne |
| G9.C16 | Temne | Female | Married | 6 | Petty Trading | 1.25 | Temne |
| G9.C17 | Temne | Female | Married | 6 | Petty Trading | 1.30 | Temne |
| G10.C18 | Temne | Female | Married | 4 | Petty Trading | 1.50 | Temne |
| G10.C19 | Temne | Female | Married | 3 | Petty Trading | 0.85 | Temne |
| G11.C20 | Loko | Female | Married | 5 | Petty Trading | 0.30 | Krio |
| G12.C21 | Temne | Female | Married | 1 | Petty Trading | 0.15 | Temne |
| G12.C22 | Temne | Female | Married | 3 | Petty Trading | 0.40 | Temne |
| G13.C23 | Susu | Male | Married | 6 | Petty Trading | 2.00 | Krio |
| G13.C24 | Temne | Female | Married | 3 | Petty Trading | 0.68 | Temne |
| G13.C25 | Mende | Female | Married | 4 | Petty Trading | 0.15 | Krio |
| G14.C26 | Mandingo | Female | Married | 2 | Petty Trading | 1.20 | Temne |
| G14.C27 | Temne | Female | Married | 6 | Petty Trading | 0.70 | Krio |
| G14.C28 | Mende | Female | Married | 3 | Petty Trading | 0.20 | Krio |
| G15.C29 | Temne | Female | Married | 2 | Petty Trading | 0.20 | Temne |
| G15.C30 | Temne | Female | Married | 3 | Petty Trading | 1.30 | Temne |
| G16.C31 | Temne | Female | Married | 3 | Petty Trading | 0.80 | Temne |
| G16.C32 | Temne | Female | Married | 4 | Petty Trading | 0.80 | Krio |
| G16.C33 | Temne | Female | Married | 4 | Petty Trading | 1.00 | Temne |
| G17.C34 | Temne | Female | Married | 2 | Petty Trading | 0.90 | Temne |
| G17.C35 | Temne | Female | Married | 3 | Food | 0.95 | Temne |
| G18.C36 | Temne | Female | Married | 3 | Petty Trading | 0.90 | Temne |
| G18.C37 | Temne | Female | Married | 3 | Petty Trading | 0.80 | Temne |
| G19.C38 | Loko | Female | Married | 1 | Food | 0.76 | Temne |
| G20.C39 | Temne | Female | Married | 2 | Petty Trading | 0.95 | Temne |
| G20.C40 | Loko | Female | Married | 3 | Petty Trading | 0.95 | Krio |
| G21.C41 | Sherbro | Female | Married | 4 | Petty Trading | 0.40 | Krio |
| G21.C42 | Mende | Male | Married | 4 | Clothing | 0.30 | Krio |
| G22.C43 | Temne | Female | Married | 2 | Petty Trading | 0.90 | Krio |
| G22.C44 | Loko | Male | Married | 2 | Petty Trading | 1.00 | Krio |
| G22.C45 | Temne | Female | Widowed | 2 | Petty Trading | 0.80 | Temne |
| G23.C46 | Temne | Female | Married | 3 | Petty Trading | 1.00 | Temne |
| G23.C47 | Susu | Female | Married | 5 | Petty Trading | 0.90 | Krio |
| G24.C48 | Temne | Female | Married | 4 | Food | 1.05 | Temne |
| G24.C49 | Temne | Female | Married | 2 | Palm Oil | 0.95 | Temne |
| G25.C50 | Temne | Female | Married | 3 | Petty Trading | 1.50 | Temne |
| G25.C51 | Temne | Female | Married | 3 | Petty Trading | 1.35 | Temne |
| G26.C52 | Susu | Female | Married | 2 | Petty Trading | 0.65 | Temne |
| G26.C53 | Temne | Male | Married | 3 | Petty Trading | 1.50 | Temne |
| G27.C54 | Limba | Female | Married | 2 | Petty Trading | 0.50 | Temne |
| G28.C55 | Temne | Female | Married | 5 | Firewood | 0.50 | Temne |
| G29.C56 | Temne | Female | Married | 5 | Petty Trading | 1.00 | Temne |
| G30.C57 | Temne | Female | Married | 3 | Petty Trading | 1.20 | Temne |
| G31.C58 | Yalunka | Female | Married | 2 | Petty Trading | 0.85 | Krio |
| G32.C59 | Temne | Female | Married | 4 | Food | 1.00 | Temne |
| G33.C60 | Limba | Female | Married | 6 | Petty Trading | 1.30 | Krio |
| G33.C61 | Temne | Female | Widowed | 7 | Petty Trading | 0.60 | Temne |
| G34.C62 | Temne | Female | Married | 3 | Petty Trading | 1.00 | Krio |
| G34.C63 | Temne | Female | Married | 1 | Petty Trading | 0.45 | Temne |
| G35.C64 | Temne | Female | Married | 5 | Petty Trading | 1.20 | Temne |

Notes: [a]Self-report at time of the interview. [b]Recorded prior to the first loan disbursement. [c]In millions of SLL. [d]Primary interview language at the preference of the client.

# Extended Data Table 4 | Coded evidence of self-reported motivations for cooperation from client interviews

| Client ID | Pay Own Share[a] | | | | Pay Other's Share[b] | | | |
|---|---|---|---|---|---|---|---|---|
| | Econ[c] | Duty/ Moral[d] | Reput/ Embarr[e] | Solid/ Trust[f] | Econ[c] | Duty/ Moral[d] | Reput/ Embarr[e] | Solid/ Trust[f] |
| G1.C1 | X | X | | X | | | | X |
| G1.C2 | | X | | | | | | |
| G2.C3 | X | X | X | X | | | | X |
| G2.C4 | X | | X | X | | | | X |
| G3.C5 | | X | | X | | X | | |
| G4.C6 | | X | | | | | | X |
| G4.C7 | X | | X | | | | | |
| G5.C8 | | X | X | X | | X | | |
| G6.C9 | X | X | | X | | | | X |
| G6.C10 | | X | X | X | | X | | |
| G7.C11 | X | X | | X | | X | | X |
| G8.C12 | X | X | | X | | | | X |
| G8.C13 | X | X | | X | | | | |
| G8.C14 | | X | | | | | | |
| G8.C15 | | | | X | | | | X |
| G9.C16 | X | X | | X | | | | X |
| G9.C17 | X | X | X | X | | X | | X |
| G10.C18 | X | X | | X | | | | |
| G10.C19 | X | X | | X | | | | |
| G11.C20 | | X | X | X | | X | | |
| G12.C21 | | | X | | | | X | |
| G12.C22 | X | | X | X | | | X | |
| G13.C23 | X | | | | | X | | |
| G13.C24 | | X | | | | | | |
| G13.C25 | X | | | | | | | |
| G14.C26 | | X | X | X | | | | X |
| G14.C27 | | | | X | | | | |
| G14.C28 | | X | X | X | | | | X |
| G15.C29 | | X | X | X | | | | |
| G15.C30 | | | | X | | | | |
| G16.C31 | | X | | X | | | | X |
| G16.C32 | | X | | X | | | | |
| G16.C33 | | | X | | | | | |
| G17.C34 | | X | X | X | | | | X |
| G17.C35 | | X | | X | | | | |
| G18.C36 | | X | | X | | | | X |
| G18.C37 | | X | | X | | X | | |
| G19.C38 | | X | | X | | | | |
| G20.C39 | X | X | X | X | | | | |
| G20.C40 | | X | | X | | | | |
| G21.C41 | X | X | X | X | | | | X |
| G21.C42 | | | | X | | | X | |
| G22.C43 | | X | | X | | | | X |
| G22.C44 | | X | | X | | | | |
| G22.C45 | X | X | X | X | | | | |
| G23.C46 | | X | | X | | X | | X |
| G23.C47 | | X | X | | | | | |
| G24.C48 | | | X | X | | | | |
| G24.C49 | | X | | X | | | | |
| G25.C50 | | | X | | | | | |
| G25.C51 | X | | X | | | X | | |
| G26.C52 | | X | X | X | | | | |
| G26.C53 | | | | X | | X | | X |
| G27.C54 | | | | X | | | | X |
| G28.C55 | | X | X | | | | | X |
| G29.C56 | | | | | | | | |
| G30.C57 | | X | X | | | X | X | |
| G31.C58 | | X | | X | | | | |
| G32.C59 | X | X | | X | | X | | |
| G33.C60 | X | | | | | X | | |
| G33.C61 | | | | X | | | | X |
| G34.C62 | | X | | X | | | | |
| G34.C63 | | X | X | X | | | | |
| G35.C64 | X | | X | X | | | | |
| Total[g] | 34.4% | 68.8% | 40.6% | 75.0% | 0.0% | 21.9% | 6.3% | 34.4% |

Notes: [a]Motives to pay one's own share of the group loan. [b]Motives to pay for another member's share of the group loan. [c]Economic coding criteria: reports related to financial incentives, e.g. future loan access. [d]Duty/Morality coding criteria: reports related to duty, obligation, morality. [e]Reputation/Embarrassment coding criteria: reports related to social reputation, shame, or embarrassment. [f]Solidarity/Trust coding criteria: reports related to solidarity, trust, prosocial help, inequality aversion. [g]Total values calculated as the percent of all interviewed clients reporting the motive.

**Extended Data Table 5 | Coded evidence of free-riding from client interviews**

| Group ID | Evidence of Free-Riding[a] | | | Compensating for Free-Riders[b] | | Excluding of Free-Riders[c] | |
| --- | --- | --- | --- | --- | --- | --- | --- |
| | Full | Partial | None | Yes | No | Remained | Excluded |
| G1 | | X | | X | | X | X |
| G2 | | X | | X | | X | |
| G3 | | X | | X | | X | X |
| G4 | | X | | X | X | X | X |
| G5 | | X | | | X | X | |
| G6 | | X | | X | X | X | X |
| G7 | | X | | X | | X | |
| G8 | | | X | | | | |
| G9 | | X | | X | | X | |
| G10 | | X | | | X | X | |
| G11 | | X | | X | | | X |
| G12 | | X | | X | | | X |
| G13 | | X | | X | | | X |
| G14 | | | X | | | | |
| G15 | | | X | | | | |
| G16 | | X | | X | | X | |
| G17 | | X | | X | | X | |
| G18 | | X | | X | X | X | |
| G19 | | X | | | X | | X |
| G20 | | X | | | X | X | |
| G21 | | X | | X | | | X |
| G22 | | X | | X | | X | |
| G23 | | X | | X | | | X |
| G24 | | X | | | X | | X |
| G25 | | X | | X | | X | X |
| G26 | | X | | | X | X | |
| G27 | | X | | X | X | | X |
| G28 | | X | | X | | | X |
| G29 | | X | | X | | X | |
| G30 | | X | | X | | | X |
| G31 | | X | | | X | X | |
| G32 | | X | | X | | | X |
| G33 | | X | | | X | X | |
| G34 | | X | | X | | X | |
| G35 | | X | | | X | | X |
| Total[d] | 0% | 91.4% | 8.6% | 65.7% | 37.1% | 57.1% | 48.6% |

Notes: [a]Free-riding coding criteria: Full free-riding within the group by one or more individuals, coded if reported that at least one member received the loan disbursement and then made zero contributions; Partial free-riding within the group by one or more individuals, coded if reported that at least one member did not make full contributions, all months, on-time; No free-riding by any individuals within the group, coded if reported that all members made full contributions, all months, on-time. [b]Compensating for free-riders coding criteria: Coded "Yes" if the group included a full or partial free-rider and at least one group member reported paying some portion of the given free-rider's share. Coded "No" if the group included a full or partial free-rider and no one in the group reported paying some portion of the given free-rider's share. Yes/No is not mutually exclusive: a group may have more than one free-rider and compensate for one free-rider, but not another. [c]Excluding of free-riders coding criteria: Coded if reported that at least one full or partial free-rider was allowed to remain in (or be excluded) from the next loan cycle. We include the Remained/Excluded information for all groups with free-riders even if it was uncertain at the time of the interview whether the group would advance to a subsequent loan cycle. Remained/Excluded is not mutually exclusive: a group may have more than one free-rider and exclude one free-rider, but not another. [d]Totals are calculated as a percent of all groups. Percents for compensating for and excluding of free-riders may total more than 100% as groups may have more than one free-rider and exhibit multiple behaviours dependent on the free-rider.

**Extended Data Table 6 | Heterogeneity of individual contributions and frequency of cooperative patterns from client interviews**

| Category[a] | Schematic of Cooperative Pattern | Description[c] | Coded Groups | Frequency |
|---|---|---|---|---|
| No Defection | (Round/Month 1–6; Members 1–5; all white cells) | No evidence of defection by any member. | G8, G14, G15 | 8.6% |
| Walk-away Defection | (Round 1–6; Members 1–5; Member 2 defects rounds 3–6) | Member(s) defect and never contribute again. Other member(s) continue to contribute. | G1, G32, G4, G11, G23, G28 | 17.1% |
| Cascade | (Round 1–6; Members 1–5; Member 2 defects rounds 3–6, Member 3 defects rounds 4–6, Member 4 defects rounds 5–6) | Increasing members defect and never contribute again. Other member(s) continue to contribute. | G3, G25, G30 | 8.6% |
| Intermittent Defection | (Round 1–6; Members 1–5; intermittent defections: Member 2 round 5, Member 4 rounds 3 and 5, Member 5 round 6) | Member(s) defect on one or more rounds, but contribute in later rounds. | G2, G12, G20, G31, G5, G13, G21, G33, G6, G16, G22, G34, G7, G17, G24, G35, G9, G18, G26, G10, G19, G29 | 62.9% |
| Full Collapse | (Round 1–6; Members 1–5; Members 1,5 defect rounds 4–6; Members 2,3,4 defect rounds 3–6) | All members defect and stop contributing. | G27 | 2.9% |
| Total | | | 35 | 100.0% |

▢ Individual contribution[b]

◼ Individual defection[b]

Notes: [a]Repayment patterns based on the coding of individual contributions or defections reported in client interviews. Interviews were conducted on a representative sample of 35 joint-liability groups (consisting of 158 members). The cooperative patterns refer to the group's current or most recent loan cycle at the time of the interview to minimize recall bias, and were cross-validated with secondary interviews and group-level repayment records from the lending institution. [b]Individual Coding Criteria: Individual behaviour was characterized as defection if the member was reported as unable or unwilling to contribute their share of the loan in a given round (month). More specifically, a member was coded as exhibiting "walk-away defection" if the individual defected for more than one round and did not contribute again during the loan cycle. A member was coded as exhibiting "intermittent defection" if the individual defected for one or more rounds, but later contributed in a subsequent round of the loan cycle. [c]Group Categorization Criteria: A group's category was determined by aggregating its members' individual patterns of defection to identify its predominate pattern. The categories were defined to capture substantive differences in within-group repayment dynamics. See Supplementary Information on Heterogeneity of Cooperative Dynamics for further detail on coding and categorization.

**Extended Data Table 7 | Cycle-to-cycle transition frequencies of group size for continuing groups**

Group Size Cycle 2

| | 2 | 3 | 4 | 5 | 6 | N |
|---|---|---|---|---|---|---|
| 2 | 0.90 | 0.10 | 0.00 | 0.00 | 0.00 | 10 |
| 3 | 0.17 | 0.81 | 0.02 | 0.00 | 0.00 | 101 |
| 4 | 0.04 | 0.17 | 0.78 | 0.01 | 0.00 | 134 |
| 5 | 0.01 | 0.03 | 0.07 | 0.89 | 0.00 | 771 |
| 6 | 0.10 | 0.10 | 0.20 | 0.10 | 0.50 | 10 |

(rows: Group Size Cycle 1)

Group Size Cycle 3

| | 2 | 3 | 4 | 5 | 6 | N |
|---|---|---|---|---|---|---|
| 2 | 0.88 | 0.06 | 0.06 | 0.00 | 0.00 | 17 |
| 3 | 0.15 | 0.80 | 0.02 | 0.03 | 0.00 | 60 |
| 4 | 0.13 | 0.17 | 0.69 | 0.02 | 0.00 | 108 |
| 5 | 0.03 | 0.05 | 0.07 | 0.86 | 0.00 | 552 |
| 6 | 0.00 | 0.00 | 0.00 | 0.00 | 1.00 | 4 |

(rows: Group Size Cycle 2)

Group Size Cycle 4

| | 2 | 3 | 4 | 5 | 6 | N |
|---|---|---|---|---|---|---|
| 2 | 0.96 | 0.04 | 0.00 | 0.00 | 0.00 | 25 |
| 3 | 0.17 | 0.83 | 0.00 | 0.00 | 0.00 | 59 |
| 4 | 0.09 | 0.27 | 0.64 | 0.00 | 0.00 | 70 |
| 5 | 0.05 | 0.12 | 0.11 | 0.72 | 0.00 | 305 |
| 6 | 0.00 | 0.00 | 0.00 | 0.67 | 0.33 | 3 |

(rows: Group Size Cycle 3)

Group Size Cycle 5

| | 2 | 3 | 4 | 5 | 6 | N |
|---|---|---|---|---|---|---|
| 2 | 1.00 | 0.00 | 0.00 | 0.00 | 0.00 | 19 |
| 3 | 0.23 | 0.72 | 0.04 | 0.00 | 0.00 | 47 |
| 4 | 0.14 | 0.06 | 0.80 | 0.00 | 0.00 | 35 |
| 5 | 0.14 | 0.19 | 0.12 | 0.56 | 0.00 | 95 |
| 6 | 0.00 | 0.00 | 1.00 | 0.00 | 0.00 | 1 |

(rows: Group Size Cycle 4)

Notes: Values indicate the proportion of groups that become of a given group size in the subsequent cycle (columns) conditional on the group size in the current cycle (rows). Column N indicates the conditional sample size.

# Reporting Summary

## Statistics

For all statistical analyses, confirm that the following items are present in the figure legend, table legend, main text, or Methods section.

| n/a | Confirmed | |
|---|---|---|
| ☐ | ☒ | The exact sample size (*n*) for each experimental group/condition, given as a discrete number and unit of measurement |
| ☐ | ☒ | A statement on whether measurements were taken from distinct samples or whether the same sample was measured repeatedly |
| ☐ | ☒ | The statistical test(s) used AND whether they are one- or two-sided<br>*Only common tests should be described solely by name; describe more complex techniques in the Methods section.* |
| ☐ | ☒ | A description of all covariates tested |
| ☐ | ☒ | A description of any assumptions or corrections, such as tests of normality and adjustment for multiple comparisons |
| ☐ | ☒ | A full description of the statistical parameters including central tendency (e.g. means) or other basic estimates (e.g. regression coefficient) AND variation (e.g. standard deviation) or associated estimates of uncertainty (e.g. confidence intervals) |
| ☐ | ☒ | For null hypothesis testing, the test statistic (e.g. *F*, *t*, *r*) with confidence intervals, effect sizes, degrees of freedom and *P* value noted<br>*Give P values as exact values whenever suitable.* |
| ☒ | ☐ | For Bayesian analysis, information on the choice of priors and Markov chain Monte Carlo settings |
| ☐ | ☒ | For hierarchical and complex designs, identification of the appropriate level for tests and full reporting of outcomes |
| ☐ | ☒ | Estimates of effect sizes (e.g. Cohen's *d*, Pearson's *r*), indicating how they were calculated |

*Our web collection on statistics for biologists contains articles on many of the points above.*

## Software and code

Policy information about availability of computer code

| | |
|---|---|
| Data collection | The quantitative data was originally collected for the purpose of administering microfinance services in Sierra Leone from 2005 to 2011. The data were electronically recorded by the lending and accounting staff in the organization's Management Information System (MIS). The quantitative data were recorded independently of the research team. |
| Data analysis | For the quantitative data, the statistical analysis and figure generation were conducted using STATA 18.5, including package Markstat 2.1, and the software R version 4.2.3.<br><br>For the interview data, audio recordings of the interviews were manually transcribed verbatim by the research team. The data was managed and coded using NVivo software version 14.24.1.<br><br>Code to reproduce the statistical analysis and figure source data are publicly available at the Open Science Framework (https://doi.org/10.17605/OSF.IO/26BFC) |

For manuscripts utilizing custom algorithms or software that are central to the research but not yet described in published literature, software must be made available to editors and reviewers. We strongly encourage code deposition in a community repository (e.g. GitHub). See the Nature Portfolio guidelines for submitting code & software for further information.

# Data

Policy information about availability of data

All manuscripts must include a data availability statement. This statement should provide the following information, where applicable:

- Accession codes, unique identifiers, or web links for publicly available datasets
- A description of any restrictions on data availability
- For clinical datasets or third party data, please ensure that the statement adheres to our policy

The deidentified group lending data is publicly available at the Open Science Framework (https://doi.org/10.17605/OSF.IO/26BFC)

# Research involving human participants, their data, or biological material

Policy information about studies with human participants or human data. See also policy information about sex, gender (identity/presentation), and sexual orientation and race, ethnicity and racism.

| | |
|---|---|
| Reporting on sex and gender | The data includes information on self-reported gender as recorded by the lending institution during the loan application process.  Consistent with the organization's social objectives, 73.3% of the clients are self-reported female. Regression analysis in the supplementary information includes the proportion of female clients in a group as a covariate. In the main text, the econometric model employs group fixed effects such that time-consistent group-level covariates are not necessary. |
| Reporting on race, ethnicity, or other socially relevant groupings | Self-reported ethnic group was recorded in the interview sample.  Anonymized descriptive statistics for the interview sample are included in the Extended Data.<br><br>We do not include data on race, ethnicity, or other socially relevant groupings in the quantitative analysis. In the supplemental analyses, we do include direct economic measures of clients' monthly sales and business equity as reported to and verified by the lending institution. |
| Population characteristics | See below in "Research sample." |
| Recruitment | The quantitative data sample is limited to clients in Sierra Leone that received a joint-liability loan at the microfinance institution described in the study. Consistent with the organization's focus on poverty alleviation, all clients are low-income. Furthermore, microcredit group members must meet basic eligibility criteria. Specifically, each client is required to have their own micro-business capable of meeting the minimum loan repayments and members of the same group cannot be direct kin, i.e., parents, spouses, or siblings. The sample is representative of the population of joint-liability clients at the microfinance institution. However, please note that microfinance clients are not representative of the adult population of Sierra Leone more broadly, given that individuals self-select into applying for loans at the microfinance institution. Therefore, results do not necessarily generalize to the adult population of Sierra Leone more broadly, and we make no claims in the paper of such generalizability.<br><br>The sample of clients for interviews was drawn from the overall quantitative dataset, using a two-stage cluster random sampling, plus a purposive enhancement. The sample of clients for interviews is representative of the quantitative data sample, but, as stated previously, cannot be seen as being representative of the adult population of Sierra Leone more broadly, and we make no claims in the paper of such generalizability. In the first stage of the random sampling, we used simple randomization of groups, after restricting the population of potential groups based on two criteria: (i) we geographically restricted the pool to groups that were administered at the lending institution's principal branch. This was implemented for practical efficiency of interview logistics; (ii) we restricted the pool to groups that had been engaged in borrowing within the last six months. This was implemented to reduce recall bias during the interviews. This resulted in 35 joint liability groups drawn by simple randomization from the subpopulation. Supplementary Table 6 provides descriptive statistics of the interview sample at the group-level. In the second stage of the random sampling, we selected one member per group to be interviewed using simple randomization within the group. We then enhanced this sampling design by implementing a purposive sampling of an additional member from within the randomly selected groups. The choice of whether to conduct an additional interview and with which specific member was based on the content provided in the first member's interview, following the researcher's discretion regarding which additional group member's perspective would provide the most valuable information. For example, if the first interviewee indicated that a specific member "X" had been the main source of cooperative disruption in the group, member X was selected for a direct interview to hear his or her version of the events. The intent of additional within-group interviews was to cross-validate the initial interview, collect potentially contradictory data, and understand a complex phenomenon from different points of view. This resulted in 29 additional interviews, producing a total of 64 client interviews. Extended Data Table 3 provides descriptive statistics of the client interview sample at the individual-level. Interviews were also conducted with a non-random sample of 9 staff members of the lending institution – including three loan officers, two information and accounting officers, two loan portfolio managers, and two executive directors – regarding organization policies, practices in the field, and the organization's record keeping process. Supplementary Table 7 provides descriptive statistics of the staff interview sample. The interview content was instrumental in accurately interpreting the quantitative dataset and provided context and further cross-validation of the borrowers' descriptions. |
| Ethics oversight | This study conducts analysis of secondary data originally collected by the microfinance institution and primary interview data collected by the principal investigator. This research was approved by the Central University Research Ethics Committee |

(CUREC) at the University of Oxford. Reference Number: SSD/CUREC1A/10-099. The approval included the collection process and analysis of administrative microfinance data and of primary data from human participants collected through semi-structured interviews. The research was performed in accordance with all relevant guidelines and regulations outlined by CUREC. The analysis and publication of the secondary administrative data follow the Data Use Agreement indicating that the research team may disclose and/or publish data such that "the data shall be anonymized and/or aggregated so that no reference is made to individuals' names, applying to both [microfinance institution] clients and staff." Before each interview during primary data collection, the research purpose and use of the interview data was explained to the client or staff member and a physical document of informed consent was reviewed together. All participants provided informed consent by signature or thumbprint. To safeguard participant confidentiality, personally identifiable information was redacted from the interview data.

Note that full information on the approval of the study protocol must also be provided in the manuscript.

# Field-specific reporting

Please select the one below that is the best fit for your research. If you are not sure, read the appropriate sections before making your selection.

☐ Life sciences  ☒ Behavioural & social sciences  ☐ Ecological, evolutionary & environmental sciences

For a reference copy of the document with all sections, see nature.com/documents/nr-reporting-summary-flat.pdf

# Behavioural & social sciences study design

All studies must disclose on these points even when the disclosure is negative.

| | |
|---|---|
| Study description | A field study of long-term, high-stakes cooperative behavior that is based on the analysis of a longitudinal quantitative dataset and an interview-based qualitative dataset. |
| Research sample | The sample involves microfinance clients in Sierra Leone participating in group loans. The microfinance institution had approximately 18,000 borrowers at the time of data collection spread throughout multiple geographic regions in Sierra Leone spanning both urban and rural areas. The lending groups are small, typically five members, and are formed through a self-selection process. Loan officers ensure that each member meets basic eligibility criteria; most notably, each client is required to have their own micro-business capable of meeting the minimum loan repayments. Typical micro-business examples include petty trading, food service, barbershop, tailoring, motorbike taxi service. Members of the same group cannot be direct kin, i.e., parents, spouses, or siblings.<br><br>The quantitative data sample is representative of the population of joint-liability clients at the microfinance institution. However, please note that microfinance clients are not representative of the adult population of Sierra Leone more broadly, given that individuals select into applying for loans at the microfinance institution. Therefore, results do not necessarily generalize to the adult population of Sierra Leone more broadly, and we make no claims in the paper of such generalizability.<br><br>The rational for analyzing the quantitative research sample is that it provides a real-world social dilemma in which cooperative behavior with high stakes to the participants are tracked in the long-term. The social dilemma occurs because borrowers enter a joint-liability contract such that if the group loan is not repaid in full, all group members are held financially responsible regardless of who defaulted. The same dilemma is faced by group members on a monthly basis for up to five years. The quantitative data sample was chosen because it comprised all clients who received a joint-liability loan at the microfinance institution during the study period. 73.3% of the sample were female. 93.6% of the sample were married. The average number of children was 3.1. We have no information regarding age of the clients.<br><br>The rational for the qualitative research sample was to perform two key functions in the study: (1) the data help contextualize the cooperative dilemma, to ensure proper understanding of the quantitative patterns and their appropriate interpretation; (2) the data provide insight to the behavioral mechanisms underlying the longitudinal trends. |
| Sampling strategy | For the quantitative dataset, the sample includes all group loans administered by the microfinance institution during a period between 2005 and 2011, with minor data exclusions noted below. The dataset includes 47,931 group payments, partial or full, (corresponding to 31,199 scheduled monthly payments) made by 7,108 borrowers (constituting 1,589 unique lending groups) over five years. Sample size was not pre-determined. Models in the study document the statistical significance and confidence intervals of the empirical relationships. The cooperative dynamics in the main analysis are strongly significant with P-values less than 0.001.<br><br>For the interview dataset, the sample consists of 73 in-depth semi-structured interviews: 64 interviews with group lending clients and 9 interviews with members of the lending institution staff. The sample of clients for interviews was selected to be representative of the quantitative data sample. Moreover, it included a purposive second stage to capture perspectives by multiple clients involved in a given cooperative incident.<br><br>In the first stage of the random sampling, we used simple randomization of groups, after restricting the population of potential groups based on two criteria: (i) we geographically restricted the pool to groups that were administered at the lending institution's principal branch. This was implemented for practical efficiency of interview logistics; (ii) we restricted the pool to groups that had been engaged in borrowing within the last six months. This was implemented to reduce recall bias during the interviews. This resulted in 35 lending groups drawn by simple randomization from the subpopulation. In the second stage of the random sampling, we selected one member per group to be interviewed using simple randomization within the group. We then enhanced this sampling design by implementing a purposive sampling of an additional member from within the randomly selected groups. The choice of |

whether to conduct an additional interview and with which specific member was based on the content provided in the first member's interview, following the researcher's discretion regarding which additional group member's perspective would provide the most valuable information. For example, if the first interviewee indicated that a specific member "X" had been the main source of cooperative disruption in the group, member X was selected for a direct interview to hear his or her version of the events. The intent of additional within-group interviews was to cross-validate the initial interview, collect potentially contradictory data, and understand a complex phenomenon from different points of view. This resulted in 29 additional interviews, producing a total of 64 client interviews. Interviews were also conducted with a non-random sample of 9 staff members of the lending institution, including three loan officers, two information and accounting officers, two loan portfolio managers, and two executive directors, regarding organization policies, practices in the field, and the organization's record keeping process.

**Data collection**

The quantitative data was collected by the lending institution for the primary purpose of administering microfinance services. Client demographics and group loan repayment behavior were electronically recorded in the organization's Management Information System (MIS) by staff. The research team was not involved with the quantitative data collection. The group repayments occur at the local branch office of the microfinance institution. It is not required that all group members be present when making a group payment.

The interview data is based on 73 in-depth semi-structured interviews. On average, client interviews lasted 39 minutes and staff interviews lasted 1 hour and 34 minutes. Interview time totaled 56 hours. The interviews were conducted in person in Sierra Leone by the principal investigator between April 5, 2011 and June 6, 2011, contemporaneous to the collection of the quantitative data. The principal investigator was not blinded to the purpose of the interview, which was to collect clients' perspectives and experiences.

**Timing**

The group loans were administered between 2005 and 2011 by the microfinance institution.

**Data exclusions**

The dataset includes all group loans of up to five loan cycles disbursed and scheduled to complete repayment during a period between 2005 and 2011. Loan cycles greater than five are not included in the analysis. The sample size decreases with each loan cycle, with few groups having taken more than five cycles (approximately 3% of groups). We restrict the analysis to the first five loan cycles because the sample size of the remaining cycles is not sufficient for statistical analysis of longitudinal trends. We exclude from the analysis three groups due to missing data on an antecedent cycle (e.g., existing data on loan cycle 1 and 3 but missing data on cycle 2).

**Non-participation**

After the clients of a group receive a loan disbursement, the dataset has a full record of each group's longitudinal repayment behavior. In addition to timely repayment, this record may include a group's late, partial, or complete lack of repayment (which may be conceptualized as "non-participation.") This information is used to measure each group's cooperation over time.

**Randomization**

The collection and construction of the quantitative data did not involve randomization, as we used the full sample of group-lending clients of the microfinance institution. Randomization was involved in two-stage clustered random sampling used to determine the set of clients to be interviewed. Randomization was implemented in spreadsheet software.

# Reporting for specific materials, systems and methods

We require information from authors about some types of materials, experimental systems and methods used in many studies. Here, indicate whether each material, system or method listed is relevant to your study. If you are not sure if a list item applies to your research, read the appropriate section before selecting a response.

## Materials & experimental systems

| n/a | Involved in the study |
|---|---|
| ☒ | ☐ Antibodies |
| ☒ | ☐ Eukaryotic cell lines |
| ☒ | ☐ Palaeontology and archaeology |
| ☒ | ☐ Animals and other organisms |
| ☒ | ☐ Clinical data |
| ☒ | ☐ Dual use research of concern |
| ☒ | ☐ Plants |

## Methods

| n/a | Involved in the study |
|---|---|
| ☒ | ☐ ChIP-seq |
| ☒ | ☐ Flow cytometry |
| ☒ | ☐ MRI-based neuroimaging |

# Plants

Seed stocks

*Report on the source of all seed stocks or other plant material used. If applicable, state the seed stock centre and catalogue number. If plant specimens were collected from the field, describe the collection location, date and sampling procedures.*

Novel plant genotypes

*Describe the methods by which all novel plant genotypes were produced. This includes those generated by transgenic approaches, gene editing, chemical/radiation-based mutagenesis and hybridization. For transgenic lines, describe the transformation method, the number of independent lines analyzed and the generation upon which experiments were performed. For gene-edited lines, describe the editor used, the endogenous sequence targeted for editing, the targeting guide RNA sequence (if applicable) and how the editor was applied.*

Authentication

*Describe any authentication procedures for each seed stock used or novel genotype generated. Describe any experiments used to assess the effect of a mutation and, where applicable, how potential secondary effects (e.g. second site T-DNA insertions, mosiacism, off-target gene editing) were examined.*

