## [Peer Review File · Nature]

Punctuated Decline of Human Cooperation

Corresponding Author: Dr Nicholas Sabin

Version 0:

Reviewer comments:

Referee #1

(Remarks to the Author)

This paper present fascinating field data from a microfinance institution in Sierra Leone. This institution gives loans to groups of people who are mutually responsible to repay the loan. Thus, repaying provides a public good for the group, and the data allows to investigate how cooperation develops over time. Two observations are made. First, cooperation declines. It declines within a credit cycle, and the decline increases across the cycles. Second, there are strong restart effects. After each credit cycle the cooperation jumps back to almost full cooperation. This pattern resembles restart effect in public good experiments. However, there are important differences to the lab studies. The main difference is that there are strong reputation effects. Not repaying spoils your reputation towards the group members and beyond. In addition, if the group does not repay, the group members will not receive a loan again. In particular, this last point makes turns the public goods game into a threshold public goods game, which has a very different game theoretic structure than the standard public goods game, in which there is a dominant strategy in payoffs.

The paper is generally well written, the study interesting and the data analysis convincing. In the following, I first raise my main concerns and then mention some further points

Main concerns

It is unclear to me how exactly the threat works. I guess that loans are possible again when the loan is repaid, even when the repayment is delayed; and what is measured is repayment in time. A better description of this mechanism is necessary in order to understand the incentives in case someone is in trouble or unwilling to repay. The authors explain in detail that failed repayment is not due to inability to repay but due to lack of cooperation. This part was particularly puzzling to me because I wondered how the defectors will be treated by the other group members and the microfinance institution. It is astonishing that "voluntary" free riding makes sense. However, if people notice that the consequences of free riding are not that harsh, learning could lead to a decline in cooperation.

I miss theory; not necessarily in the form of a fully elaborated model but in the form of an analysis of the incentives. As mentioned above, the game has more the characteristic of a step level public goods game. This means that players have an incentive to offset or compensate the other players' contributions. In this respect, the question might less be why people reduce cooperation but how people punish, a question addresses by the authors in Sabin & Reed-Tsochas (2020).

Further points

Is there data on the individual level? It would be interesting to analyze the individual observation, in particular how people respond to underpayment.

The first sentence in the abstract, "[d]espite the tendency of human cooperation to decay over time, it can be punctuated by 13 substantial rebounds in cooperative behavior" raises the expectation that the punctuations occur spontaneously.

However, the restart time points are given by the economic environment.

Reference

Sabin, Nicholas, and Felix Reed-Tsochas. 2020. 'Able but Unwilling to Enforce: Cooperative Dilemmas in Group Lending', *American Journal of Sociology*, 125: 1602-67.

Referee #2

(Remarks to the Author)

The study documents longitudinal patterns in cooperation over a 5-year period in the context of group-lending in Sierra Leone, using data from the micro-finance organization that provided these loans. Unlike a lot of empirical studies of cooperation, the data for this study come from a real-life context. This feature is important because uncertainties about the external validity of laboratory economic games severely limits what inferences we can draw about real-world behavior from experiments. Moreover, unlike most naturalistic settings of cooperation, this setting offers some of the useful features of consistency and control that we seek in experiments: the economic incentives are the same across the loan cycles; the composition of each group stays the same; there is a well-defined time cycle that marks when a round of cooperation starts and ends; there are objective measurable outcomes (amount of loan repaid each month, timeliness of making partial or complete payments) to keep track of the level of cooperation. The data span 5 years, which is exceptionally long, given that most of the empirical studies of cooperation in repeated games is from experiments that last a few hours.

The authors argue that the results show a robust pattern that reveals something about the psychological mechanisms of cooperation: Cooperation deteriorates within rounds of a loan cycle, but also rebounds in each new loan cycle (even though group composition has not changed). The rate of decay increases across cycles. While other studies have documented declining cooperation over time, this study has a couple of neat additional insights. First, because cooperation rebounds in each new cycle, despite no change in group composition or the incentive structure, it is clear that the decline of cooperation is not due to people learning about the game structure over time. Second, the authors argue that changes in ability to cooperate is unlikely to explain the behavior, as people had a reserve account in which there was enough money kept aside to repay the loan, in case the business they were operating did poorly or other hardships arose.

While it is a useful study documenting an interesting finding, I feel that the paper doesn't have the theoretical import that I was hoping to see from a study of cooperation in consideration for publication at Nature.

First, as far as I can tell, this is not the typical kind of dilemma that characterizes evolutionary models of n-person cooperation. In evolutionary models (and in most laboratory games that attempt to test these models) the game is a public-goods game — individual payoff is maximized by not cooperating, and group payoff is maximized if everyone in the group cooperates. In the current study, individuals maximize their payoffs by cooperating. Receiving a loan is financially beneficial; therefore, repaying the loan so that you get a loan again in the next cycle is in the individual's best interest. The fact that individuals are failing to repay therefore is not because they are "defecting". Rather, it is something else about the psychology that is leading them to make a decision that is not in their longterm financial interest. The group-lending schema is only helping to bolster against the decay in individuals' irrational behavior, by making it beneficial for group mates to pressure their peers to stay on course.

The only way that "non-repayment" is "defection" is if the other group mates step in to pay for the laggard in their group. This could potentially happen, but we have absolutely no information about whether this was how people behaved. The only data the authors have is from what the lending organization documented as the group's payment. In that sense, the data is too thin (despite its amazing time-depth).

The authors argue that the temporal pattern of decay can be attributed to psychological mechanisms, and not learning or strategic motivations. This is true, but there is no contextual data on what happened within these groups that shed light on what these psychological mechanisms are. Obviously, getting this information would not be feasible, given that the study is based on an existing dataset created by the lending organization. The authors allude to conditional cooperation as a mechanism for the decay, but is it? An alternative is that the decline is due to extrinsic factors that affect individuals within a group with some probability, and affects individuals' ability to pay or motivation to continue to run his or her business. The accumulation of these events within a cycle to individuals within a group could explain the decline.

To rule out this and other possibilities, we need complementary interview or survey data at least from a subset of these groups on why people stopped repaying, and how the decision of others in their group affected their motivation to repay. Group-lending assumes that ostracism and sanctioning may boost loan compliance, but we have no idea if there was any social sanctioning within the study groups. As a thought experiment, if instead of group-lending, we are looking at a similar dataset in which lenders give out loans to individuals, how different would the loan compliance behavior look? It is possible that the same pattern we see here will be seen in individual-lending too. So, it is not clear if these patterns are stemming from group-dynamics of cooperation, or from individuals making not-so-great choices for themselves.

Lastly, the pattern of decline & rebound is much reduced when looking at groups that go on to another loan cycle. This means that most of the pronounced effects of Figure 1 are driven by the self-selection effect. Many groups fail to sustain cooperation and are removed from the mix. The rebound in Fig 1 is possibly from groups that ended high in the previous cycle, starting similarly high in the next cycle but without the failing groups pulling down the average financial contribution rate on the graph. The authors have done the appropriate analysis to examine what is left after the self-selection effect, but this is shown later. Fig 1 should not be the main figure in the paper as it does not isolate what the decline and rebound look like without the massive self-selection effect.

There are some issues with clarity in the write up. In part this likely stems from the word limit for a Letter in Nature, and reading the SI was essential to make sense of the paper. For instance, all through my reading of the main paper, I assumed that variability in how the business was performing would affect an individual's ability to pay, and could explain the result. Only in the SI did I see that this was unlikely to be an issue. At least refer to it in the main paper, even if you expand on it only in the SI. The structure of the repayment, and how it is used to assess the key outcome variable (financial contribution and effort score) also only made sense to me after reading the SI.

Some things were still not clear to me even after reading the SI.

How can a group have a decline in financial contribution and still be selected for a new loan, when only groups that did not fail (which I assume meant groups that "paid in full") are eligible for the new loan? The eligibility criteria for a new loan must be such that even groups that have declining financial contribution across rounds within a cycle have a chance to get a new loan. If not, there wouldn't be any data to document the re-start effect. Clearly, I am missing some key piece of information, but I've looked in both the paper and SI and haven't found the information I need to make sense of how the lender's eligibility criteria for getting another round of loan maps on to the outcome variables in this study.

The lending institution only gives a loan out to a group in the next cycle if they haven't failed to repay. A substantial number of groups do not go on to the next cycle, and this self-selection effect is a key element driving the pattern in Figure 1 of the main paper. It is absolutely critical for interpreting the results to know how this selection occurs. Does the lender solely make the decision? Or do groups voluntarily withdraw from seeking out new loans based on the behavior of their group mates or changes in their business and other undertakings? Is the consequence of not repaying the loan just that they don't qualify for another loan with the lender? Or do they still owe the money to the lender, and have debt collectors showing up at their doors? If the former, then as soon as they see that one person won't be able to repay, shouldn't everyone in the group immediately stop repaying, as it is a total waste of their money to repay?

In summary, I do think this is a worthwhile empirical analysis of an interesting behavioral context. However, because of the nature of the data and the properties of the strategic context, it has limited relevance to the theories of cooperation. The authors may also consider deemphasizing framing the study around the topic of cooperation (if I am correct in my interpretation of the strategic consequences) and potentially emphasizing connections with the literature on judgment&decision-making/irrational behavior/future discounting/heuristics&biases. Also, a lengthier write-up is essential to do justice to this study, and allow a reader to see the merits of the study context.

Referee #3

(Remarks to the Author)

This paper explores the long-term dynamics of cooperation using a field study of group lending in Sierra Leone. Analyzing 47,931 group payments from 7,025 borrowers over a five-year period, the authors reveal a trend where initially high levels of cooperation gradually decrease, only to experience significant rebounds at the start of each new loan cycle. These findings demonstrate the critical role of restarts in temporarily revitalizing cooperative behavior, although they also show that cooperation declines increasingly faster after each reset.

I find the empirical results of the paper compelling. The empirical setting is particularly well-suited for testing repeated social dilemmas in stable groups with periodic restarts. The paper is clearly articulated, and the empirical analysis is robust and methodologically rigorous. I especially appreciate the thoughtfulness of the authors to consider two different measures of human cooperative behavior, the financial contribution rate and the cooperative effort rate. However, I believe the contributions of the paper may not meet the criteria for novelty and broad scientific impact expected in Nature. The reasons for this assessment are detailed as follows:

(1) While the paper successfully extends the laboratory setting used to test human cooperation to a field setting involving real financial stakes, the results primarily provide external validation for well-established patterns observed in laboratory experiments (Andreoni 1988, Cookson 2000). The main observations of the paper do not fundamentally challenge or broaden our existing knowledge from laboratory studies in a way that would be considered groundbreaking.

(2) The authors rule out cumulative and strategic mechanisms as explanations for their data, consistent with the lab experiment literature, and propose plausible behavioral mechanisms such as social norms and temporal boundaries, which I find to be reasonable. Unfortunately, the empirical exercise does not conclusively pinpoint the exact behavioral mechanisms responsible for the observed patterns of cooperation, primarily due to the absence of analysis on the individual-level data. Without a clear identification of these mechanisms, the study stops short of providing a deeper understanding of the causal processes at play, which limits its ability to suggest concrete, targeted interventions or policy recommendations.

Additionally, I believe the authors could enhance their description of the empirical setting. For instance, on page 3, lines 107-108, it is stated that "If a loan is not repaid in full, all members lose access to future credit." Yet, Figures 1 and 2 suggest that some groups that did not fully repay in one cycle were able to participate in subsequent cycles. Clarifying this discrepancy would help readers better comprehend the study's context.

Version 1:

Reviewer comments:

Referee #1

(Remarks to the Author)

As written in my first report, I consider the study as relevant and rich. The point that I raised in the previous version have been satisfactorily addressed. I like in particular that economic environment, which a step level public goods, is now clearer. There are only three minor issues that should be addressed in a further revision.

1. The first paragraph ends with "However, an essential unresolved question is why, even under favorable conditions, cooperation is dynamic and prone to decline." The next paragraph presents several approaches to explain this phenomenon. The authors of these papers might disagree with the view that the question is unresolved. A better formulation could be to replace "unresolved" with "still disputed".
2. Table 2. The regressions only include an interaction term. This makes the interpretation of this variable difficult. It implicitly assumes that there is no round effect in Cycle 0 and no cycle effect in round 0.
3. Figure 4a) It is unclear what the pyramid of under "Lending Staff" means.

Referee #2

(Remarks to the Author)

The authors have improved the paper substantially, and I appreciate the effort they have taken to meaningfully address the concerns raised. Specifically, they have added more details on the ethnographic context and delineated the incentive structure of the group loan payment more clearly; they have clarified the theoretical connection of the study to the literature on cooperation by discussing the loan scheme as a threshold public goods game; they have provided crucial data from interviews they conducted with a subset of the participants that supports the inferences they draw from the aggregate loan repayment data from the lender. These revisions have addressed some of the key concerns I had. I now more clearly see the value of the study, and support publication. I do however have some points I'd like to raise that I think still needs to be addressed.

1. I do not think the term "punctuated" communicates the results of the paper effectively. The pattern being shown is that of "decline and rebound". In evolution, punctuated implies "stasis and change", and we don't see any stasis in cooperation rates here. So it might be confusing to use the term in this way given its usage in biology.

2. The structure of the Introduction is unhelpful for readers to understand the paper. It spends too much real estate on stating the importance of the paper rather than describing the empirical context so that a reader can come to that conclusion themselves. For example, after a terse paragraph 2 describing the empirical context, the paper jumps in paragraph 3 to explain the uniqueness of the empirical context. Without reading the supplementary material a reader does not have enough information to evaluate the claims being made in paragraph 3. The rest of the Introduction describes the results and its broader implications, which again I think jumps the gun and comes at the expense of providing the reader with enough details on how the loan scheme worked.

3. The authors should leverage the empirical findings from their previous paper to strengthen the interpretation of the results of this paper. The previous paper (reference 41) is important background work, and yet it is being referred to very superficially. By describing their previous results from the analysis of the interview data, in addition to the novel analysis of the interview data done in this paper, the authors are in a stronger position to pinpoint (and reassure readers of) the underlying cooperative behavioral mechanisms driving the aggregate data.

4. I am still unclear on the behavioral mechanism being implicated here. The term "behavioral mechanisms" blackboxes all kinds of psychological biases and motivations, and the inability to disambiguate which behavioral mechanisms are driving the decline in cooperation a weakness that should be clearly articulated. The interview data did convince me that there are cooperative motives at play when paying one's own share, and when stepping in to pay someone else's share. But the reason why people don't pay their own share is still unclear. For instance it could be because some people are conditional cooperators and so their motivation to pay their own share decreases when they sense cheap riding going on in their group. On the other hand it could be that everyone slowly becomes lethargic about payment as time passes. This latter scenario does not implicate motives stemming from a conditional cooperative psychology. Rather, it is more in line with what you might see if you signed people up for an exercise program at a gym, and checked in with them every 6 months to assess their health metrics and give them a pep talk. There will likely be a decline in gym attendance over the 6 month period, and a rebound when there's a check-in. These behavioral trends would stem from motives that are not directly (or exclusively) related to cooperation. At the least, I think the paper should more clearly state this point.

Overall, I think this paper contributes valuable and unique empirical data on real world behavior in an important behavioral domain. It would instigate further studies of suitable natural contexts in which long-term cooperation can be analyzed. Such data would greatly benefit a field that has so far been dominated by theoretical models and lab experiments, which although valuable, cannot be relied on to illuminate motivations that may persist or decline over longer time scales.

Referee #3

(Remarks to the Author)

The revised manuscript is a substantial improvement over the previous version. I appreciate the authors' efforts, especially the addition of new figures, the clearer contextual background, the new theoretical framework based on a threshold public goods game, and--most importantly--the inclusion of individual-level survey data, which provides micro-level evidence for the proposed mechanism. The study's value is now much more evident: the long-run decay in cooperation, even after

several stable loan cycles, is something that cannot be captured in lab settings and has important practical implications.

However, I am not fully convinced by the behavioral mechanism proposed by the authors and hope they can further engage with the client interview data. My concern is as follows:

The authors suggest that the key mechanism behind the decline in cooperation is a gradual erosion of cooperative motivation-i.e., a natural shift from other-regarding to more self-interested mindset. But in my view, this account is incomplete and does not fully explain the observed aggregate decline in cooperation. Most of the interviews cited in the supplementary materials do not come from individuals whose own motivation declined (except the final one), but rather from frustrated group members who are angry at others for ceasing to contribute. This suggests to me that the drop in cooperation may have been triggered by a few "bad apples"-individuals who stopped contributing, thereby making it no longer a best response for others to continue contributing toward the threshold.

What remains underexplored is why these "bad apples" emerged in the first place. As the authors note, clients understood the importance of meeting the threshold to maintain access to future loans. Yet some chose to use the loan for personal expenses, stopped repaying, and seemed to disengage entirely, or even burned bridges with their fellow group members. This behavior appears more like a full breakdown of participation than simple cheap-riding. I therefore encourage the authors to revisit the interviews of those who ceased contributing-there may be clues to uncover, such as whether some clients simply no longer needed access to future loans.

Additionally, I am curious whether the client interviews might shed light on the restart effect. While the interviews suggest a shift from other-regarding motives to a more self-interested mindset within a cycle, is there any indication of a reversal in mindset across cycles? Any supporting evidence on this point would be helpful.

Version 2:

Reviewer comments:

Referee #2

(Remarks to the Author)

I appreciate the authors' efforts in revising the paper and clarifying the few topics that were unclear in earlier drafts. This paper will be a useful contribution to the literature. I recommend publication.

Minor suggestions:

Line 98-99 "The results advance our understanding of cooperative dynamics and have significant theoretical and practical implications".

This is a fairly general and therefore vague statement. Would be better if the topic sentence of this paragraph more specifically cued the reader to the specific patterns being discussed in the remainder of the paragraph.

Line 146: "is highly similar". Maybe better to say "has key similarities" or "corresponds"

Referee #3

(Remarks to the Author)

I am satisfied with the authors' responses to my comments from the last round. The newly added interviews and supplementary analyses provide readers with a much clearer understanding of the behavioral mechanisms underlying the decline in cooperation. I appreciate the authors' substantial revision efforts, which now make the paper feel complete: from mechanism to phenomenon.

Response to Reviewers

Nature

Manuscript Title: Punctuated Decline of Human Cooperation

General Comments:

We would like begin by expressing our sincere appreciation for the thoughtful and constructive feedback we have received from the editor and all the reviewers. Every comment has inspired some meaningful change to the manuscript and the result is a substantial improvement in the quality of the paper. We have benefited from the fact that the editor and reviewers offered clear guidance on how to address their concerns. Since the previous version, we have now integrated new data and analysis at the individual level, considerably refined our theoretical framework, identified more precisely the mechanisms underlying the pattern of punctuated decline, and have drawn more novel contributions and practical implications. Thank you for helping us produce this revised manuscript for your consideration.

Given that several of the major updates are relevant to the editor and all the reviewers, we begin by offering a summary of the changes here, before responding to each comment point-by-point.

Summary of Major Revisions:

- 1. *Integration of new data at the individual level:*** *We were able to deepen the empirical analysis with 73 semi-structured in-depth interviews totaling 56 hours with group lending clients and lending institution staff. The data was collected using two-stage cluster random sampling, plus a purposive enhancement. The design provides a representative sample of the larger quantitative dataset, that is further strengthened by secondary interviews to cross-validate responses and enhance internal validity. It provides detailed accounts of the cooperative dynamics through the direct experience of the group members.*

*These interviews were conducted in Sierra Leone during the same time period that the quantitative data were collected, so that any insights into behaviors correspond to contemporaneous payment patterns. We would like to point out that a small portion of these interview data were used previously for an article published in the *American Journal of Sociology*. However, the focus of that article was on a markedly different topic: a cross-sectional view of how social connectedness and spatial proximity affect social punishment. For the purpose of this manuscript, the majority of the interview data remains unused, specifically regarding the behavioral mechanisms underlying the punctuated decline of cooperation over time. Before proceeding with the analysis, we confirmed with the editor that our revision plan was appropriate and would not produce any conflict for potential publication with *Nature*.*

The additional data has proven invaluable for clarifying how the social dilemma is implemented in the field and identifying the behavioral mechanisms that underly the longitudinal patterns. Relevant revisions occur throughout the manuscript, but several new additions are worth highlighting here:

- *Results subsection on behavioral mechanisms;*
- *Methods subsection detailing sampling, data collection, transcription, coding, and analysis process of the qualitative data;*
- *Semi-structured interview protocol (File S1);*
- *Descriptive statistics of the interview sample (Tables S11-S12);*
- *Coded frequency of cooperative and free-riding behavior (Table S13);*
- *Self-reported cooperative motivations (S14);*
- *Codified themes from the interview data in Methods:*
 1. *Cooperative Dilemma Structure and Client Understanding in the Field;*
 2. *Extent and Evidence of Free-Riding;*
 3. *Self-Reported Motivations for Cooperation;*
 4. *Changes in Behavior over Time;*
- *Extensive supporting direct quotes in Methods.*

We would also like to clarify what type of individual data is not available to us and why. For the long-term quantitative data set, cooperative behavior is only recorded by the lending institution at the group level, i.e. they do not record how much each individual client contributed. There is an important reason why the institution (and many institutions like it) intentionally does not track individual contributions: the group lending model is based on the idea that the group members have a shared fate and will not be held individually accountable by the lending institution. Tracking individual contributions would undermine this cooperative model and make the collective outcomes data questionable as a true measure of cooperation. As our primary interest is understanding how and why collective outcomes change over time in a real-world social dilemma, reliable group outcome measures complemented with a random sample of individual-level data is valuable approach for maintaining integrity of the collective measures while exploring the underlying mechanisms.

2. ***Refined theoretical framework:*** *The guidance of the reviewers and editor on this point has been particularly constructive. We have now articulated the theoretical framework more precisely, conceptualizing the group lending structure as a threshold social dilemma, i.e. the provision of the collective good is non-linear and occurs only when aggregate contributions reach a specified level. The theoretical and experimental literatures on threshold dilemmas are widely interdisciplinary and this framework has proven very useful in clarifying the strategic incentives, the nature of the cooperative conflict, and the multiple equilibria that result. This has been particularly useful in interpreting our data because it serves as the appropriate model for interpreting*

deviations from standard rational behavior. The theoretical framework and its key elements are described concisely in the main text and elaborated upon in the supplementary materials.

- 3. Identification of mechanisms and novel contributions:** *The refined theoretical framework allows us to more precisely define behavior that is compatible with rational strategy versus deviations from it. By integrating the in-depth interviews, we can now show that the pattern of punctuated decline is driven by behavioral deviations based on systematic reductions in cooperative motivation and effort over time. The changes in motivation are substantiated with direct answers from interviewees, as well as behavior embedded in cross-validated descriptions of cooperation and defection within the group.*

The causal mechanisms identified in this study are counter to prevalent explanations associating cooperative decline with rational or increasingly rational behavior. We suggest that predictable changes in psychological states may play a larger role than currently understood and should be considered as key causal drivers of cooperative decline. Rather than a stable disposition towards cooperation or defection, we find that cooperative motivation and effort naturally decay, intensifying the tendency to cheap ride over time.

The mechanisms of behavioral decline have detrimental consequences because they tend to push groups into a space between equilibria i.e. between zero-contribution and reaching the collective threshold. This has negative consequences for both the individual and the group. For example, group members partially repaying their loan but not achieving access to future credit.

Importantly, it is now clearer in theoretical terms that the risk of behavioral decline is not specific to group lending, but rather arises from a natural tension in the theoretical framework. Incorporating behavioral dynamics into models of collective action may be essential for explaining why cooperation falters even under structurally favorable conditions, with direct practical applications for institutions. We believe that this dynamic is theoretically compelling and applicable to a broad class of social dilemmas that occur frequently in natural settings.

- 4. Revision of all key figures:** *With the goal of data transparency and improved communication, we have substantially revised all the figures in the main text.*
- 5. Additional empirical context:** *We have added descriptions of subsequent loan cycle eligibility; informal and formal loan enforcement processes; and the extent and implications of free-riding behavior. Thanks to your feedback we have realized which additional details were needed for the reader to appropriately interpret the results.*
- 6. New subsection on alternative explanations:** *Given the complex nature of the field data, we found it useful to succinctly organize all the evidence regarding alternative*

explanations in a single subsection. The result is a more coherent evidence-base for our leading explanation centered on behavioral mechanisms.

We greatly appreciate that we were given the opportunity to improve the manuscript. We hope you find the revision to be a compelling analysis of this important topic. Below is a point-by-point response. The editor's and reviewers' original comments are left-justified and non-italic. Our interleaved responses are indented and italic.

Referees' comments:

Referee #1 (Remarks to the Author):

This paper present fascinating field data from a microfinance institution in Sierra Leone. This institution gives loans to groups of people who are mutually responsible to repay the loan. Thus, repaying provides a public good for the group, and the data allows to investigate how cooperation develops over time. Two observations are made. First, cooperation declines. It declines within a credit cycle, and the decline increases across the cycles. Second, there are strong restart effects. After each credit cycle the cooperation jumps back to almost full cooperation. This pattern resembles restart effect in public good experiments. However, there are important differences to the lab studies. The main difference is that there are strong reputation effects. Not repaying spoils your reputation towards the group members and beyond. In addition, if the group does not repay, the group members will not receive a loan again. In particular, this last point makes turns the public goods game into a threshold public goods game, which has a very different game theoretic structure than the standard public goods game, in which there is a dominant strategy in payoffs.

Your comments on theory, here and below, have been extremely helpful and has been instrumental in guiding our revisions overall. We agree that the group lending context has more the characteristic of a threshold social dilemma than a standard linear model. We have accordingly refined our theoretical framework and discuss the incentives of this class of games, relevant equilibria, and how they map to our setting. This has been particularly helpful for interpreting individual level behavior (described in the client interviews) with an appropriate benchmark for strategic behavior in this setting.

The relevant changes related to the refined theory are throughout the paper, particularly: framing in the abstract and introduction; a concise theoretical framework at the beginning of the Results section; and throughout the Discussion and Methods sections when interpreting individual-level behavior. We also include a new section in the Supplementary Materials entitled Theoretical Background that provides more detail regarding relevant theoretical and experimental work on threshold public goods in relation to our study.

The paper is generally well written, the study interesting and the data analysis convincing.

Thank you very much for the positive feedback.

In the following, I first raise my main concerns and then mention some further points

Main concerns

It is unclear to me how exactly the threat works. I guess that loans are possible again when the loan is repaid, even when the repayment is delayed; and what is measured is repayment in time. A better description of this mechanism is necessary in order to understand the incentives in case someone is in trouble or unwilling to repay.

Thank you for pointing this out. We appreciate that more details regarding the practical process of group repayment and loan renewals are necessary to fully interpret the significance of the group behavior. We have now added specifics regarding eligibility criteria for loan renewal and the (informal and formal) processes of loan enforcement, i.e. the threat and consequences if someone does not cooperate. For convenience, we provide a short summary here (we include the full details in the Methods section).

Eligibility Criteria: Full loan repayment in the previous cycle is required for approval of a new loan, but timeliness of monthly payments throughout the whole previous loan cycle also influences the decision. The decisions are made jointly by the group's Loan Officer and Loan Portfolio Manager. Note that the lending institution only approves a subsequent loan cycle at the group level, i.e. it does not differentiate how much each individual contributed to repayment.

Enforcement Process: The process operates in two phases: (1) internal enforcement of active loans and (2) institutional enforcement of delinquent loans. During the active phase, group members are responsible for enforcing on each other. Members use a variety of positive and negative enforcement mechanisms, such as encouragement, social pressure, embarrassment, and ostracism. During this phase, the collective good is still achievable, i.e. access to future credit. After approximately 30 days of delayed payment, enforcement gradually shifts to the institutional phase. In this phase, the possibility of qualifying for a subsequent loan rapidly declines. The loan is classified as inactive, and the institution initiates formal recovery procedures – typically involving debt collectors and legal threats. These efforts generally continue for at least a year until the group repays or the organization officially records the loan as a write-off and ceases efforts at collection. We intentionally account for these two different phases in our quantitative

modelling approach. We define cooperative outcome measures using a 30-day window to capture meaningful behavior attributable to internal group dynamics, distinct from institutional debt recovery efforts.

Excludability: One of the informal tools available to the group is the exclusion of a member from the next loan cycle. The group's choice is often based on the perceived motives for defection, frequency, attitude, and process of communication. Some groups will even decide to allow one defector to remain in the group and another to be excluded. In the Methods section we provide extended interview data regarding different informal responses to non-cooperative behavior (including expulsion from the group) and summary tables of such codified behavior.

The authors explain in detail that failed repayment is not due to inability to repay but due to lack of cooperation. This part was particularly puzzling to me because I wondered how the defectors will be treated by the other group members and the microfinance institution. It is astonishing that “voluntary” free riding makes sense. However, if people notice that the consequences of free riding are not that harsh, learning could lead to a decline in cooperation.

We have revised the text to clarify how free-riding occurs in practice. As you have rightly pointed out, it would seem strange that full free-riding occurs in this context. That would imply that a group member receives their share of the loan, contributes nothing towards repayment, and is allowed to receive another share of the following loan disbursement. Drawing on our interviews, this does not happen in practice. However, partial free-riding happens frequently. That is, someone contributes less than their fair share of repayment, other members compensate for the partial free-rider, and the partial free-rider retains full equal access to the loan disbursement in the following cycle. We estimate the frequency of partial free-riding behavior to occur in approximately 91.4% of groups based on our representative sample of interviewed groups (see Table S13).

This is consistent with the appropriate theoretical framework based on a threshold social dilemma. If some members are able to compensate for defectors, a distributional conflict arises. The individual incentive is to pay as little as possible (cheap ride) and still maintain access to the collective good. This tension is fundamental to the progression of the cooperative pattern over time. The manuscript text has now been updated throughout to reflect this. A significant portion of the interview data focuses on providing the empirical frequency of different types of free-riding behavior, self-reported motivations for paying for oneself versus compensating for another member, and the consequences of partial free-riding. See the Methods for codified themes from the interview data and Tables S13-S14 for frequencies of coded behaviors.

Additionally, we have organized the evidence regarding learning and changes in financial ability into a single subsection of the Results titled “Alternative Explanations.” In this section we summarize the empirical evidence as to why these alternative mechanisms are not likely to be the key drivers of the observed trends in our setting.

I miss theory; not necessarily in the form of a fully elaborated model but in the form of an analysis of the incentives. As mentioned above, the game has more the characteristic of a step level public goods game. This means that players have an incentive to offset or compensate the other players' contributions. In this respect, the question might less be why people reduce cooperation but how people punish, a question addresses by the authors in Sabin & Reed-Tsochas (2020).

In response to this and your earlier comments, we have expanded the theoretical framing in the manuscript, with particular attention to the threshold nature of the social dilemma in group lending. As you rightly note, group lending involves a threshold dilemma in which members may compensate for one another, which creates the potential for distributional conflict. Given the limited resources, when members attempt to cheap ride, it raises questions about how others will respond and the behavioral implications for the group's long-term outcomes. Our interview data indicate that this distributional conflict is a salient concern in borrowers' decision-making and reveal considerable heterogeneity in how members respond to it. The empirical evidence supports the presence of distributional conflict in our setting and suggests that individuals' efforts to navigate this dilemma are central to the observed decline and rebound in cooperation rates.

Further points

Is there data on the individual level? It would be interesting to analyze the individual observation, in particular how people respond to underpayment.

We have now incorporated the individual-level data into the manuscript as described above. There is significant information regarding how people respond to underpayment. We believe that the combined use of quantitative data and interview data has strengthened our ability to understand the mechanisms and motivations behind the punctuated decline of cooperation in our setting.

The first sentence in the abstract, “[d]espite the tendency of human cooperation to decay over time, it can be punctuated by 13 substantial rebounds in cooperative behavior” raises the expectation that the punctuations occur spontaneously. However, the restart time points are given by the economic environment.

Thank you for pointing out the ambiguity in that sentence. We have now edited the abstract and the relevant parts of the main text to avoid creating the expectation that punctuations occur spontaneously. Our new abstract reads: “Sharp rebounds occur when loans are restarted and clients resensitized to their cooperative responsibilities....”

Reference

Sabin, Nicholas, and Felix Reed-Tsochas. 2020. 'Able but Unwilling to Enforce: Cooperative Dilemmas in Group Lending', *American Journal of Sociology*, 125: 1602-67.

Referee #2 (Remarks to the Author):

The study documents longitudinal patterns in cooperation over a 5-year period in the context of group-lending in Sierra Leone, using data from the micro-finance organization that provided these loans. Unlike a lot of empirical studies of cooperation, the data for this study come from a real-life context. This feature is important because uncertainties about the external validity of laboratory economic games severely limits what inferences we can draw about real-world behavior from experiments. Moreover, unlike most naturalistic settings of cooperation, this setting offers some of the useful features of consistency and control that we seek in experiments: the economic incentives are the same across the loan cycles; the composition of each group stays the same; there is a well-defined time cycle that marks when a round of cooperation starts and ends; there are objective measurable outcomes (amount of loan repaid each month, timeliness of making partial or complete payments) to keep track of the level of cooperation. The data span 5 years, which is exceptionally long, given that most of the empirical studies of cooperation in repeated games is from experiments that last a few hours.

The authors argue that the results show a robust pattern that reveals something about the psychological mechanisms of cooperation: Cooperation deteriorates within rounds of a loan cycle, but also rebounds in each new loan cycle (even though group composition has not changed). The rate of decay increases across cycles. While other studies have documented declining cooperation over time, this study has a couple of neat additional insights. First, because cooperation rebounds in each new cycle, despite no change in group composition or the incentive structure, it is clear that the decline of cooperation is not due to people learning about the game structure over time. Second, the authors argue that changes in ability to cooperate is unlikely to explain the behavior, as people had a reserve account in which there was enough money kept aside to repay the loan, in case the business they were operating did poorly or other hardships arose.

Thank you very much for the clear summary of the original manuscript and for appreciating several of the unique features of this dataset.

While it is a useful study documenting an interesting finding, I feel that the paper doesn't have the theoretical import that I was hoping to see from a study of cooperation in consideration for publication at Nature.

Thank you for pointing out that you were hoping for more theoretical insight. Thanks to your feedback and the feedback of the other reviewers and editor, we have now substantially refined our theoretical framework and have incorporated the analysis of new individual-level data. The process of jointly refining theory and analyzing a new data source has led to what we believe are more insightful and generalizable theoretical contributions to the understanding of human cooperation, as we describe in detail in our responses to your comments below.

First, as far as I can tell, this is not the typical kind of dilemma that characterizes evolutionary models of n-person cooperation. In evolutionary models (and in most laboratory games that attempt to test these models) the game is a public-goods game — individual payoff is maximized

by not cooperating, and group payoff is maximized if everyone in the group cooperates. In the current study, individuals maximize their payoffs by cooperating. Receiving a loan is financially beneficial; therefore, repaying the loan so that you get a loan again in the next cycle is in the individual's best interest. The fact that individuals are failing to repay therefore is not because they are "defecting". Rather, it is something else about the psychology that is leading them to make a decision that is not in their longterm financial interest. The group-lending schema is only helping to bolster against the decay in individuals' irrational behavior, by making it beneficial for group mates to pressure their peers to stay on course.

This comment, combined with the theoretical points made by Reviewer #1, has been extremely useful because it has helped us to clarify our thinking about the structure of our setting and has led us to be more explicit in our manuscript regarding our theoretical framework and the individual incentives participants face. We now make clear that group lending can be better characterized as a threshold public good, whereby provision of the collective good is non-linear and occurs only when aggregate contributions reach a specified level. As you rightly pointed out, this implies that providing for the good — i.e., repaying the loan as a group — is an equilibrium, and a Pareto superior one relative to the zero-contribution equilibrium. This is in contrast to the standard linear public goods model, in which zero contribution is the only equilibrium.

However, a threshold public good structure does not imply that individual payoff is maximized through cooperation. An individual maximizes her payoff by contributing as little as possible of her own private resources while still achieving provision, a behavior that the literature has referred to as partial free riding or cheap riding. The possibility of cheap riding leads to a conflict between opportunistic individual behavior and collective interest, and is what makes both the threshold game in general and group lending in particular a cooperative dilemma. You rightly point out below that this is relevant in contexts where group members can compensate for laggards. We now spend significant space in the main text and the supplementary materials clarifying this incentive structure and giving evidence from our interview data that borrowers in our setting often engage in this attempt to pay less than their fair share while still achieving provision and maintaining access to future credit.

Clarifying the structure of our setting in this way has also led us to better describe what we think are important, unique insights that result from studying a long-term, threshold cooperative dilemma as we do. Unlike the linear public goods model in which provision is continuous in anyone's contribution, a feature of a threshold cooperative dilemma is a clear demarcation between an individual's contribution towards his or her own share of the collective project and his or her contribution towards someone else's share. Our evidence indicates that these two aspects of cooperative behavior are driven by systematically different motivations and have consequences on the likelihood of behavioral decay. We believe these insights may prove to be widely applicable in the study of cooperation and its decline over the long term, especially since distributional conflict is not specific to group lending but rather is inherent in the threshold dilemma.

The only way that "non-repayment" is "defection" is if the other group mates step in to pay for the laggard in their group. This could potentially happen, but we have absolutely no information about whether this was how people behaved. The only data the authors have is from what the lending organization documented as the group's payment. In that sense, the data is too thin (despite its amazing time-depth).

Thank you for this important comment. It has helped us appreciate the value of incorporating additional empirical evidence into the manuscript to allow readers to fully understand how the cooperative dilemma operates in practice. In the paper, we now draw on the interview data to address your comment and to provide answers to related practical questions that the reader may have. We summarize content here that is now detailed in the paper:

(1) Are borrowers held collectively responsible by the lending institution in practice and do clients understand the consequences? The interview data confirms that groups are indeed held collectively responsible and that the clients do understand this from the outset. See Methods "Theme 1: Cooperative Dilemma Structure and Client Understanding in the Field" for detailed evidence.

(2) Do clients defect, leaving their share to be compensated by other members? Full free-riding is rare, but partial free-riding, or "cheap riding," is common. We use the representative sample of groups to provide an estimate of the frequency of free-riding behavior. More than 90% of groups interviewed mentioned at least one instance of partial free-riding. Please see Table S13 for coded behaviors and Methods "Theme 2: Extent and Evidence of Free-riding" for analysis and direct quotes.

(3) Do group members compensate for partial free-riders in their group? Yes, but it varies across groups and circumstances. Our interview data shows that group members are willing to compensate for others' lack of payment, hypothetically and empirically, typically if they deem that lack of payment is justified. See Table S13 and Methods "Theme 2: Extent and Evidence of Free-Riding" for subthemes on heterogeneity in compensation behavior and supporting examples.

(4) Is the distributional conflict — i.e., the tension between the collective interest of renewing the loan and the individual incentive to contribute as little as possible of one's own private resources while still achieving provision — central to the group behavior? The interview evidence suggests that this issue is front-of-mind for clients and how they resolve or fail to resolve the conflict shapes group outcomes over time. The interview data indicates that cheap-riding is common and that groups monitor such behavior. See Figure 4 and Table S14 for self-reported motivations in regard to the distributional conflict, and see Methods "Theme 3: Self-Reported Motivations for Cooperation" and "Theme 4: Changes in Behavior over Time" for analysis and direct quotes.

(5) Do group members enforce on each other? They may use several informal practices, such as social pressure, public embarrassment, and ostracism, dependent on the specific

group. Their relevance is summarized in the current manuscript and is consistent with the literature on group lending.

The authors argue that the temporal pattern of decay can be attributed to psychological mechanisms, and not learning or strategic motivations. This is true, but there is no contextual data on what happened within these groups that shed light on what these psychological mechanisms are. Obviously, getting this information would not be feasible, given that the study is based on an existing dataset created by the lending organization. The authors allude to conditional cooperation as a mechanism for the decay, but is it? An alternative is that the decline is due to extrinsic factors that affect individuals within a group with some probability, and affects individuals' ability to pay or motivation to continue to run his or her business. The accumulation of these events within a cycle to individuals within a group could explain the decline.

To rule out this and other possibilities, we need complementary interview or survey data at least from a subset of these groups on why people stopped repaying, and how the decision of others in their group affected their motivation to repay. Group-lending assumes that ostracism and sanctioning may boost loan compliance, but we have no idea if there was any social sanctioning within the study groups. As a thought experiment, if instead of group-lending, we are looking at a similar dataset in which lenders give out loans to individuals, how different would the loan compliance behavior look? It is possible that the same pattern we see here will be seen in individual-lending too. So, it is not clear if these patterns are stemming from group-dynamics of cooperation, or from individuals making not-so-great choices for themselves.

Thank you for encouraging us to look for additional data to shed light on the underlying mechanisms. One of our main revisions to the paper has been to add individual-level depth to the study by incorporating interview data that can speak to mechanisms directly. We now analyze data from 56 hours of interviews to group members and staff that we conducted contemporaneously with the empirical data collection (the Methods section gives details on the interview sampling design and the interview protocol). The combined quantitative and qualitative analysis supports that the observed pattern of punctuated decline is driven by systematic changes in borrowers' cooperative motivation and effort over time.

Your comments on the potential role of individually irrational behavior, combined with the more refined strategic framework, have been very helpful in refining the analysis of the individual-level data. Unlike in standard linear social dilemmas, thresholds produce a partial alignment between individual and collective interests: the maintenance of the good provides personal benefit, yet full cooperation fails to maximize individual payoffs. The empirical data show that as motivation to cooperate decays, groups increasingly struggle to reach the collective threshold every month. In line with your comments, we examined the interview data through a key lens: does the decline in contributions result from a decrease in other-regarding interests or a decrease in long-term self-regarding interests? We found empirical evidence for multiple motivations, with examples of inconsistent regard for one's own self-interests and decreased regard for those of the

group. However, the interview data indicated that other-regarding interests were significantly more central to the clients' repayment choices. Figure 4c summarizes the frequency of different categories of self-reported motivations for cooperation. We found in the interviews that group members strongly distinguish between paying for what they view as their own share versus paying for another member's share. While paying one's own share was associated with a diversity of motivations, including economic self-interest, compensating for another member's share drew on a much narrower set of motivations, almost exclusively prosocial- or norm-based. Notably, we did not limit clients to reporting one motivation for a given behavior. In the semi-structured interview format, they were free to report multiple motivations as they saw relevant.

In general, we find that the threshold structure gives rise to other-regarding interests playing a central role in the decision making of individual members. The inclusion of more explicit theory in the updated manuscript clarifies that the tradeoffs faced here are not unique to group lending, but arise from the theoretical framework of threshold dilemmas, characterized by the distributional conflict and multiple equilibria. Within this generalizable framework, we find it empirically compelling to observe what participants focus on and how they choose to navigate the dilemma.

In addition, we strongly agree with your comments regarding the importance of accounting for extrinsic factors that could systematically affect a borrower's ability to pay. This topic has been a key element woven throughout our analysis, but your comment convinced us that it would be valuable to bring together the primary points of evidence in a single place in the main text. We have now created a new subsection of the Results to summarize the counter evidence for the most relevant alternative explanations.

Here we provide the section of text relevant to extrinsic factors: "The final class of alternative explanations, specific to group lending, is based on systematic change in the borrowers' financial ability to pay. Within loan cycles, the decline could be driven by a progressive reduction in available cash or the gradual accumulation of exogenous shocks to the members' businesses. Across loan cycles, it is plausible that as loan amounts increase, it becomes more difficult to repay. Though clients are often faced with economic constraints, four different types of evidence indicate that they are not the cause of the longitudinal patterns. (1) Organizational policies: A group loan is rescheduled by the microfinance institution if the loan officers judge the group to have genuinely encountered an inability to repay. In addition, the microfinance institution requires that groups keep a portion of their original loan in a savings account to use in case of inability to pay. This is rarely accessed by defaulting groups, indicating unwillingness rather than inability to pay. (2) Client interviews: Groups are often able to overcome genuine financial difficulty, e.g. a house fire (Client Interview G22.C43), unexpected medical bills (Client Interview G9.C16), slow business (Client Interviews G5.C8, G1.C1), or theft (Client Interview G7.C11), but consistently struggle to repay when a member's behavior becomes uncooperative (e.g. Client Interviews G12.C22; G27.C54; G35.C64). (3) Statistical analysis and controls: Decline within a loan cycle is not monotonic. Cooperation often increases in the final rounds of a loan cycle (Table S10), indicating behavioral motivations rather than a strict decline in financial ability over rounds.

Statistical models include group-level controls for variation in ability to recover from economic shocks as group fixed-effects or as covariates (e.g. average monthly sales, business diversity, geographic region), temporal controls for seasonal weather shocks and other idiosyncratic adverse events by year, and cycle controls for progressive changes in borrowers' financial situation in terms of loan amount and duration (Tables 1-2, S3-S7,S9). (4) Alternative variable construction: The Cooperative Effort Rate as an alternative outcome to financial contribution, applied to the entire quantitative dataset, minimizes the role of financial liquidity by considering timeliness of any partial payment, no matter how small. The longitudinal patterns and statistical tests largely replicate across outcome measures (Figures 1-3, S1-S2; Tables 1-2, S4-S7, S9-S10).” We hope the text now effectively communicates the multiple types of evidence that suggest extrinsic factors and systematic changes in financial ability over time are not the primary drivers of behavior in our setting.

Lastly, the pattern of decline & rebound is much reduced when looking at groups that go on to another loan cycle. This means that most of the pronounced effects of Figure 1 are driven by the self-selection effect. Many groups fail to sustain cooperation and are removed from the mix. The rebound in Fig 1 is possibly from groups that ended high in the previous cycle, starting similarly high in the next cycle but without the failing groups pulling down the average financial contribution rate on the graph. The authors have done the appropriate analysis to examine what is left after the self-selection effect, but this is shown later. Fig 1 should not be the main figure in the paper as it does not isolate what the decline and rebound look like without the massive self-selection effect.

Thank you for the constructive feedback on Figure 1. Your comment has prompted us to look for ways to communicate the key results as transparently and effectively as possible. After extensive discussion and experimentation, we have arrived at four substantially revised figures for the main text. Here we offer a few comments regarding the rationale behind the specific figures.

On Figure 1: We agree with you that the previous version of Figure 1 could be misinterpreted as the role of selective attrition could be easily overlooked. We have now redesigned the figure to highlight the role of selective attrition in the overall dataset. You will now find the aggregate cooperative patterns overlaid on the full dataset with each joint liability group color-coded based on whether they continue to the next loan cycle or not. Plotting the underlying data also now conveys that the total sample shrinks across cycles. The surrounding text also more directly addresses the role of selective attrition in the dataset and analysis. To help clarify differences between Figure 1 and other figures based on data subsets, the lines connecting the average trend from one cycle to the next have been removed, so as to avoid giving the impression that there is a single sample that continues unchanged across cycles. One reason we think it is important to start with a figure that presents the full sample rather than a subsample without attrition is that the figure can show the pattern and extent of the collapse of groups, which is a key feature of the decline of cooperation over time.

On Figure 2: This figure shows cooperation rates removing the effect of selective attrition, by keeping the sample fixed in any subfigure. You rightly noticed that the pattern of decline and rebound is reduced when restricting the sample to groups that continue. However, it is important to point out that this is to be expected given that this analysis removes low-performing groups. In light of this, it is striking that even when restricting the sample to highly successful groups, we still observe a pattern of decline and rebound, which is highly statistically significant. New additions to the manuscript provide detail on the eligibility criteria for subsequent loans and clarify how it relates to the subset analysis.

On Figure 3: We have further refined our analysis of the restart effect and the figure presenting the results. The figure shows sequential restart effects on cooperation rates for continuing groups, by comparing average round-over-round change in the rounds leading up to a restart to the change in the restart round, holding the samples fixed for only groups that participate in that restart. The rebound of cooperation is large and statistically significant, as is evident by comparing average change within the cycle (blue and green bars) to average restart change (red bars) for the same groups. Supporting regression results in Tables S4-S7.

Figure 4 is entirely new and is based on the interview data. It provides a depiction of the sampling strategy, descriptive statistics of the interview sample at the group- and individual-level, and a summary of self-reported cooperative motivations in regard to paying one's own share versus paying another member's share.

There are some issues with clarity in the write up. In part this likely stems from the word limit for a Letter in Nature, and reading the SI was essential to make sense of the paper. For instance, all through my reading of the main paper, I assumed that variability in how the business was performing would affect an individual's ability to pay, and could explain the result. Only in the SI did I see that this was unlikely to be an issue. At least refer to it in the main paper, even if you expand on it only in the SI. The structure of the repayment, and how it is used to assess the key outcome variable (financial contribution and effort score) also only made sense to me after reading the SI.

Thank you for pointing out that this topic needs to be better communicated. We have made several revisions related to this issue. In the main text we have clarified our motivation for the construction of the cooperation rates regarding the issue of financial ability to pay. As we have mentioned earlier, we have also added a specific subsection to the results, "Alternative Explanations," in which we concisely explain why systematic variation in ability to pay is not a likely driver of the results.

Some things were still not clear to me even after reading the SI.

How can a group have a decline in financial contribution and still be selected for a new loan,

when only groups that did not fail (which I assume meant groups that “paid in full”) are eligible for the new loan? The eligibility criteria for a new loan must be such that even groups that have declining financial contribution across rounds within a cycle have a chance to get a new loan. If not, there wouldn’t be any data to document the re-start effect. Clearly, I am missing some key piece of information, but I’ve looked in both the paper and SI and haven’t found the information I need to make sense of how the lender's eligibility criteria for getting another round of loan maps on to the outcome variables in this study.

The lending institution only gives a loan out to a group in the next cycle if they haven’t failed to repay. A substantial number of groups do not go on to the next cycle, and this self-selection effect is a key element driving the pattern in Figure 1 of the main paper. It is absolutely critical for interpreting the results to know how this selection occurs. Does the lender solely make the decision? Or do groups voluntarily withdraw from seeking out new loans based on the behavior of their group mates or changes in their business and other undertakings? Is the consequence of not repaying the loan just that they don’t qualify for another loan with the lender? Or do they still owe the money to the lender, and have debt collectors showing up at their doors? If the former, then as soon as they see that one person won’t be able to repay, shouldn’t everyone in the group immediately stop repaying, as it is a total waste of their money to repay?

Thank you for pointing out the need for more information on the eligibility criteria for subsequent loans. It is an excellent point that all 3 reviewers noticed. We have now added detail to the manuscript regarding the eligibility criteria, the informal and formal loan enforcement process, and the consequences of nonrepayment.

“Eligibility for a subsequent loan cycle depends on a group’s repayment history in the previous cycle. Full repayment is required, but timeliness of payments throughout the whole loan cycle also influences the decision. The institution incentivizes better group repayment with greater potential increases in the subsequent loan amounts. Groups that repay in full and on time receive the standard maximum loan increase for the next cycle. If the loan is paid in full, but with delayed payment(s), the group may be assigned a lesser loan increase, no loan increase, or no loan renewal, based on the frequency and severity of the delayed payment(s). The outcome is the same for all group members. Decisions regarding loan renewal and amount are made jointly by the group’s Loan Officer and Loan Portfolio Manager. Variation in subsequent loan amounts implies that a single provision point for the collective good is a theoretical simplification of the full set of potential group outcomes. However, the core distributional conflict and individual incentive to contribute as little as possible while reaching each provision point remains consistent.”

“Group loan enforcement operates in two phases: (1) internal enforcement of active loans and (2) institutional enforcement of delinquent loans. During the active phase, group members are responsible for enforcing on each other. Members use a variety of positive and negative enforcement mechanisms, such as encouragement, social pressure, embarrassment, and ostracism. During this phase, the collective good is still achievable,

i.e. access to future credit. After approximately 30 days of delayed payment, enforcement gradually shifts to the institutional phase. In this phase, the possibility of qualifying for a subsequent loan rapidly declines. The loan is classified as inactive, and the institution initiates formal recovery procedures – typically involving debt collectors and legal threats. These efforts generally continue for at least a year until the group repays or the organization officially records the loan as a write-off and ceases efforts at collection. In the Modelling section below, we define cooperative outcome measures using a 30-day window to capture meaningful behavior attributable to internal group dynamics, distinct from institutional debt recovery efforts.”

You have also emphasized the importance of attrition in the dataset. We fully agree with you regarding its importance and, since the outset, we have intentionally crafted our analysis to deal with this issue. But we fully appreciate your point that the reader needs more detail on this topic to be confident in interpreting the results for themselves. We have now added Table S2: Group continuation status and sources of attrition across cycles (which we refer readers to in the Main Text). The table provides a full breakdown of sources of attrition and the percentage of groups falling into each category by loan cycle. In short, groups may not continue in the dataset because they choose not to continue after successful repayment, they were not approved for a subsequent loan because of poor performance, or their potential next loan falls outside the end of the data collection window. The important takeaway from the table is that group attrition is not random and is more often a result of poor performance in the previous loan cycle. This is why we have approached the analysis very carefully to make sure that selective attrition is not driving the results. We test all the principal results with both the overall dataset and fixed subsets of only continuing groups with no attrition. When effects are estimated in the paper, for example, the size of the restart effect, we only use fixed subsets of continuing groups. This approach is conservative and most likely underestimates the true effect, as low-performing groups are the groups whose cooperation rates have the most room to rebound and are the groups not allowed to restart.

Your last comment above asked if one person is not going to pay, shouldn't everyone in the group immediately stop repaying? This would be true if members could not compensate for each other, in which case, the threshold dilemma becomes primarily an issue of coordination among members and the avoidance of wasted contributions. However, members can compensate for each other in group lending which produces the distributional conflict and the incentive to cheap ride. This information has been added to the manuscript in the main text and detailed in the Methods to clarify the nature of the principal conflict.

In summary, I do think this is a worthwhile empirical analysis of an interesting behavioral context. However, because of the nature of the data and the properties of the strategic context, it has limited relevance to the theories of cooperation. The authors may also consider deemphasizing framing the study around the topic of cooperation (if I am correct in my interpretation of the strategic consequences) and potentially emphasizing connections with the

literature on judgment&decision-making/irrational behavior/future discounting/heuristics&biases.

We have greatly valued your thoughtful and insightful comments. We hope that in the revised manuscript we have now communicated why cooperative conflict is central to group lending and why it offers an unusually rich source of long-term cooperative behavior in a natural setting. Moreover, we appreciate how this revision has greatly improved the empirical depth and clarity of the analysis.

Also, a lengthier write-up is essential to do justice to this study, and allow a reader to see the merits of the study context.

We hope you agree that the revised paper effectively communicates the context and key findings in a concise format. In the revised manuscript our approach has been to “signpost” the significance of key issues early on in the main text (as some topics were introduced too late in the original manuscript) and direct the reader to the Methods and Supplementary Materials for the extended detail. We find the format worthwhile given its potential to reach a broad, interdisciplinary readership that may benefit from the findings of this study.

Referee #3 (Remarks to the Author):

This paper explores the long-term dynamics of cooperation using a field study of group lending in Sierra Leone. Analyzing 47,931 group payments from 7,025 borrowers over a five-year period, the authors reveal a trend where initially high levels of cooperation gradually decrease, only to experience significant rebounds at the start of each new loan cycle. These findings demonstrate the critical role of restarts in temporarily revitalizing cooperative behavior, although they also show that cooperation declines increasingly faster after each reset.

I find the empirical results of the paper compelling. The empirical setting is particularly well-suited for testing repeated social dilemmas in stable groups with periodic restarts. The paper is clearly articulated, and the empirical analysis is robust and methodologically rigorous. I especially appreciate the thoughtfulness of the authors to consider two different measures of human cooperative behavior, the financial contribution rate and the cooperative effort rate.

Thank you for the positive feedback. We are glad to hear the features of the manuscript that you found particularly compelling.

However, I believe the contributions of the paper may not meet the criteria for novelty and broad scientific impact expected in Nature. The reasons for this assessment are detailed as follows:

(1) While the paper successfully extends the laboratory setting used to test human cooperation to a field setting involving real financial stakes, the results primarily provide external validation for well-established patterns observed in laboratory experiments (Andreoni 1988, Cookson 2000). The main observations of the paper do not fundamentally challenge or broaden our existing knowledge from laboratory studies in a way that would be considered groundbreaking.

We hope you agree that with the addition of new individual-level interview data, a more specific theoretical framework, and deeper empirical analysis, the revised manuscript now provides an unusual combination of quantitative and qualitative data and important theoretical insights that meaningfully advance our understanding of cooperation in several respects. We have articulated the updated empirical and theoretical contributions in the General Comments above. We wish to add a few more comments here.

We agree that the observed pattern of decline and rebound is consistent with a broad body of laboratory-based research. We hope our revision has more clearly articulated that advancing the study of cooperation into these different stakes and timescales is an important contribution in itself, but just one of the study's contributions.

Regarding external validity, we believe it is not evident from laboratory work that those patterns should be expected to emerge in a setting with so vastly different stakes and timescales as ours. The long-term field setting also allows novel insights that would be difficult to uncover in the laboratory. By examining the long timescales that we do, in combination with the new interview data, we have been able to provide evidence indicating that behavioral motivations underlying cooperation are not necessarily simply "fixed traits" or "fixed types," as they are typically conceptualized in laboratory-based

research. Rather, motivations are prone to natural decay or change over time, and this decay or change is in itself a source of the decline of cooperation. Groups are at risk of such decay even after months or years of successful interaction. These observations and insights are underappreciated or missed in the theories and knowledge that emerge from laboratory studies.

Another key contribution of the updated manuscript is to demonstrate the value of studying the problem of cooperation through the lens of a threshold dilemma rather than the more standard prisoner's dilemma or linear public goods game. The threshold dilemma structure accentuates specific behavioral responses that we can empirically show. Unlike the linear public goods model, in which provision is continuous in anyone's contribution, a feature of a threshold cooperative dilemma is a clear demarcation between an individual's contribution towards his or her own share of the collective project and his or her contribution towards someone else's share. The interview data indicate that this a fundamental distinction for borrowers and it strongly shapes their decision making. Figure 4 provides a summary of self-reported cooperative motivations following this distinction and extended detail is provided in "Theme 3: Self-Reported Motivations for Cooperation" and "Theme 4: Changes in Behavior over Time" in the Methods. The evidence indicates that these two aspects of cooperative behavior are driven by different types of motivations and that the reported motivations associated with willingness to compensate for another member are particularly prone to decay over time. The observation of this motivational decay in the long-term and quantifying its associated consequences on group outcomes are distinctively enabled by the five years of field data. We believe these insights may prove to be widely applicable in the study of cooperation, as distributional conflict is not specific to group lending but rather is inherent in the threshold dilemma.

(2) The authors rule out cumulative and strategic mechanisms as explanations for their data, consistent with the lab experiment literature, and propose plausible behavioral mechanisms such as social norms and temporal boundaries, which I find to be reasonable. Unfortunately, the empirical exercise does not conclusively pinpoint the exact behavioral mechanisms responsible for the observed patterns of cooperation, primarily due to the absence of analysis on the individual-level data. Without a clear identification of these mechanisms, the study stops short of providing a deeper understanding of the causal processes at play, which limits its ability to suggest concrete, targeted interventions or policy recommendations.

Thank you for encouraging us to provide more specificity regarding causal mechanisms and practical implications. The revised manuscript now has empirical depth based on the client interviews that enables us to speak directly to the behavioral mechanisms at play. In the revised manuscript we now identify the mechanisms and provide individual-level evidence in the section "Behavioral Mechanisms."

A unique practical benefit of this study is the ability to show the long-term effects of the behavioral mechanisms in a natural context. This provides a more direct evidence base for potential interventions and policies. We highlight the persistence or decay of the

behavioral mechanisms on collective outcomes over the five-year period, in particular regarding the long-term risk of behavioral decay even in successful groups; the impact and persistence of group restarts; and the accelerating rate of decline after each restart.

With the refined theoretical framework and detail on the behavioral mechanisms, the manuscript is now able to suggest a particularly relevant class of solutions. In the Discussion section we suggest: "Incorporating behavioral dynamics into models of collective action may be essential for explaining why cooperation falters even under structurally favorable conditions. For institutions reliant on long-term cooperation, the findings suggest a class of solutions aimed at mitigating behavioral decay, such as contribution automation, strategic resets, intrinsic motivation enhancement, or habit formation."

Additionally, I believe the authors could enhance their description of the empirical setting. For instance, on page 3, lines 107-108, it is stated that "If a loan is not repaid in full, all members lose access to future credit." Yet, Figures 1 and 2 suggest that some groups that did not fully repay in one cycle were able to participate in subsequent cycles. Clarifying this discrepancy would help readers better comprehend the study's context.

Thank you for bringing our attention to the lack of clarity regarding the eligibility criteria and the loan reapproval process. We have now concisely integrated these topics into the main text and provide detailed description in the Methods:

"Eligibility for a subsequent loan cycle depends on a group's repayment history in the previous cycle. Full repayment is required, but timeliness of payments throughout the whole loan cycle also influences the decision. The institution incentivizes better group repayment with greater potential increases in the subsequent loan amounts. Groups that repay in full and on time receive the standard maximum loan increase for the next cycle. If the loan is paid in full, but with delayed payment(s), the group may be assigned a lesser loan increase, no loan increase, or no loan renewal, based on the frequency and severity of the delayed payment(s). The outcome is the same for all group members. Decisions regarding loan renewal and amount are made jointly by the group's Loan Officer and Loan Portfolio Manager. Variation in subsequent loan amounts implies that a single provision point for the collective good is a theoretical simplification of the full set of potential group outcomes. However, the core distributional conflict and individual incentive to contribute as little as possible while reaching each provision point remains consistent."

"Group loan enforcement operates in two phases: (1) internal enforcement of active loans and (2) institutional enforcement of delinquent loans. During the active phase, group members are responsible for enforcing on each other. Members use a variety of positive and negative enforcement mechanisms, such as encouragement, social pressure, embarrassment, and ostracism. During this phase, the collective good is still achievable, i.e. access to future credit. After approximately 30 days of delayed payment, enforcement gradually shifts to the institutional phase. In this phase, the possibility of qualifying for a subsequent loan rapidly declines. The loan is classified as inactive, and the institution

initiates formal recovery procedures – typically involving debt collectors and legal threats. These efforts generally continue for at least a year until the group repays or the organization officially records the loan as a write-off and ceases efforts at collection. In the Modelling section below, we define cooperative outcome measures using a 30-day window to capture meaningful behavior attributable to internal group dynamics, distinct from institutional debt recovery efforts.”

In addition, your comments motivated us to make features of the underlying quantitative data more transparent throughout the manuscript. We hope you agree that our revised figures make the underlying data more intuitive for the reader.

In closing, we appreciate the time you have dedicated to improving this study. We hope we have successfully addressed all your comments.

Response to Reviewers

Nature

Manuscript Title: Punctuated Decline of Human Cooperation

General Comments:

We would like to express our sincere appreciation to the editor and reviewers for their thoughtful feedback, which has been both insightful and extremely helpful in guiding further improvements to the manuscript. Since the previous version, we have made revisions in response to the feedback from the editor and all the reviewers. As suggested, we have conceptually refined the identified behavioral mechanisms and conducted additional analyses to provide a more complete and systematic account of the observed group behavior in the qualitative sample. We begin by offering a summary of the most substantive updates here, before responding to each comment point-by-point.

Summary of Revisions:

- 1. **Conceptual refinement of cooperative motivation in threshold dilemmas:** We have revised the text to clarify and better communicate the concept of cooperative motivation in threshold dilemmas. The refinements derive from theory generalizable to a large class of threshold dilemmas across diverse contexts and are empirically substantiated with evidence from joint-liability lending.*
- 2. **Analysis of heterogeneity of cooperative dynamics in the qualitative sample:** We analyzed the data to code for variation in the severity of individual defection behavior using the reported frequency and timing of non-contribution. For example, the analysis distinguishes between borrowers who engaged in more mild, intermittent defection (e.g., occasional missed contributions followed by resumption) from borrowers who engaged in more severe walk-away defection (e.g., ceasing contributions midway through the loan cycle and not resuming). The accounts of individual defections were cross-validated with secondary interviews and group-level repayment records from the lending institution.*

Building on this individual-level detail, we then categorized each group according to the collective pattern of defections by its members, thereby capturing within-group dynamics of cooperation. This approach allowed us to quantify and communicate how often groups exhibit gradual increases in member defections from more severe patterns, such as cascades of defection or full collapse. In the revised manuscript, we describe the formal coding criteria in detail, apply it to the entire qualitative sample of joint-liability groups, and provide full frequency counts for each type of cooperative behavior.

This new analysis provides several benefits: (a) it provides a systematic and more complete account of within-group behavior in the qualitative sample. The analysis documents the full spectrum of defection behavior, showing more severe, but less common behavior with their relative frequencies; (b) by quantifying the frequency of different behaviors in the qualitative sample, it better substantiates how we arrived at our primary interpretation of the observed mechanisms; (c) the coded qualitative patterns are consistent with the group-level repayment data recorded by the lending institution, providing an additional empirical bridge between the interview analysis and the aggregate quantitative trends.

The analysis has been incorporated in the following locations:

- *Table S16. Heterogeneity of individual contributions and frequency of cooperative patterns from client interviews;*
- *Methods subsection on “Heterogeneity of Cooperative Dynamics” provides full coding criteria and analysis process;*
- *The findings are incorporated in the “Behavioral Mechanisms” section of the main text.*

- 3. Further empirical detail and validation of the identified mechanisms:** *We have broadened and deepened the empirical evidence demonstrating that changes in defection behavior over time are driven by systematic shifts in cooperative motivation and effort. The revised manuscript draws on more examples from clients themselves, but also incorporates an additional data source that we had not previously used for the purposes of identifying mechanisms: staff interviews.*

In the prior version of the manuscript, we used staff interviews to confirm the nature of the social dilemma and to clarify how joint-liability lending was implemented in practice. However, motivated by the reviewers’ comments, we recognized that the staff accounts offer a valuable external perspective on group behavior based on their substantial experience observing and interacting with joint-liability groups. To strengthen the breadth and triangulation of our data, we now include direct quotations from staff relating to cooperative dynamics. Such third-person accounts are particularly useful for understanding socially undesirable behaviors, such as why group members defect. The qualitative analysis now triangulates evidence from (a) first-hand accounts from defectors, (b) within-group descriptions from other members, and (c) third-person perspectives from outside the group itself.

The analysis has been incorporated in the following locations:

- *Table S13. Interview sample staff descriptive statistics and anonymized identifiers;*

- *Further client and staff examples are integrated throughout the coded themes and direct quotations in the Supplementary Materials;*
- *The evidence is incorporated in the "Behavioral Mechanisms" section of the main text.*

4. Limitations and Future Research: *We now provide a clearer articulation of the study's limitations and their implications for future research. In particular, we note that future research would benefit from further distinguishing the psychological processes underlying cooperative decay in more controlled settings and from examining its potential interaction with other cooperative mechanisms across diverse empirical contexts.*

Again, we greatly appreciate the editor and all the reviewers for their constructive feedback. We hope that you find our revisions have addressed your comments thoughtfully and effectively. Below is a point-by-point response. The editor's and reviewers' original comments are left-justified and non-italic. Our interleaved responses are indented and italic.

Referees' comments:

Referee #1 (Remarks to the Author):

As written in my first report, I consider the study as relevant and rich. The point that I raised in the previous version have been satisfactorily addressed. I like in particular that economic environment, which a step level public goods, is now clearer. There are only three minor issues that should be addressed in a further revision.

We are happy to hear that you found the integration of the step level public goods framing effective. Thank you for helping us substantially improve the manuscript.

1. The first paragraph ends with “However, an essential unresolved question is why, even under favorable conditions, cooperation is dynamic and prone to decline.” The next paragraph presents several approaches to explain this phenomenon. The authors of these papers might disagree with the view that the question is unresolved. A better formulation could be to replace “unresolved” with “still disputed”.

We appreciate the recommendation. We have made the suggested change in the text.

2. Table 2. The regressions only include an interaction term. This makes the interpretation of this variable difficult. It implicitly assumes that there is no round effect in Cycle 0 and no cycle effect in round 0.

Thank you for the helpful suggestion. Previously the regressions in Table 2 included Round, Loan Cycle, and Round x Loan Cycle interaction terms, but the Table only showed the results for the interaction term. Table 2 has been updated and now includes the results for Round, Loan Cycle, and the Round x Loan Cycle interaction.

3. Figure 4a) It is unclear what the pyramid of under “Lending Staff” means.

We have made two edits to clarify this. (1) In the graphic we have updated the label above the pyramid to now read as “Lending Staff Hierarchy” under “Purposive Sampling.” (2) We have also updated the caption of the figure to further clarify the schematic, “A purposive sample of.... staff from multiple levels of the lending institution hierarchy was added to cross-validate and enhance the internal validity of the findings.”

Once again, we are grateful for your comments, which have helped us improve the manuscript in its theoretical, empirical, and practical aspects.

Referee #2 (Remarks to the Author):

The authors have improved the paper substantially, and I appreciate the effort they have taken to meaningfully address the concerns raised. Specifically, they have added more details on the ethnographic context and delineated the incentive structure of the group loan payment more clearly; they have clarified the theoretical connection of the study to the literature on cooperation by discussing the loan scheme as a threshold public goods game; they have provided crucial data from interviews they conducted with a subset of the participants that supports the inferences they draw from the aggregate loan repayment data from the lender. These revisions have addressed some of the key concerns I had. I now more clearly see the value of the study, and support publication.

We are delighted to hear that you found the revisions to have substantially strengthened the manuscript. Thank you for your support and effort in improving this study.

I do however have some points I'd like to raise that I think still needs to be addressed.

1. I do not think the term "punctuated" communicates the results of the paper effectively. The pattern being shown is that of "decline and rebound". In evolution, punctuated implies "stasis and change", and we don't see any stasis in cooperation rates here. So it might be confusing to use the term in this way given its usage in biology.

Thank you for raising your concerns about the use of the term "punctuated" in the title and the possibility that some readers might interpret it differently than we intended. Obviously, there is no question that we would not want readers from fields such as evolutionary biology to be misled by this phrasing. In articulating our response, we feel that it is useful to provide some background on what motivated our choice, and why we think it is an effective approach for communicating the behavior that we observed in the study.

When we were deciding on a title for our manuscript, we were directly inspired by Stephen Jay Gould and Niles Eldredge's important work on "punctuated equilibrium," published in the 1970s (e.g. Gould & Eldredge, 1977), with its longer-term influence reflected by a subsequent review article published in Nature (Gould & Eldredge, 1993). In Gould and Eldredge's original context the term "punctuated equilibrium" proved to be a very effective way of communicating the notion of discontinuous rather than gradualist evolutionary processes. As you suggested, it reflected long periods of ecosystem stability (equilibrium states) punctuated by rapid bursts of speciation and the sudden extinctions of species. Since the original use of the term in the context of evolutionary biology, "punctuated equilibrium" has been extended to other disciplines and contexts, such as theories of public policy (Givel, 2010) and models of technology diffusion (Loch & Huberman, 1999).

Abstracting across these examples, we agree that when "punctuated" is used in conjunction with "equilibrium" it corresponds to a class of dynamics in systems where extended periods of stability are punctuated by intermittent rapid change. By contrast,

the dynamic behavior that we report in our study shows consistent decline, punctuated by sudden rebounds. Although the underlying dynamics for these two types of patterns are indeed different (i.e. equilibrium vs. consistent decline), in both cases the respective dynamics are punctuated by short intervals of rapid change. This is why we chose to label the patterns that we observe “punctuated decline,” in contradistinction to “punctuated equilibrium,” reflecting both the similarities and differences in the two types of dynamics. The use of the word “punctuated” on its own could indeed potentially give rise to confusion, but we believe that this is not the case when it is used in conjunction with the word “decline.”

It is worth adding that we have carefully considered a range of other formulations, but find “punctuated decline” the most effective communication of the reported behavior. For example, the concern with alternative titles based on “decline and rebound” is that they lead to a different form of potential misinterpretation. We believe that they could be seen to suggest a more balanced oscillation between cooperation and defection. By contrast, the predominant pattern we observe is one of accelerating aggregate decline, only temporarily punctuated by short rebounds.

Given the above, we believe the use of “punctuated decline” is accurate and effective in describing the observed behavior and would like to maintain its usage in the title. However, we are of course open to changing the nomenclature in the title if the perceived risk of cross-disciplinary confusion persists despite the explanation that we have offered.

- *Gould, SJ, & Eldredge, N (1977). Punctuated equilibria: The tempo and mode of evolution of reconsidered. *Paleobiology*, 3(2), 115-151.*
- *Gould, SJ, & Eldredge, N (1993). Punctuated equilibrium comes of age. *Nature*, 366(6452), 223-227.*
- *Loch, CH, & Huberman, BA (1999). A punctuated-equilibrium model of technology diffusion. *Management Science*, 45(2),160-177.*
- *Givel, M (2010). The evolution of the theoretical foundations of punctuated equilibrium theory in public policy. *Review of Policy Research*, 27(2), 187-198.*

2. The structure of the Introduction is unhelpful for readers to understand the paper. It spends too much real estate on stating the importance of the paper rather than describing the empirical context so that a reader can come to that conclusion themselves. For example, after a terse paragraph 2 describing the empirical context, the paper jumps in paragraph 3 to explain the uniqueness of the empirical context. Without reading the supplementary material a reader does not have enough information to evaluate the claims being made in paragraph 3. The rest of the Introduction describes the results and its broader implications, which again I think jumps the gun and comes at the expense of providing the reader with enough details on how the loan scheme worked.

Thank you for drawing our attention to this point. We agree with your assessment and have accordingly revised the introduction of the main text. The paragraph describing the empirical context has been expanded to incorporate key details regarding how joint

liability works. In particular, we have added that group members are expected to compensate for each other, both in terms of institutional expectation and as empirically demonstrated in the field by borrowers. In addition, the revised introduction explains that members employ informal social sanctions to enforce cooperation within the groups. This information was previously introduced later in the manuscript. We now appreciate that including this type of information early is essential for helping readers understand why the cooperative dilemma is central to group lending. In line with your guidance, we have also streamlined or removed material from other paragraphs of the introduction to use the space more effectively.

3. The authors should leverage the empirical findings from their previous paper to strengthen the interpretation of the results of this paper. The previous paper (reference 41) is important background work, and yet it is being referred to very superficially. By describing their previous results from the analysis of the interview data, in addition to the novel analysis of the interview data done in this paper, the authors are in a stronger position to pinpoint (and reassure readers of) the underlying cooperative behavioral mechanisms driving the aggregate data.

We are pleased that you found our previous paper on this topic useful. Following your guidance, we now leverage the paper more in both the main text – as useful background work on how joint liability groups informally enforce cooperation – and in the qualitative methods section, where it provides clarity regarding how cooperative motivation and effort shape group dynamics. In particular, we now refer to the prior paper more explicitly to communicate how natural variation across groups – in their social and spatial structures – has a direct connection to collective outcomes across groups from a cross-sectional perspective. Such information helps to establish the underlying nature of cooperative tension within groups, providing a useful foundation before addressing the dynamics of longitudinal cooperation, as undertaken in the present study. Thank you for this recommendation.

4. I am still unclear on the behavioral mechanism being implicated here. The term “behavioral mechanisms” blackboxes all kinds of psychological biases and motivations, and the inability to disambiguate which behavioral mechanisms are driving the decline in cooperation a weakness that should be clearly articulated. The interview data did convince me that there are cooperative motives at play when paying one's own share, and when stepping in to pay someone else's share. But the reason why people don't pay their own share is still unclear. For instance it could be because some people are conditional cooperators and so their motivation to pay their own share decreases when they sense cheap riding going on in their group. On the other hand it could be that everyone slowly becomes lethargic about payment as time passes. This latter scenario does not implicate motives stemming from a conditional cooperative psychology. Rather, it is more in line with what you might see if you signed people up for an exercise program at a gym, and checked in with them every 6 months to assess their health metrics and give them a pep talk. There will likely be a decline in gym attendance over the 6 month period, and a rebound when there's a check-in. These behavioral trends would stem from motives that are not directly (or

exclusively) related to cooperation. At the least, I think the paper should more clearly state this point.

Thank you for pointing out that the manuscript could provide additional clarity on this topic. We have made several improvements to the manuscript to address this, summarized here in three points.

(1) The distinction between other-regarding motives to cooperate and long-term self-regarding motives to cooperate. As you rightly note, part of the decline may result from decay in self-serving behavior, aptly illustrated by the gym membership metaphor. Moreover, we agree that this aspect of the behavior is not exclusive to cooperation and is not a theoretical factor in some common depictions of cooperation, such as the linear public goods game where full free-riding constitutes the dominant strategy. Therefore, we greatly appreciate that these distinctions should be clearly communicated to the reader because it enhances the study's conceptual clarity and its practical relevance. We have updated the text to consistently describe this theoretically and empirically as follows.

First, we explicate the theoretical framework of a threshold dilemma, highlighting that incentives to contribute to a threshold collective good can derive from both other-regarding interests and from self-interest. This framing is crucial for clarifying that these motives are not specific to joint-liability lending but characteristic of a broad class of threshold cooperative dilemmas observed across diverse contexts. Second, before addressing the decline in group outcomes, we now give greater emphasis to evidence from the qualitative data that the motivations to contribute to group repayment indeed range from other-regarding motives to self-interest, citing clients' reasons for contributing that range from individual economic self-interest to other-regarding motives such as group solidarity. Third, we now more explicitly indicate that the decline in motivation to contribute to the group loan derives from both a decay in other-regarding interests – with empirical examples of reduced concern for group members and the weakening of previously strong social ties – and a decay in long-term self-interest – with empirical examples of shifts towards more immediate personal consumption. We note that we do find evidence compatible with a psychology of conditional cooperation, but also clearly position it as not being the only driver of decline in this context. In the revised text, our goal has been to accurately distinguish and empirically demonstrate the multiple sources contributing to the decline in motivation, while communicating what we observe as the primary drivers based on the weight of the empirical evidence.

To this end, you will see that we have added quotations and supporting analysis to the Supplementary Materials and revised the “Behavioral Mechanisms” section of the main text to highlight more clearly the distinctions that you raised above. In the point below, we describe how we take an additional step to code and present the frequency of different patterns of defection behavior within the groups to further aid the readers' ability to evaluate the qualitative data for themselves.

(2) Heterogeneity in defection behavior. Your queries regarding the extent to which members uniformly experience motivational decay, the extent to which their behaviors influence one another's decisions, and, together with the feedback from Reviewer 3,

motivated us to produce a substantial addition to the analysis presented in the paper. As introduced in the General Comments above, we used the qualitative data to code for the variation in the severity of individual defection. For example, we can distinguish an individual who was reported to have skipped one or two payments during a loan cycle from an individual who was reported to have stopped making payments half-way through the loan cycle and never contributed again. We coded the former as “intermittent defection” and the latter as “walk-away defection.” The full coding criteria and analysis process are provided in the Methods section. Building on this individual-level detail, we then categorized each group according to its predominant cooperative pattern, cross-validating it with secondary interviews and the group-level repayment data. The frequencies of these cooperative patterns are summarized in the new Table S16 and incorporated into the “Behavioral Mechanisms” section of the main text. This analysis offers a more complete account of within-group behavior and substantiates how we arrived at our primary interpretation of the qualitative data.

The analysis, as we would expect, reveals substantial heterogeneity in individual defection behavior. A large percent of groups, 91.4%, reported some form of defection by one or more of its members across the whole loan cycle. The most prevalent pattern of defection identified in our analysis of the interviews was “Intermittent Defection,” reported in 62.9% of groups. It is reflective of gradual decay with an increasing likelihood of individuals burdening the group over time, but still attempting to maintain access to the collective good. In 17.1% of groups we found evidence of more abrupt “Walk-away Defection” in which an individual abandons their cooperative responsibility, but the remaining members still attempt to complete the loan cycle. The final two categories reflect more severe group decline in which defectors cause cooperation to unravel or fully breakdown. We found 8.6% of groups exhibited a cascade of escalating defections; and 2.9% of groups exhibited full collapse, in which all members ultimately defected. This additional analysis allows the reader to better assess the role of gradual, intermittent decay, while also understanding the relative frequencies of less common cooperative patterns in the group lending dataset.

We believe this analysis provides a valuable additional picture of within-group behavior, particularly the prevalence and extent of distributional conflict, that would not be readily apparent in the quantitative dataset alone. The analysis strengthens the bridge between the qualitative accounts and the aggregate trends observed in the quantitative dataset.

(3) Limitations. As you rightly point out, there are limitations to the level of specificity that can be drawn from our field data. We now articulate the study’s limitations more clearly and highlight the implications for future research. In particular, we note that future research would benefit from further distinguishing the psychological processes underlying cooperative decay and its potential interaction with additional cooperative mechanisms, which could be fruitfully examined through more controlled experimental designs and the study of other cooperative settings.

Overall, I think this paper contributes valuable and unique empirical data on real world behavior in an important behavioral domain. It would instigate further studies of suitable natural contexts

in which long-term cooperation can be analyzed. Such data would greatly benefit a field that has so far been dominated by theoretical models and lab experiments, which although valuable, cannot be relied on to illuminate motivations that may persist or decline over longer time scales.

We greatly appreciate your thoughtful and encouraging comments. We share your view that advancing empirical research in natural contexts is important for understanding how cooperation unfolds over longer time scales. We thank you for helping us make this study clearer and more valuable to readers.

Referee #3 (Remarks to the Author):

The revised manuscript is a substantial improvement over the previous version. I appreciate the authors' efforts, especially the addition of new figures, the clearer contextual background, the new theoretical framework based on a threshold public goods game, and---most importantly---the inclusion of individual-level survey data, which provides micro-level evidence for the proposed mechanism. The study's value is now much more evident: the long-run decay in cooperation, even after several stable loan cycles, is something that cannot be captured in lab settings and has important practical implications.

Thank you for the positive feedback. Your comments have been very helpful in refining the analysis and improving the communication of the findings.

However, I am not fully convinced by the behavioral mechanism proposed by the authors and hope they can further engage with the client interview data. My concern is as follows:

The authors suggest that the key mechanism behind the decline in cooperation is a gradual erosion of cooperative motivation-i.e., a natural shift from other-regarding to more self-interested mindset. But in my view, this account is incomplete and does not fully explain the observed aggregate decline in cooperation. Most of the interviews cited in the supplementary materials do not come from individuals whose own motivation declined (except the final one), but rather from frustrated group members who are angry at others for ceasing to contribute. This suggests to me that the drop in cooperation may have been triggered by a few "bad apples"-individuals who stopped contributing, thereby making it no longer a best response for others to continue contributing toward the threshold.

What remains underexplored is why these "bad apples" emerged in the first place. As the authors note, clients understood the importance of meeting the threshold to maintain access to future loans. Yet some chose to use the loan for personal expenses, stopped repaying, and seemed to disengage entirely, or even burned bridges with their fellow group members. This behavior appears more like a full breakdown of participation than simple cheap-riding. I therefore encourage the authors to revisit the interviews of those who ceased contributing-there may be clues to uncover, such as whether some clients simply no longer needed access to future loans.

Thank you for your constructive comments and for pointing out specific ways that the manuscript could be improved. Motivated by your comments and by those of Reviewer 2, our revision approach consisted of two primary additional analyses. (1) We greatly appreciate the value of providing a more complete and systematic account of the variation in defection behavior observed in the qualitative sample. We have formally coded and presented the heterogeneity of individual contributions within groups. The analysis clarifies to readers how we arrived at our interpretation and shows the full spectrum of defection behaviors and their frequencies. (2) Within that broader empirical framing, we have strengthened the evidence base and communication of the primary observed behavioral mechanism. We have conceptually clarified cooperative motivation

in threshold dilemmas and have integrated additional empirical data from client and staff interviews. Here we provided a summary of the additional analyses, particularly in response to your comments above. Full details of the analyses and their implications have been incorporated into the revised manuscript.

(1) Heterogeneity of cooperative dynamics and a more complete account of defection behavior.

We have gone back to the interview data and now systematically differentiate the severity in defection behavior at the individual-level reported in each interview. For example, we distinguish between borrowers who engage in more mild defection (intermittently defecting but continuing to contribute) versus more severe walk-away defection that you noted above (such as stopping contributions half-way through the loan cycle and never contributing again). The coding and analysis process can be conceptualized as involving two key steps: (i) the classification of each group member's cooperation/defection behavior based on the information reported in the interviews, and (ii) the categorization of groups into various patterns based on the collective sequence of individual member's behaviors. This second step allows us to examine how often the more severe walk-away defection produces more dramatic consequences in the group, such as cascades of defection or full collapse. To conduct this analysis we defined formal coding criteria (detailed in the Methods) and cross-validated the individual- and group-level patterns with the secondary interviews and the group-level repayment data. The frequencies of these cooperative patterns are summarized in the new Table S16. You will also see that the table includes schematics that are useful for visualizing the differences in the cooperative patterns.

The results show the heterogeneity in individual defection behavior - from participants who maintained full cooperation across all rounds to those who rapidly shifted to defection and never contributed again. We now recognize that directly communicating this empirical variation in the qualitative sample is important for the reader. The results also show the frequency of different cooperative patterns within each group to better understand the collective consequences. The most prevalent pattern was intermittent defection, reported in 62.9% of groups. It is reflective of gradual decay with an increasing likelihood of individuals burdening the group over time, but still attempting to maintain access to the collective good. More abrupt shifts in cooperation were also present: 17.1% of groups exhibited walk-away defections, in which members defected for more than one round without returning to contribute. In these groups (as distinct from cascades), the remaining members continued to contribute after the walk-away defection and completed payment of the loan cycle – interviews indicating primarily out of regard for the remaining members or a sense of obligation. We found 8.6% of groups exhibited a cascade of escalating defections; and 2.9% of groups exhibited full-collapse, in which all members ultimately defected. We note in the text that these patterns of cascades and full collapse are indicative of walk-away defections having triggered additional mechanisms for non-cooperation, such as social contagion or financial reassessment, as you suggested in your comments. We also point out in the text that the interaction between

changes in motivation and other cooperative mechanisms represents a valuable topic for future research.

This analysis speaks directly to the questions you raised regarding whether the decline in cooperation tends to be the result of a few individuals who stopped contributing or a more generalized behavior. In the sample of interviewed groups, which is representative of groups in the quantitative dataset, the most commonly exhibited pattern was intermittent defection, which is consistent with members cheap riding but still attempting to contribute and maintain access to group credit. We appreciate you motivating us to situate the frequency and role of these defection patterns in context.

(2) Further empirical evidence and clarity regarding the identified behavioral mechanism.

Your comments also motivated us to provide more analysis and clarity regarding why group members defect in the first place. We now draw greater attention in the theoretical framework and in the analysis of interviews to the fact that incentives to contribute to a threshold collective good can derive from other-regarding interests and from self-interest. This framing is crucial for clarifying that these motives are not specific to joint-liability lending but characteristic of a broad class of threshold cooperative dilemmas observed across diverse contexts. Before addressing mechanisms for the decline in group outcomes, we now give evidence from the qualitative data that the motivations to contribute to group repayment indeed range from other-regarding motives to self-interest, citing clients' reasons for contributing that range from individual economic interest to other-regarding motives such as group solidarity.

As mentioned in the General Comments, the qualitative analysis now triangulates evidence from (a) first-person accounts from defectors, (b) third-person accounts from other members, and (c) third-person accounts from outside the group itself, i.e. staff interviews. You will see that we have added quotations and supporting analysis to the Supplementary Materials and revised the "Behavioral Mechanisms" section of the main text. Across these descriptions, we observe that as time passes members became more likely to allocate their own potential monthly repayment towards self-interested uses at the cost of burdening the group. Consistent with the conceptual distinction above, the decline in motivation to contribute to the group reflects both a decay in other-regarding interests – with examples of reduced concern for group members and the weakening of previously strong social ties – and a decay in long-term self-interest – with examples of shifts towards more immediate personal consumption. As you rightly observed in your comments, there is heterogeneity in the severity of the behavior change, and as we documented in the heterogeneity analysis summarized above, the more severe walk-away defections can potentially trigger additional mechanisms for non-cooperation. However, for both intermittent defectors and walk-away defectors, the interview data suggest that defection is predominately the result of a behavioral shift towards self-interested uses, rather than a response to unexpected exogenous shocks, a change in credit needs, or notably different financial circumstances.

When interpreting the qualitative data, we wish to draw attention to the particular value of staff interviews when they are used in conjunction with client interviews. Your comments motivated us to go back to these interviews and read them not just to corroborate the structure of joint liability, but now also to obtain additional evidence on mechanisms. Their value derives from the observation that borrowers may attempt to protect their self-image by portraying non-contribution as involuntary or justified by circumstance, whereas third-person perspectives more readily identify deliberate shifts in behavior even when they are socially undesirable. For example, a staff member described how a change in willingness to contribute may often be portrayed as being beyond the borrower's control, "They complain that the sales are not going fast the most, we are not getting sales.... But when you look at all their business, actually, it's very good. Very good looking" (Staff Interview S4). The combined evidence from the staff interviews corroborated that the decline within and across cycles is most commonly a result of decay in cooperative motivation, rather than meaningful changes in borrowers' financial circumstances. "Initially, they have the kind of zeal, motivation to repay..." (Staff Interview S7), but the motivation is described as naturally eroding with each monthly payment, as borrowers are increasingly likely to redirect potential repayments toward personal use. Table S13 presents descriptive statistics and anonymized IDs for the interviewed staff members. Their quotations are now integrated throughout the qualitative analysis as an additional source of validation for the cooperative mechanisms identified.

Additionally, I am curious whether the client interviews might shed light on the restart effect. While the interviews suggest a shift from other-regarding motives to a more self-interested mindset within a cycle, is there any indication of a reversal in mindset across cycles? Any supporting evidence on this point would be helpful.

Thank you for encouraging us to examine the interviews for indications of a reversal in mindset. Information on a reversal of mindset did emerge in some staff interviews, although our client interview data is rather limited on this topic, as the client interview questions were not designed to address this directly. However, your comments prompted us to consider a broader view of the interview data for additional insights on this topic. As a result, we have included a new qualitative subsection that focuses on attempts at reducing or reversing behavioral decay – drawing on both client and staff accounts. Many clients informally try to slow the decay in cooperation throughout the loan. With a similar intent of resetting cooperative behavior, the lending institution "resensitizes" clients at the restart of a loan cycle. The combination of client and staff perspectives is constructive, as they offer different perspectives on efforts to mitigate the same problem.

Here we provide a summary of the key insights: To combat cooperative decay, both clients and staff repeatedly emphasize two aspects of cooperative motivation: the financial consequences of successfully repaying, which may revive long-term self-interest, and the social obligations to fellow group members, which may draw on more emotional and prosocial sentiments. The examples consistently highlight the significance of repetition in maintaining motivation. Clients note that stating the consequences and obligations once is not enough: "I continue to say we must work very hard to pay this

money, otherwise they will not give us another loan” (Client Interview G35.C64) and “We have sympathy to each other. We normally encourage each other....” (Client Interview G17.C35). Staff interviews reveal that loan officers are aware that group motivation tends to decline slowly within and across loan cycles. They note that groups with members that actively visit each other to maintain collective motivation tend to perform better. More formally, the institution’s resensitization process takes place with all group members at the start of each new loan cycle, reemphasizing the principles of group solidarity lending and the shared financial consequences. The interviews suggest that these efforts do play a role in resetting cooperative motivation, as one staff member describes, “They begin to drag their feet.... So we [resensitize] them.... and that will, like, give them a rethink that, ‘Oh, I should stop this....’ So they will catch up again” (Staff Interview 7). However, interviews also indicate varying degrees of effectiveness at the individual-level, “While others, you keep reminding them, you keep reminding them, they will not pay” (Client Interview G25.C50).

The implications are integrated into the “Behavioral Mechanisms” section of the main text. Full client and staff quotes illustrating these concepts are elaborated in the Supplementary Materials.

In closing, we greatly appreciate your constructive feedback. We hope that the revised manuscript more clearly communicates the primary findings from the empirical data, while also effectively conveying the underlying variation.

Response to Reviewers

Nature

Manuscript Title: Punctuated Decline of Human Cooperation

We would like to express our sincere appreciation to the editor and reviewers for their very constructive feedback throughout the review process. Below is a point-by-point response to the remaining comments from the referees. The original comments are left-justified and non-italic. Our interleaved responses are indented and italic.

Referee #2 (Remarks to the Author):

I appreciate the authors' efforts in revising the paper and clarifying the few topics that were unclear in earlier drafts. This paper will be a useful contribution to the literature. I recommend publication.

Minor suggestions:

Line 98-99 "The results advance our understanding of cooperative dynamics and have significant theoretical and practical implications".

This is a fairly general and therefore vague statement. Would be better if the topic sentence of this paragraph more specifically cued the reader to the specific patterns being discussed in the remainder of the paragraph.

Thank you for pointing out this sentence. We agree that the sentence does not offer enough specific value. We have removed the sentence and now proceed directly to the substantive results in this paragraph.

Line 146: "is highly similar". Maybe better to say "has key similarities" or "corresponds"

Thank you for suggesting how the phrasing could be better communicated. We have now updated the sentence to use the suggested phrase "has key similarities."

Referee #3 (Remarks to the Author):

I am satisfied with the authors' responses to my comments from the last round. The newly added interviews and supplementary analyses provide readers with a much clearer understanding of the behavioral mechanisms underlying the decline in cooperation. I appreciate the authors' substantial revision efforts, which now make the paper feel complete: from mechanism to phenomenon.

We are pleased to hear that our revisions substantially strengthened the manuscript. Thank you for providing clear guidance on how to improve the study.